# Use of signals of positive and negative selection to distinguish cancer genes and passenger genes

László Bányai[1], Maria Trexler[1], Krisztina Kerekes[1], Orsolya Csuka[2], László Patthy[1]*

[1]Institute of Enzymology, Research Centre for Natural Sciences, Budapest, Hungary; [2]Department of Pathogenetics, National Institute of Oncology, Budapest, Hungary

**Abstract** A major goal of cancer genomics is to identify all genes that play critical roles in carcinogenesis. Most approaches focused on genes positively selected for mutations that drive carcinogenesis and neglected the role of negative selection. Some studies have actually concluded that negative selection has no role in cancer evolution. We have re-examined the role of negative selection in tumor evolution through the analysis of the patterns of somatic mutations affecting the coding sequences of human genes. Our analyses have confirmed that tumor suppressor genes are positively selected for inactivating mutations, oncogenes, however, were found to display signals of both negative selection for inactivating mutations and positive selection for activating mutations. Significantly, we have identified numerous human genes that show signs of strong negative selection during tumor evolution, suggesting that their functional integrity is essential for the growth and survival of tumor cells.

## Introduction

### Genetic, epigenetic, transcriptomic, and proteomic changes driving carcinogenesis

In the last two decades, the rapid advance in genomics, epigenomics, transcriptomics, and proteomics permitted an insight into the molecular basis of carcinogenesis. These studies have confirmed that tumors evolve from normal tissues by acquiring a series of genetic, epigenetic, transcriptomic, and proteomic changes with concomitant alterations in the control of the proliferation, survival, and spread of affected cells.

The genes that play key roles in carcinogenesis are usually assigned to two major categories: proto-oncogenes that have the potential to promote carcinogenesis when activated or overexpressed and tumor suppressor genes (TSGs) that promote carcinogenesis when inactivated or repressed.

Several alternative mechanisms can modify the structure or expression of a gene in a way that promotes carcinogenesis. These include subtle genetic changes (single nucleotide substitutions, short indels), major genetic events (deletion, amplification, translocation and fusion of genes to other genetic elements), as well as epigenetic changes affecting the expression of cancer genes. These mechanisms are not mutually exclusive: there are many examples illustrating the point that multiple types of the above mechanisms may convert the wild-type form of a cancer gene to a driver gene.

Exomic studies of common solid tumors revealed that usually several cancer genes harbor subtle somatic mutations (point mutations, short deletions, and insertions) in their translated regions but malignancy-driving subtle mutations can also occur in all genetic elements outside the coding region, namely in enhancer, silencer, insulator, and promoter regions as well as in 5'- and 3'-

*For correspondence:
patthy.laszlo@ttk.mta.hu

Competing interests: The authors declare that no competing interests exist.

**eLife digest** The DNA in the cells of the human body is usually copied correctly when a cell divides. However, errors (mutations) are sometimes introduced during the copying process. Although the majority of mutations have no major impact on cells, many mutations are harmful: they decrease the ability of cells to survive. There are, however, mutations that can lead to cells dividing more frequently or gaining the ability to spread, which can lead to cancer. These mutations are known as 'driver mutations' because they drive the growth of tumors. Since such 'driver mutations' provide a growth advantage to tumor cells, they are subject to positive selection, this is, their frequency in the tumor increases over time. Because of their selective advantage, driver mutations accumulate at significantly higher rates than the neutral 'passenger mutations' that do not play a role in tumor growth.

Genes that carry driver mutations are called driver genes, while genes that carry only passenger mutations are known as passenger genes. Certain genes, however, do not fit into either category. For example, some genes that are essential for tumor growth must get rid of harmful mutations to maintain activity. Mutations of such 'tumor essential genes' are thus subject to 'negative' or 'purifying selection'.

A major goal of cancer research is to identify genes that play critical roles in tumor growth. Earlier studies have identified numerous driver genes positively selected for driver mutations, exploiting the fact that driver genes show significantly higher mutation rates than passenger genes. Identification of tumor essential genes, however, is inherently more difficult since the paucity of mutations of negatively selected genes hinders the analysis of the mutation data. The failure to provide convincing evidence for negative selection in tumors has led to suggestions that it has no role in cancer evolution.

Bányai et al. used a novel approach to address the question of whether negative selection occurs in cancer. Based on characteristic differences in the patterns of mutations in cancer they distinguished clusters of passenger genes, driver genes and tumor essential genes. The group of tumor essential genes includes genes that serve to satisfy the increased demand of rapidly dividing tumor cells for nutrients' and genes that are essential for cell migration and metastasis (the spread of cancer cells to other areas of the body).

The tumor essential genes that Bányai et al. identified may prove to be valuable targets for cancer therapy, illustrating the importance of genome sequencing in cancer research. Identification of additional tumor essential genes is, however, hindered by the fact that they are likely to have low levels of mutations, which can exclude them from meaningful analyses. Progress with genomic sequencing of tumors is expected to overcome this limitation and help identify additional genes that are essential for cancer growth.

untranslated regions. Intron or splice site mutations that alter the splicing pattern of cancer genes can also drive carcinogenesis (*Diederichs et al., 2016*). A recent study has presented a comprehensive analysis of driver point mutations in non-coding regions across 2658 cancer genomes (*Rheinbay et al., 2020*). A noteworthy example of how subtle mutations in regulatory regions may activate proto-oncogenes is the telomerase reverse transcriptase gene *TERT* that encodes the catalytic subunit of telomerase. Recurrent somatic mutations in melanoma and other cancers in the *TERT* promoter cause tumor-specific increase of *TERT* expression, resulting in the immortalization of the tumor cell (*Heidenreich et al., 2014*).

In addition to subtle mutations, tumors also accumulate major chromosomal changes (*Li et al., 2020*). Most solid tumors display widespread changes in chromosome number, as well as chromosomal deletions and translocations (*Lengauer et al., 1998*). Homozygous deletions of a few genes frequently drive carcinogenesis and the target gene involved in such deletions is always a TSG (*Cheng et al., 2017*). Somatic copy-number alterations, amplifications of cancer genes are also widespread in various types of cancers. Amplifications usually contain an oncogene (OG) whose protein product is abnormally active simply because the tumor cell contains 10–100 copies of the gene per cell, compared with the two copies present in normal cells (*Beroukhim et al., 2010*; *Verhaak et al., 2019*). Chromosomal translocations may also convert wild-type forms of TSGs into forms that drive

carcinogenesis if the translocation inactivates the genes by truncation or by separating them from their promoter. Similarly, translocations may activate proto-oncogenes by changing their regulatory properties (*Haller et al., 2019*).

Epigenetic mechanisms such as DNA methylation and histone modifications may also alter the activity of cancer genes. It is now widely accepted that genetic and epigenetic changes go hand in hand in carcinogenesis: numerous genes involved in shaping the epigenome are mutated in common human cancers, and epigenetic changes affect many genes carrying driver mutations (*Yang and Yu, 2013*; *Chen et al., 2017b*; *Di Domenico et al., 2017*; *Roussel and Stripay, 2018*; *Chatterjee et al., 2018*). For example, promoter hypermethylation events may promote carcinogenesis if they lead to silencing of TSGs; the tumor-driving role of promoter methylation is obvious in the case of TSGs that are frequently inactivated by mutations in cancer (*Pfeifer, 2018*). Conversely, there is now ample evidence that promoter hypomethylation can promote carcinogenesis if it leads to increased expression of proto-oncogenes (*Van Tongelen et al., 2017*).

Non-coding RNAs (ncRNAs) also play key roles in carcinogenesis (*Slack and Chinnaiyan, 2019*). An explosion of studies has shown that – based on complementary base pairing – ncRNAs may function as OGs (by inhibiting the activity of TSGs), or as tumor suppressors (by inhibiting the activity of OGs or tumor essential genes [TEGs]).

Alterations in the splicing of primary transcripts of protein-coding genes also contribute to carcinogenesis. Recent studies on cancer genomes have revealed that recurrent somatic mutations of genes encoding RNA splicing factors (e.g. *SF3B1*, *U2AF1*, *SRSF2*, *ZRSR2*) lead to altered splice site preferences, resulting in cancer-specific mis-splicing of genes. In the case of proto-oncogenes, changes in the splicing pattern may generate active oncoproteins, whereas abnormal splicing of TSGs is likely to generate inactive forms of the tumor suppressor protein (*Dvinge et al., 2016*).

There is now convincing evidence that dysregulation of processes responsible for proteostasis also contributes to the development and progression of numerous cancer types (*Mofers et al., 2017*; *Chen et al., 2017c*; *Voutsadakis, 2017*). Recent studies on tumor tissues have revealed that genetic alterations and abnormal expression of various components of the protein homeostasis pathways (e.g. *FBXW7*, *VHL*) contribute to progression of human cancers by excessive degradation of tumor-suppressor molecules or through impaired disposal of oncogenic proteins (*Ge et al., 2018*; *Bernassola et al., 2019*).

## Hallmarks of cancer and the function of genes involved in carcinogenesis

Hanahan and Weinberg have defined a set of hallmarks of cancer that allow the categorization of cancer genes with respect to their role in carcinogenesis (*Hanahan and Weinberg, 2011*). These hallmarks describe the biological capabilities usually acquired during the evolution of tumor cells: these include sustained proliferative signaling, evasion of growth suppressors, evasion of cell death, acquisition of replicative immortality, acquisition of capability to induce angiogenesis and activation of invasion and metastasis. Underlying all these hallmarks are defects in genome maintenance that help the acquisition of the above capabilities. Additional emerging hallmarks of potential generality have been suggested to include tumor promoting inflammation, evasion of immune destruction and reprogramming of energy metabolism in order to most effectively support neoplastic proliferation (*Hanahan and Weinberg, 2011*).

*Figure 1* summarizes our current view of the cellular processes that play key roles in tumor evolution to emphasize their contribution to the various major hallmarks of cancer. Changes in the maintenance of the genome, epigenome, transcriptome, and proteome occupy a central position because they increase the chance that various constituents of other cellular pathways will experience alterations that favor the acquisition of capabilities that permit the proliferation, survival, and metastasis of tumor cells.

## Chronology of tumor evolution: initiation and progression

In the first phase of carcinogenesis, a cell may acquire a mutation that permits it to proliferate abnormally, and in the next phase, other mutations allow the expansion of cell number and this process of mutations (and associated epigenetic, transcriptomic and proteomic alterations) continues, thus generating a primary tumor that can eventually metastasize to distant organs. Recent studies on the

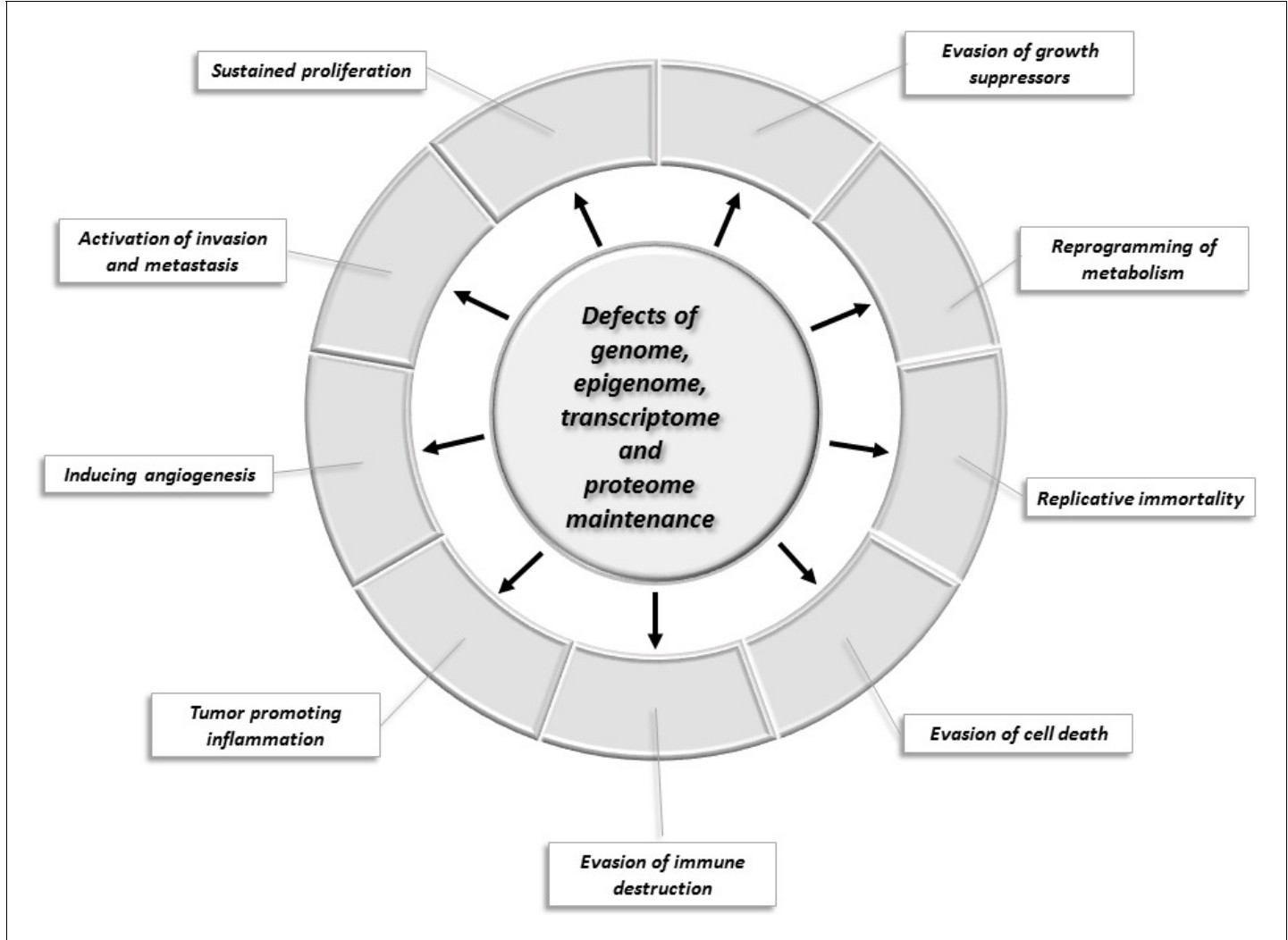

**Figure 1.** Changes of key cellular processes contributing to carcinogenesis. The central circle refers to processes involved in the maintenance of the integrity of the genome, epigenome, transcriptome, and proteome: defects in these processes increase the chance that genes and proteins of other cellular pathways (represented by segments of the outer circle) will suffer alterations that favor the acquisition of capabilities that permit the proliferation, survival, and metastasis of tumor cells.

chronology and genomic landscape of the events that drive carcinogenesis suggest that complex structural changes of the genome occur early, whereas point mutations occur in later disease phases (*Maura et al., 2019*; *Voronina et al., 2020*).

According to current estimates, the number of cancer driving mutations needed for the full development of cancer ranges from two-eight depending on cancer type (*Vogelstein and Kinzler, 2015*; *Anandakrishnan et al., 2019*). A recent integrative analysis of 2658 whole-cancer genomes and their matching normal tissues across 38 tumor types revealed that, on average, cancer genomes contain four to five driver mutations (*Campbell et al., 2020*).

Although the temporal order of the mutations affecting genes of key pathways differs among cancer types, it appears that a common feature is that mutations of genes that regulate apoptosis occur in the early phases of tumor progression, whereas mutations of genes involved in invasion pathways occur only in the last stages of carcinogenesis (*Gerstung et al., 2011*). It has been suggested that the reason why the loss of apoptotic control is a critical step for initiating cancer is that the larger the surviving cell population, the higher the number of cells at risk of acquiring additional mutations.

Analyses of the mutation landscapes and evolutionary trajectories of various tumor tissues have identified *BRAF, KRAS, TP53, RB*, or *APC* as the key genes whose mutation is most likely to initiate carcinogenesis, permitting the cell to divide abnormally (*Vogelstein and Kinzler, 2015*). In the case of ovarian cancers, *TP53* mutation is believed to be the earliest tumorigenic driver event, with presence in nearly all cases of ovarian cancer (*Bashashati et al., 2013*). The prevalence of *TP53* mutations and *BRCA* deficiency in these tumors leads to incompetent DNA repair promoting subsequent steps of carcinogenesis. Studies on the evolution of melanoma from precursor lesions have revealed that the vast majority of melanomas harbor *TERT* promoter mutations, indicating that these immortalizing mutations are selected at an unexpectedly early stage of neoplastic progression (*Shain et al., 2015*).

The life history and evolution of mutational processes and driver mutation sequences of 38 types of cancer has been analyzed recently by whole-genome sequencing analysis of 2658 cancers. This study has shown that early oncogenesis is characterized by mutations in a constrained set of driver genes and that the driver mutations that most commonly occur in a given cancer also tend to occur the earliest (*Gerstung et al., 2020*).

## Cancer genes and passenger genes

The prominent role of *KRAS* and *TP53* genes in initiating carcinogenesis has been evident from the observation that their mutation rate in tumors far exceeds those of other genes, suggesting that their mutations are subject to positive selection during tumor evolution.

Several types of approaches exploit this principle for the identification of genes that drive carcinogenesis: the rate of mutation of 'driver genes' must be significantly higher in the tumor tissue than those of 'passenger genes' (PGs) that have no role in the development of cancer but simply happen to mutate in the same tumor (*Parmigiani et al., 2009*; *Meyerson et al., 2010*).

Unfortunately, methods based on mutation frequency alone cannot reliably indicate which genes are cancer drivers because the background mutation rates differ significantly as a consequence of intrinsic characteristics of DNA sequence and chromatin structure (*Michaelson et al., 2012*). Intrinsic mutation hotspots are mutation hotspots that depend on the nucleotide sequence context, the mechanism of mutagenesis and the action of the repair and replication machineries (*Rogozin and Pavlov, 2003*). Genes enriched in intrinsic mutation hotspots may accumulate mutations at a significantly higher rate than other genes, creating the illusion of positive selection; based on recurrent mutations they may be mistakenly identified as cancer driver genes (*Carter, 2019*; *Buisson et al., 2019*).

In principle, we can avoid this danger if we compare the mutation pattern of the gene in the tumor tissue with that in the normal tissue the tumor has originated from. However, since the rate of mutation in such hotspots depends not only on the nucleotide sequence but also on the mechanism of mutagenesis and the integrity of DNA repair pathways (*Buisson et al., 2019*; *Poulos et al., 2018*) mutation hotspots that arise during carcinogenesis could still create the illusion of positive selection.

Chromatin organization also has a major influence on regional mutation rates in human cancer cells (*Schuster-Böckler and Lehner, 2012*; *Gonzalez-Perez et al., 2019*). Since large-scale chromatin features, such as replication time and accessibility influence the rate of mutations, this may hinder the distinction of cancer driver genes whose high mutation rate reflects positive selection and PGs whose high mutation rate is the result of the distinctive features of the chromatin region in which they reside. Moreover, since the cell-of-origin chromatin organization shapes the mutational landscape, rates of somatic mutagenesis of genes in cancer are highly cell-type-specific (*Polak et al., 2015*). Actually, since regional mutation density of 'passenger' mutations across the human chromosomes correlates with the cell type the tumor had originated from, this feature may be used to classify human tumors (*Salvadores et al., 2019*).

Through the comparison of the exome sequences of 3083 tumor-normal pairs *Lawrence et al., 2013* have discovered an extraordinary variation in mutation frequency and spectrum within cancer types across the genome, which is strongly correlated with DNA replication timing and transcriptional activity. The authors have shown that by incorporating mutational heterogeneity into their analyses, they could eliminate many of the apparent artefactual findings, improving the identification of genes truly associated with cancer. In a more recent study *Lawrence et al., 2014* compared the frequency of somatic point mutations in exome sequences from 4742 human cancers and their matched normal-tissue samples across 21 cancer types and identified 33 genes that were not

previously known to be significantly mutated in cancer. They have concluded that 224 genes are significantly mutated in one or more tumor types.

However, since background mutational frequency estimates are not sensitive enough, the list of driver genes (defined as genes with increased somatic mutation rate) is likely to be incomplete, but may also contain false positives. To overcome these limitations of mutation rate-based approaches, several methods use additional features that may distinguish driver genes and PGs. A major group of such approaches incorporates observations about the impact of mutations on the structure and function of well-characterized proteins encoded by proto-oncogenes and TSGs. Several computational methods aim to identify driver missense mutations most likely to generate functional changes that causally contribute to tumorigenesis (*Kaminker et al., 2007*; *Carter et al., 2009*; *Nussinov et al., 2019*).

In a different type of approach *Youn and Simon, 2011* identified cancer driver genes as those for which the non-silent mutation rate is significantly greater than a background mutation rate estimated from silent mutations, indicating that the non-silent mutations are subject to positive selection. The authors have identified 28 genes as driver genes, the majority of the significant matches (e.g. *EGFR, CDKN2A, KRAS, STK11, TP53, NF1, RB1 PTEN*, and *NRAS*), were well-characterized OGs or TSGs known from earlier studies.

In a more recent study, *Zhou et al., 2017* have identified 365 genes for which the ratio of the nonsynonymous to synonymous substitution rate was significantly increased, suggesting that they are subject to the positive selection of driver mutations. However, an obvious limitation of such approaches is that they implicitly assume that synonymous substitutions are selectively neutral and therefore the ratio of the nonsynonymous to synonymous substitution rate properly monitors selection. This is not necessarily true: some synonymous mutations may have a significant impact on splicing, RNA stability, RNA folding and translation of the transcript of the affected gene and may thus actually act as driver mutations (*Supek et al., 2014*; *Hurst and Batada, 2017*; *Sharma et al., 2019*). Furthermore, some mutation hotspots may significantly increase the rate of synonymous mutations therefore a low ratio of nonsynonymous to synonymous substitution rate does not necessarily indicate the absence of positive selection or the action of purifying selection.

*Vogelstein et al., 2013* have used a heuristic approach to identify cancer driver genes. Since the patterns of mutations in the first and best-characterized OGs and TSGs were found to be highly characteristic and nonrandom, the authors assumed that the same characteristics are generally valid and may be used to identify previously uncharacterized cancer genes. For example, since many known OGs were found to be recurrently mutated at the same amino acid positions, to classify a gene as an OG, it was required that >20% of the recorded mutations in the gene are at recurrent positions and are missense. Similarly, since in the case of known tumor suppressors the driver mutations most frequently truncate the tumor suppressor proteins, to be classified as a TSG, it was required that >20% of the recorded mutations in the gene are truncating (nonsense or frameshift) mutations. Along these lines, *Vogelstein et al., 2013* have analyzed the patterns of the subtle mutations in the Catalogue of Somatic Mutations in Cancer (COSMIC) database to identify driver genes. As a proof of the reliability of this '20/20 rule', the authors emphasized that all well-documented cancer genes passed these criteria (*Vogelstein et al., 2013*). Although this indicates that the approach detects known cancer genes, it does not guarantee that it detects all driver genes. Acknowledging that additional cancer driver genes might exist, the authors have introduced the term 'Mut-driver gene' for genes that contain a sufficient number or type of driver gene mutations to distinguish them from other genes, whereas for cancer genes that are expressed aberrantly in tumors but not frequently mutated they proposed the term 'Epi-driver gene'.

Based on these analyses, the authors have concluded that out of the 20,000 human protein-coding genes, only 125 genes qualify as Mut-driver genes, of these, 71 are TSGs and 54 are OGs (*Vogelstein et al., 2013*). The authors have expressed their conviction that nearly all genes mutated at significant frequencies had already been identified and that the number of Mut-driver genes is nearing saturation. This conclusion may not be justified since the criteria used to identify OGs and tumor suppressors appear to be too stringent and somewhat arbitrary.

In search of additional driver genes, *Tamborero et al., 2013* employed five complementary methods to find genes showing signals of positive selection and identified a list of 291 'high-confidence cancer driver genes' acting on 3205 tumors from 12 different cancer types. *Bailey et al., 2018* used multiple advanced algorithms to identify cancer driver genes and driver mutations. Based

on their PanCancer and PanSoftware analysis spanning 9423 tumor exomes, comprising all 33 of The Cancer Genome Atlas projects and using 26 computational tools they have identified 299 driver genes showing signs of positive selection. Their sequence and structure-based analyses detected >3400,400 putative missense driver mutations and 60–85% of the predicted mutations were validated experimentally as likely drivers.

*Zhao et al., 2019a* have developed driverMAPS (Model-based Analysis of Positive Selection), a model-based approach for driver gene identification that captures elevated mutation rates in functionally important sites and spatial clustering of mutations. The authors have identified 255 known driver genes as well as 170 putatively novel driver genes.

Currently, COSMIC (the Catalogue Of Somatic Mutations In Cancer, https://cancer.sanger.ac.uk/cosmic) is the most detailed and comprehensive resource for exploring the effect of subtle somatic mutations of driver genes in human cancer (*Tate et al., 2019*) but COSMIC also covers all the genetic mechanisms by which somatic mutations promote cancer, including non-coding mutations, gene fusions, and copy-number variants. In parallel with COSMIC's variant coverage, the Cancer Gene Census (CGC, https://cancer.sanger.ac.uk/census) describes a curated catalogue of genes driving every form of human cancer (*Sondka et al., 2018*). CGC has recently introduced functional descriptions of how each gene drives disease, summarized into the cancer hallmarks. CGC describes in detail the effect of a total of 719 cancer-driving genes, encompassing Tier 1 genes (574 genes) and a list of Tier 2 genes (145 genes) from more recent cancer studies that show less detailed indications of a role in cancer.

In a different type of approach, *Torrente et al., 2016* used comprehensive maps of human gene expression in normal and tumor tissues to identify cancer related genes. These analyses identified a list of genes with systematic expression change in cancer. The authors have noted that the list is significantly enriched with known cancer genes from large, public, peer-reviewed databases, whereas the remaining ones were proposed as new cancer gene candidates. A recent study has provided a comprehensive catalogue of cancer-associated transcriptomic alterations with the top-ranking genes carrying both RNA and DNA alterations. The authors have noted that this catalogue is enriched for cancer census genes (*Calabrese et al., 2020*).

Using transposon mutagenesis in mice, several laboratories have conducted forward genetic screens and identified thousands of candidate genetic drivers of cancer that are highly relevant to human cancer. The Candidate Cancer Gene Database (CCGD, http://ccgd-starrlab.oit.umn.edu/) is a manually curated database containing a unified description of all identified candidate driver genes (*Abbott et al., 2015*).

In summary, although a variety of approaches have been developed to identify 'cancer genes', there is significant disagreement as to the number of genes involved in carcinogenesis. Some of the studies argue that the number is in the 200–700 range, other approaches suggest that their number may be much higher. Since the ultimate goal of cancer genome projects is to discover therapeutic targets, it is important to identify all true cancer genes and distinguish them from PGs and candidates that do not play a significant role in the process of carcinogenesis.

We must point out, however, that the majority of genomics-based methods were biased as they defined the aim of cancer genomics as the identification of mutated driver genes (equating them with 'cancer genes') that are causally implicated in oncogenesis (*Futreal et al., 2004*). In all these studies, the underlying rationale for interpreting a mutated gene as causal in cancer development is that the mutations are likely to have been positively selected because they confer a growth advantage on the cell population from which the cancer has developed. An inevitable consequence of this focus on positive selection was that most studies neglected the possibility that negative selection may also play a significant role in tumor evolution.

## Carcinogenesis as an evolutionary process

In principle, with respect to its effect on carcinogenesis, a somatic mutation may promote or may hinder carcinogenesis or may have no effect on carcinogenesis. In cancer genomics, the mutations that promote carcinogenesis (and are subject to positive selection during tumor evolution) are called 'driver mutations' to distinguish them from 'passenger mutations' that do not play a role in carcinogenesis (and are not subject to positive or negative selection during tumor evolution). Mutations that impair the growth, survival, and invasion of tumor cells have received much less attention, although they could also play a significant role in shaping the mutation pattern of genes during

carcinogenesis. Hereafter, we will refer to this category of mutations as 'cancer blocking mutations' because they are deleterious from the perspective of tumor growth.

As discussed above, in cancer genomics, genes are usually assigned to just two categories with respect to their role in carcinogenesis: (1) 'PGs' (or bystander genes) that play no significant role in carcinogenesis and their mutations are passenger mutations; (2) 'driver genes' that drive carcinogesis when they acquire driver mutations.

The problem with this binary driver gene-PG categorization is that some genes with functions essential for the growth and survival of tumor cells (hereafter referred to as 'tumor essential genes') may not easily fit into either category. The coding sequences of driver genes (TSGs, proto-oncogenes), PGs, and TEGs are predicted to experience markedly different patterns of selection during tumor evolution.

The mutation patterns of selectively neutral, bona fide PGs are likely to reflect the lack of positive and negative selection, whereas in the case of TEGs purifying selection is predicted to dominate. In the case of TSGs, the mutation pattern is expected to reflect positive selection for inactivating driver mutations. Proto-oncogenes, however, are expected to show signs of both positive selection for activating mutations and negative selection for inactivating, 'cancer blocking' mutations as their activity is essential for their oncogenic role. In the coding regions of proto-oncogenes positive selection for driver mutations is expected to favor nonsynonymous substitutions over synonymous substitutions only at sites that are critical for the novel, oncogenic function. For these sites (and these sites only), the ratio of nonsynonymous to synonymous rates is expected to be significantly greater than one reflecting positive selection. If there are many such sites in a protein, or selection is extremely strong the overall nonsynonymous to synonymous ratio for the entire protein may also be significantly higher than one, otherwise the effect of positive selection on the synonymous to nonsynonymous ratio may be overridden by purifying selection at other sites (*Patthy, 1999*).

In harmony with some of these expectations, using just the ratio of the nonsynonymous to synonymous substitution rate as a measure of positive or negative selection, *Zhou et al., 2017* have shown that in cancer genomes, the majority of genes had nonsynonymous to synonymous substitution rate values close to one, suggesting that they belong to the PG category. The authors have identified a total of 365 potential cancer driver genes that had nonsynonymous to synonymous substitution rate values significantly greater than one (reflecting the dominance of positive selection). Conversely, 923 genes had nonsynonymous to synonymous substitution rate values significantly less than one (reflecting the dominance of negative selection), leading the authors to suggest that these negatively selected genes may be important for the growth and survival of cancer cells.

*Pyatnitskiy et al., 2015* have also used the dN/dS ratio (the ratio of nonsynonymous and synonymous substitution rates) as an indicator of selective pressure and have identified 91 protein-coding genes ('essential cancer proteins') with amino acid sequences under negative selection.

Realizing that genes whose wild-type coding sequences are needed for tumor growth are also of key interest for cancer research, *Weghorn and Sunyaev, 2017* have also focused on the role of negative selection in human cancers. The authors have used an approach based on the principle that both positive and negative selection can be inferred by comparing the observed mutation rates to the expectation under the sole action of the mutation process. As the authors have pointed out, identification, and analysis of true negatively selected,' undermutated' genes is particularly difficult since the sparsity of mutation data results in lower statistical power, making conclusions less reliable. Although the signal of negative selection was exceedingly weak, the authors have noted that the group of negatively selected candidate genes is enriched in cell-essential genes identified in a CRISPR screen (*Wang et al., 2015a*), consistent with the notion that one of the potential causes of negative selection is the maintenance of genes that are responsible for basal cellular functions. Based on pergene estimates of negative selection inferred from the pan-cancer analysis the authors have identified 147 genes with significant negative selection. The authors have noted that among the 13 genes showing the strongest signs of negative selection there are several genes (*ATAT1, BCL2, CLIP1, GALNT6, CKAP5,* and *REV1*) that are known to promote carcinogenesis.

In a similar work, *Martincorena et al., 2017* have used the normalized ratio of non-synonymous to synonymous mutations, to quantify selection in coding sequences of cancer genomes. Using a nonsynonymous-to-synonymous substitution rate value >1 as a marker of cancer genes under positive selection, they have identified 179 cancer genes, with about 50% of the coding driver mutations being found to occur in novel cancer genes. The authors, however, have concluded that purifying

selection is practically absent in tumors since nearly all (>99%) coding mutations are tolerated and escape negative selection. The authors have suggested that this remarkable absence of negative selection on coding point mutations in cancer indicates that the vast majority of genes are dispensable for any given somatic lineage, presumably reflecting the buffering effect of diploidy and the inherent resilience and redundancy built into most cellular pathways.

The key message of *Martincorena et al., 2017* that negative selection has no role in cancer evolution had a major impact on cancer genomics research as reflected by several commentaries in major journals of the field that have propagated this conclusion (*Bakhoum and Landau, 2017*; *Koch, 2017*; *Vitale and Galluzzi, 2018*).

Some more recent studies, however, contradict this conclusion. Although *Zapata et al., 2018* have also used the ratio of nonsynonymous-to-synonymous substitutions to identify genes that are under selection, they have detected significant negative selection in the case of 25 genes. *López et al., 2020*, focusing on dN/dS values for truncating mutations, have shown that purifying selection of essential genes is significant in early phases of tumor evolution (before whole genome duplications), whereas whole-genome doubling allows the accumulation of deleterious alterations. *Tilk et al., 2020* have shown that appreciable negative selection (dN/dS ~ 0.4) is present in tumors with a low mutational burden, while the majority of tumors exhibit dN/dS ratios approaching 1, suggesting that tumors with higher mutational burden do not remove deleterious mutations.

*Van den Eynden and Larsson, 2017*, however, cautioned that it is crucial to take into account mutational signatures when applying the dN/dS metric to cancer somatic mutation data. For example, the authors have shown that the low dN/dS values observed in malignant melanoma may be due to the predominance of C to T mutations in this tumor and do not necessarily indicate gene essentiality. The authors have also shown that purifying selection is very limited and similar in all tumor types if the dN/dS metric uses mutational signature-derived substitution probabilities.

In view of the contradicting conclusions about the significance of negative selection in tumor evolution, in the present work we have reexamined this question using an approach that attempts to overcome some of the problems highlighted by earlier studies.

First, most studies used a single dN/dS metric to measure nonsynonymous to synonymous substitution rates as indicators of selective pressure and paid less attention to the fact that the strength of purifying selection is an order of magnitude greater for nonsense mutations than for missense mutations (*Gorlov et al., 2006*). Furthermore, the use of a single dN/dS value for a transcript may preclude the simultaneous detection of positive and negative selection of activating and inactivating mutations, both of which might operate for a given gene. To overcome these limitations, in the present study we have used a clustering-based approach that can detect different signals of selection manifested in rates of nonsense, missense *versus* silent substitutions in the coding regions of genes.

Second, an inherent problem with the detection of purifying selection in tumor tissues is that putative TEGs are likely to be undermutated relative to PGs and driver genes, resulting in low statistical power of their analyses based on dN/dS metrics. We have reduced this problem by combining subtle somatic mutations from different tumors types and limiting our work to transcripts that have at least 100 somatic mutations in tumors. (Note that the requirement of a minimum number of mutations does not place a theoretical limit on this approach; progress with genome-wide screens and collection of more data is overcoming this limitation.)

In harmony with earlier observations, our analyses have confirmed that the vast majority of human genes are PGs that do not show detectable signals of selection, whereas known TSGs are positively selected for inactivating (primarily nonsense and frame-shift) mutations. Known OGs, however, were found to display signals of both negative selection for inactivating (nonsense, frame-shift) mutations and positive selection for activating (missense) mutations. Improved detection of signals of selection has permitted the identification of a number of novel driver genes that are likely to play important roles in carcinogenesis as TSGs or as OGs.

Significantly, we have identified a cluster of human genes that show clear signs of negative selection during tumor evolution, suggesting that their functional integrity is essential for the growth and survival of tumor cells. The group of negatively selected genes includes genes known to play critical roles in the Warburg effect of cancer cells, others are known to mediate invasion and metastasis of tumor cells, indicating that negatively selected TEGs may prove a rich source for novel targets for tumor therapy.

## Results

### Distinguishing PGs and cancer genes

The rationale of the analyses described in the present work is that — due to their different roles in carcinogenesis — proto-oncogenes, TSGs, TEGs, and PGs are expected to experience different patterns of selection during tumor evolution and this is reflected in the relative rates of missense, nonsense, and silent mutations of their protein-coding regions. To monitor these differences, we have calculated for each transcript the fraction of somatic substitutions that could be assigned to the silent (fS), missssense (fM), and nonsense (fN) category and analyzed their relative rates. (For details, the reader should consult the Materials and methods section).

Our analyses have shown that in 3D scatter plots of the fS, fM, and fN values of transcripts the majority of genes are present in a central cluster characterized by fS, fM, and fN values close to those expected assuming no mutation bias and absence of selection, consistent with the view that they correspond to PGs (*Figure 2*). Known OGs, however, were found in a separate cluster characterized by higher fM values, reflecting positive selection for missense mutations, whereas the cluster of known TSGs has higher fN values, reflecting positive selection for truncating nonsense mutations (*Figure 2B and C*).

Known cancer genes also separate from the majority of human genes in 3D scatter plots of rSM, rNM, rNS parameters, defined as the ratio of fS/fM, fN/fM, fN/fS, respectively (*Figure 3*). In these scatter plots, OGs separate from the central cluster in having lower rSM and rNM values, whereas TSGs have higher rNS and rNM values than those of the central cluster (*Figure 3*).

The separation of known cancer genes from the majority of human genes is even more manifest in 3D scatter plots of parameters rSMN, rMSN, and rNSM defined as the ratio of fS/(fM+fN), fM/(fS+fN), and fN/(fS+fM), respectively (*Figure 4*). In these plots, the transcripts form a three-pronged cluster, with known OGs and TSGs being present on separate spikes of this cluster, the rMSN and rNSM spikes, respectively (*Figure 4*).

There is, however, a fourth cluster of genes that deviates from the clusters of PGs, OGs, and TSGs (*Figures 2*, *3* and *4*). The high fS, rSM, and rSMN values of the transcripts in this group

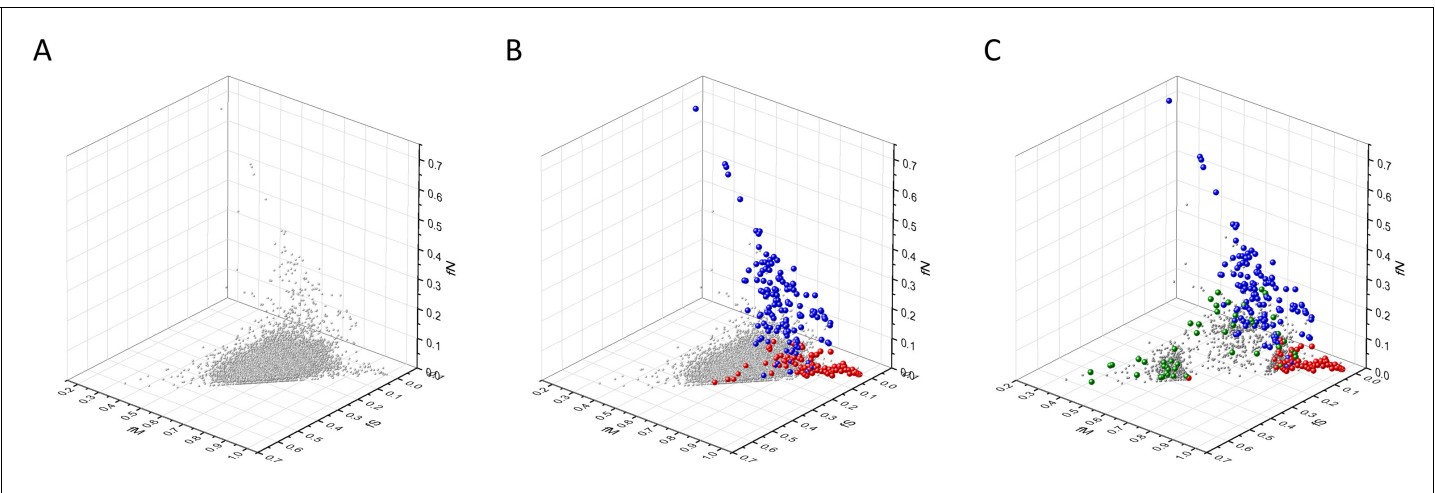

**Figure 2.** Analyses of fS, fM, and fN parameters of human protein-coding genes of tumor tissues. The figure shows the results of the analysis of 13,803 transcripts containing at least 100 subtle, confirmed somatic mutations from tumor tissues, including only mutations identified as not single-nucleotide polymorphisms (SNPs). Axes *x, y,* and *z* represent the fractions of somatic single-nucleotide substitutions that are assigned to the synonymous (fS), nonsynonymous (fM), and nonsense (fN) categories, respectively. In Panel A, each gray ball represents a human transcript; note that the majority of human genes are present in a dense cluster. Panel B highlights the positions of transcripts of the genes identified by *Vogelstein et al., 2013* as oncogenes (OGs, large red balls) or tumor suppressor genes (TSGs, large blue balls). It is noteworthy that these driver genes separate significantly from the central cluster and from each other: OGs have a significantly larger fraction of nonsynonymous, whereas TSGs have significantly larger fraction of nonsense substitutions. Panel C shows data only for candidate cancer genes present in the CG_SO$^{2SD}$_SSI$^{2SD}$ list (see Materials and methods). The positions of novel cancer gene transcripts validated in the present work are highlighted as large green balls.

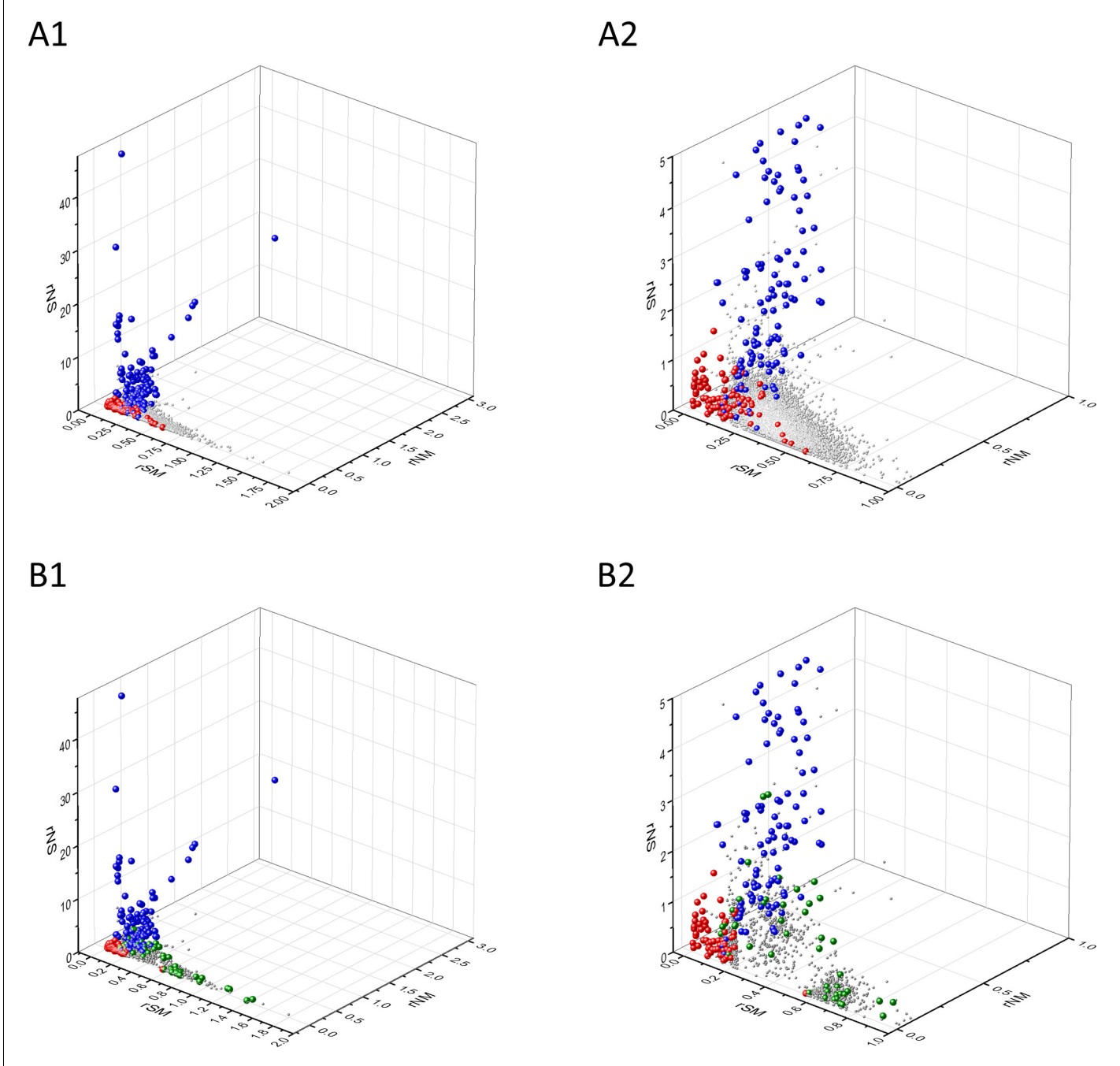

**Figure 3.** Analyses of rSM, rNM, rNS parameters of human protein-coding genes of tumor tissues. The figure shows the results of the analysis of 13,803 transcripts containing at least 100 subtle, confirmed somatic mutations from tumor tissues, including only mutations identified as not single-nucleotide polymorphisms (SNPs). Axes *x, y,* and *z* represent the rSM, rNM, rNS values defined as the ratio of fS/fM, fN/fM, fN/fS, respectively. Each ball represents a human transcript; the positions of transcripts of the genes identified by *Vogelstein et al., 2013* as oncogenes (OGs, large red balls) or tumor suppressor genes (TSGs, large blue balls) are highlighted. Panels A1, A2 show the distribution of the 13,803 transcripts at different magnification. Note that the majority of human genes are present in a dense cluster but known OGs and TSGs separate significantly from the central cluster and from each other. The rNS and rNM values of TSGs are higher, whereas the rSM and rNM values of OGs are lower than those of passenger genes. Panels B1, B2 show data only for candidate cancer genes present in the CG_SO²ˢᴰ_SSI²ˢᴰ list (see Materials and methods). The positions of novel cancer gene transcripts validated in the present work are highlighted as large green balls.

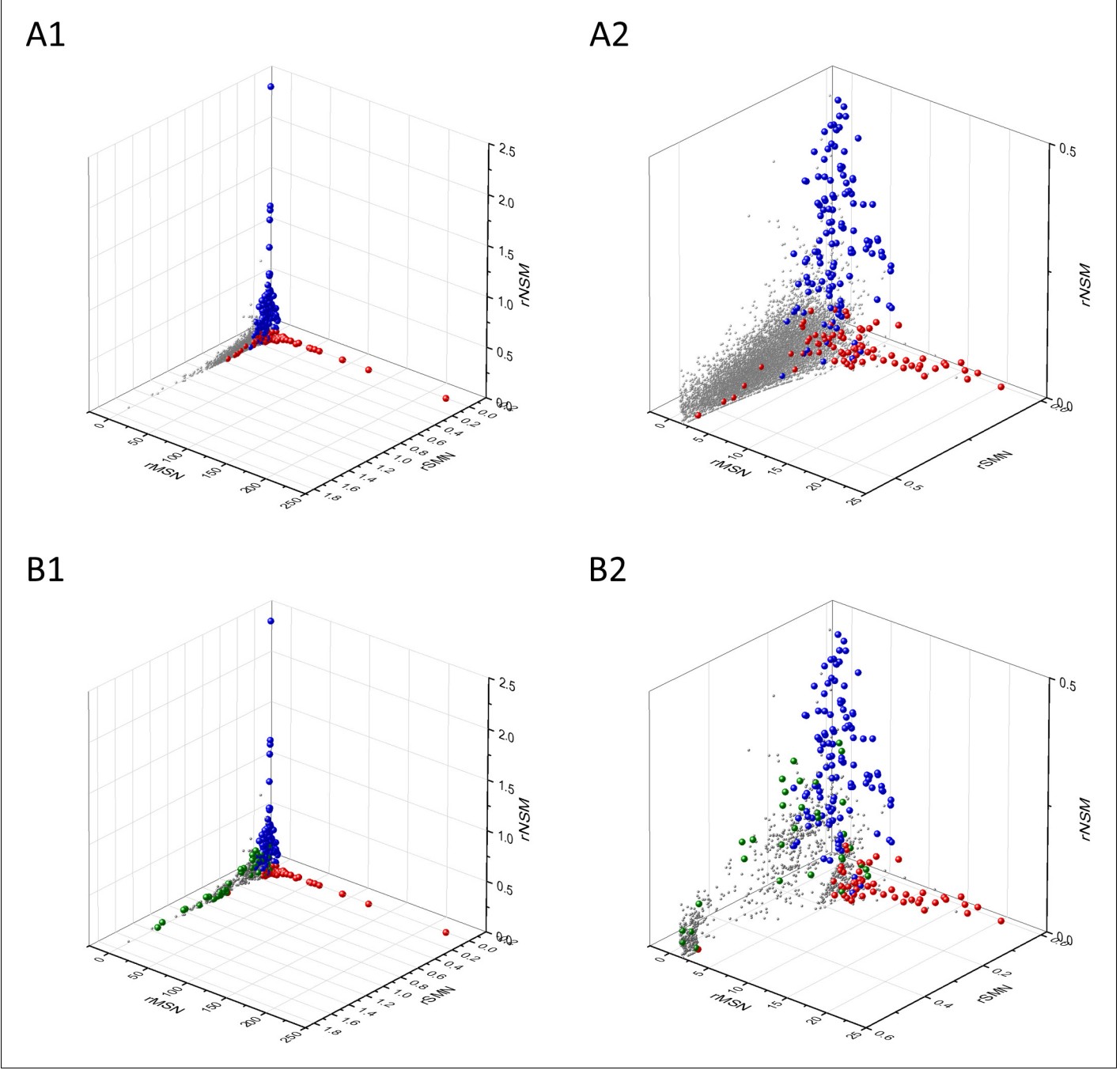

**Figure 4.** Analyses of rSMN, rMSN, and rNSM parameters of human protein-coding genes of tumor tissues. The figure shows the results of the analysis of transcripts containing at least 100 subtle, confirmed somatic mutations from tumor tissues, including only mutations identified as not single-nucleotide polymorphisms (SNPs). Axes *x, y*, and *z* represent the rSMN, rMSN, and rNSM defined as the ratio of fS/(fM+fN), fM/(fS+fN), and fN/(fS+fM). Each ball represents a human transcript; the positions of transcripts of the genes identified by *Vogelstein et al., 2013* as oncogenes (OGs, large red balls) or tumor suppressor genes (TSGs, large blue balls) are highlighted. Panels A1, A2 show the distribution of the 13,803 transcripts at different magnification. Note that the majority of human genes are present in a dense cluster but known OGs and TSGs separate significantly from the central cluster and from each other. The rNSM values of TSGs are higher, their rMSN and rSMN are lower than those of passenger genes (PGs). OGs also separate from PGs in that their rMSN values are higher and their rSMN and rNSM values are lower than those of PGs. Panels B1, B2 show data only for candidate cancer genes present in the CG_SO²ˢᴰ_SSI²ˢᴰ list (see Materials and methods). The positions of novel cancer gene transcripts validated in the present work are highlighted as large green balls.

suggest that they are subject to purifying selection during tumor evolution, raising the possibility that this group may contain genes essential for the survival of tumors.

The analyses discussed above did not take into account the impact of differences in mutation probability on the fN, fM, and fS values of transcripts. To check the influence of this factor, we have calculated the expected fN*, fM*, and fS* values for all human transcripts using the probabilities of the six substitution classes (C>A, C>G, C>T, T>A, T>C, and T>G) observed across tumors (for details the reader should consult the Materials and methods section).

The various types of observed/expected ratios (rN*, rM*, rS*; rSM*, rNM*, rNS*; rSMN*, rMSN* and rNSM*) were calculated for each transcript and the data were analyzed in 3D scatter plots as described above for the observed values. As shown in *Figures 5*, *6* and *7*, the distribution of transcripts in these 3D scatter plots are similar to those observed in the corresponding *Figures 2*, *3* and *4*, indicating that the separation of the clusters of PGs, OGs, TSGs, and TEGs is relatively insensitive to transcript-specific differences in mutation probabilities.

## Analyses of candidate cancer gene sets

We assumed that the genes whose patterns of subtle mutations deviate significantly (by more than 2SD) from those of prototypical PGs are enriched in cancer genes that play important role in carcinogenesis. The patterns of subtle mutations of candidate cancer genes assign them to one of the three main clusters that show signs of positive and/or negative selection (see *Figures 2–7*). (A) Genes positively selected for inactivating (nonsense and frame-shift) mutations – putative TSGs; (B) genes positively selected for missense mutations and negatively selected for inactivating mutations – putative proto-oncogenes; (C) negatively selected genes – putative TEGs.

The assumption that the cancer genes assigned to these three clusters play significant roles in carcinogenesis has strong support in the case of the first two categories: the approach used in the present study correctly assigned the known, 'gold standard' TSGs and OGs (*Supplementary file 1*). In the case of the third category, however, no similar gold standard exists for TEGs.

To check the validity and predictive value of the assumption that the genes assigned to the three clusters play critical roles in carcinogenesis, we have selected a number of genes at random from

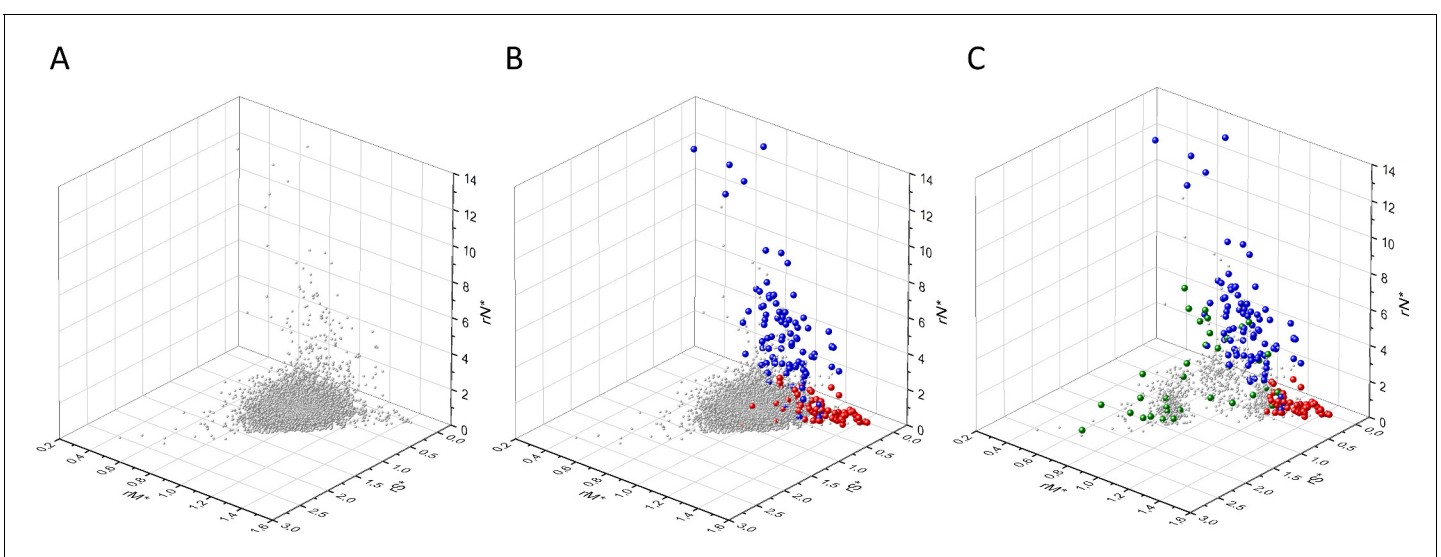

**Figure 5.** Analyses of rS*, rM*, and rN* parameters of human protein-coding genes of tumor tissues. The figure shows the results of the analysis of transcripts containing at least 100 subtle, confirmed somatic mutations from tumor tissues. Axes x, y, and z represent rS*, rM*, and rN* values, respectively. In Panel A, each gray ball represents a human transcript; note that the majority of human genes are present in a dense cluster. Panel B highlights the positions of transcripts of the genes identified by *Vogelstein et al., 2013* as oncogenes (OGs, large red balls) or tumor suppressor genes (TSGs, large blue balls). It is noteworthy that these driver genes separate significantly from the central cluster and from each other: OGs have a significantly larger fraction of nonsynonymous, whereas TSGs have significantly larger fraction of nonsense substitutions than expected. Panel C shows data only for candidate cancer genes present in the CG_SO²ˢᴰ_SSI²ˢᴰ list (see Materials and methods). The positions of novel cancer gene transcripts validated in the present work are highlighted as large green balls.

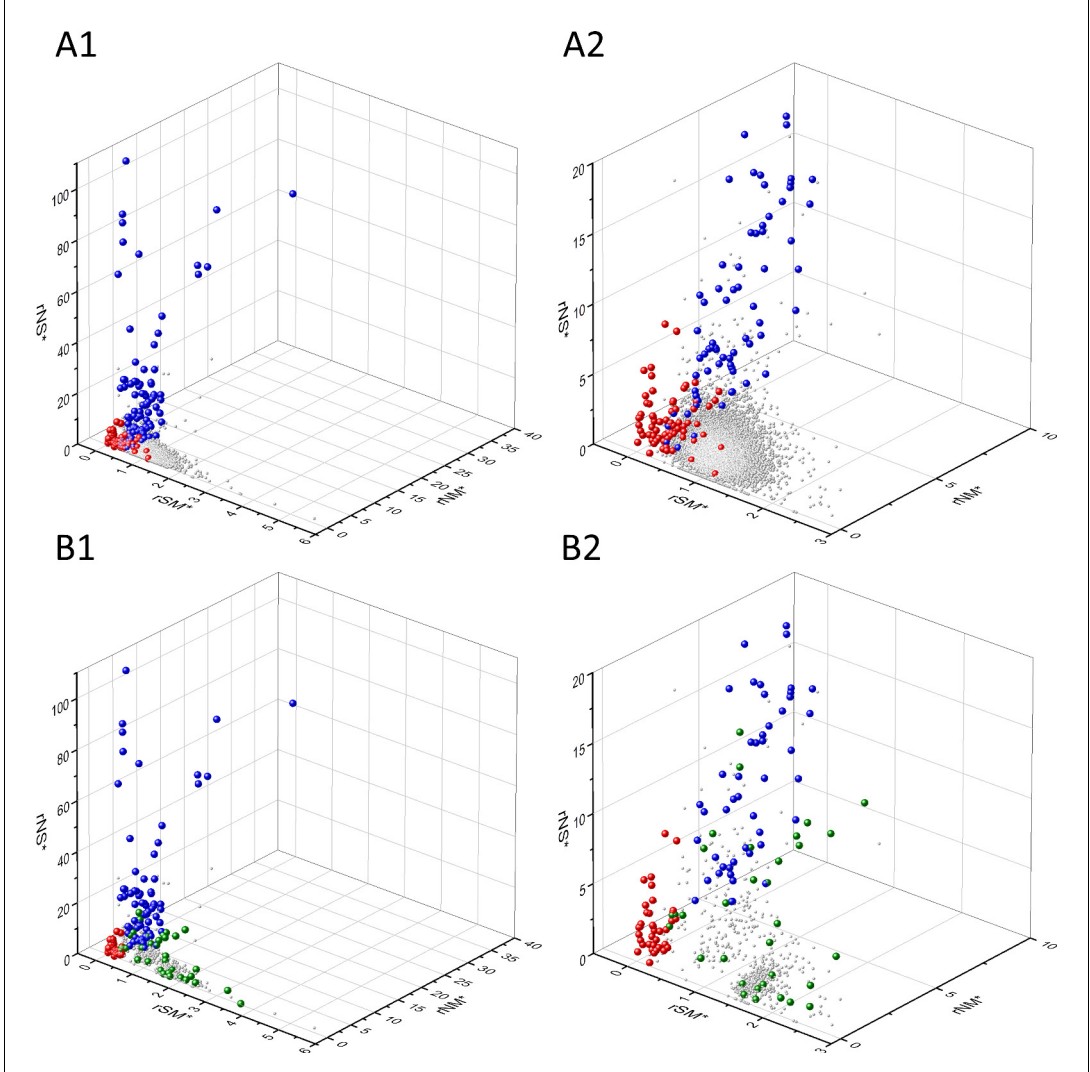

**Figure 6.** Analyses of rSM*, rNM*, rNS* parameters of human protein-coding genes of tumor tissues. The figure shows the results of the analysis of transcripts containing at least 100 subtle, confirmed somatic mutations from tumor tissues. Axes *x*, *y*, and *z* represent rSM*, rNM*, rNS* values, respectively. Each ball represents a human transcript; the positions of transcripts of the genes identified by *Vogelstein et al., 2013* as oncogenes (OGs, large red balls) or tumor suppressor genes (TSGs, large blue balls) are highlighted. Panels A1 and A2 show the distribution of the transcripts at different magnification. Note that the majority of human genes are present in a dense cluster but known OGs and TSGs separate significantly from the central cluster and from each other. The rNS* and rNM* values of TSGs are higher, whereas the rSM* and rNM* values of OGs are lower than those of passenger genes. Panels B1, B2 show data only for candidate cancer genes present in the CG_SO$^{2SD}$_SSI$^{2SD}$ list (see Materials and methods). The positions of novel cancer gene transcripts validated in the present work are highlighted as large green balls.

each cluster for further in-depth analyses. We have used three criteria to select genes for detailed analyses from the combined list of candidate cancer genes that deviate from the central clusters of PGs by more than 2SD (see Materials and methods). (1) The candidate gene is among the genes showing the strongest signals of selection characteristic of the given group. (2) The candidate gene is novel in the sense that it is not listed among the 145' gold standard' OGs and TSGs of *Vogelstein et al., 2013* or among the 719 cancer genes of CGC (*Sondka et al., 2018*). (3) There is substantial experimental information in the scientific literature on the given gene to permit the assessment of its role in carcinogenesis.

The genes discussed below include genes positively selected for truncating mutations (putative TSGs), genes positively selected for missense mutations and negatively selected for inactivating mutations (putative proto-oncogenes) and negatively selected genes (putative TEGs). In the main text, we summarize only the major conclusions of our analyses; for annotations of the individual

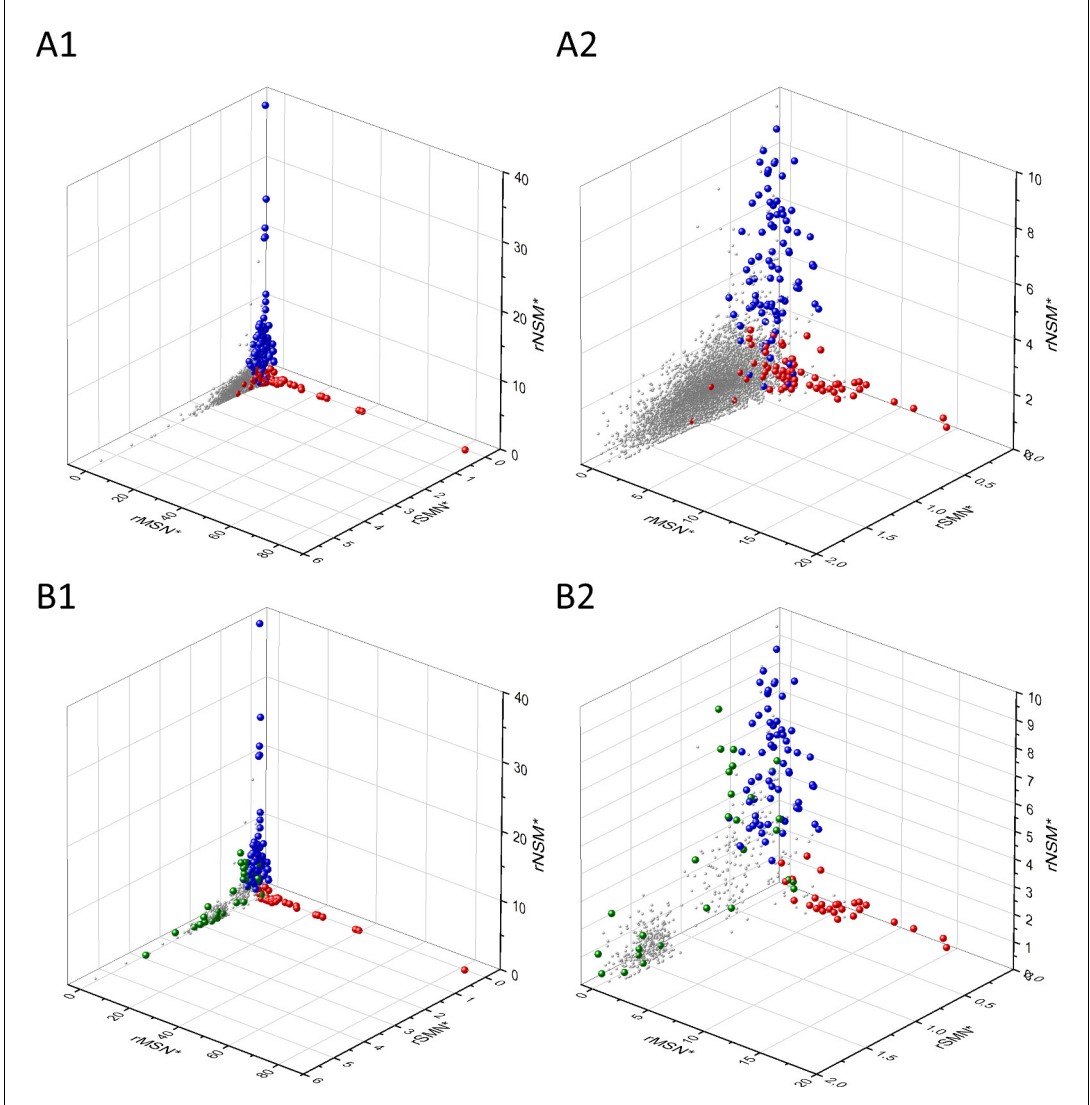

**Figure 7.** Analyses of rSMN*, rMSN*, and rNSM* parameters of human protein-coding genes of tumor tissues. The figure shows the results of the analysis of transcripts containing at least 100 subtle, confirmed somatic mutations from tumor tissues. Axes *x*, *y*, and *z* represent the rSMN*, rMSN*, and rNSM* values, respectively. Each ball represents a human transcript; the positions of transcripts of the genes identified by *Vogelstein et al., 2013* as oncogenes (OGs, large red balls) or tumor suppressor genes (TSGs, large blue balls) are highlighted. Panels A1, A2 show the distribution of the transcripts at different magnification. Note that the majority of human genes are present in a dense cluster but known OGs and TSGs separate significantly from the central cluster and from each other. The rNSM* values of TSGs are higher, their rMSN* and rSMN* are lower than those of passenger genes (PGs). OGs also separate from PGs in that their rMSN* values are higher and their rSMN* and rNSM* values are lower than those of PGs. Panels B1, B2 show data only for candidate cancer genes present in the CG_SO$^{2SD}$_SSI$^{2SD}$ list (see Materials and methods). The positions of novel cancer gene transcripts validated in the present work are highlighted as large green balls.

genes, the reader should consult Appendix 1. We discuss examples of negatively selected genes in the main text in more detail since earlier studies that focussed on positive selection of driver mutations inevitably missed these genes. We also discuss some instructive examples of' false' hits, that is cases where the mutation parameters deviate significantly from those of PGs, but this deviation is not due to selection.

## Novel cancer genes positively selected for nonsense mutations

We have selected genes positively selected for truncating mutations from the combined list of candidate transcripts, that is, transcripts whose parameters deviate from those of PGs by more than 2SD

(for details see Materials and methods). We have used the additional restriction that genes with indel_rNSM <0.125 were excluded (*Supplementary file 1*), thereby removing OGs and TEGs. Out of the 624 genes that satisfy these criteria, we have subjected *B3GALT1, BMPR2, BRD7, ING1, MGA, PRRT2, RASA1, RNF128, SLC16A1, SPRED1, TGIF1, TNRC6B, TTK, ZNF276, ZC3H13, ZFP36L2,* and *ZNF750* to further analysis.

Annotation of the majority of these genes (*BMPR2, BRD7, ING1, MGA, PRRT2, RASA1, RNF128, SLC16A1, SPRED1, TGIF1, TNRC6B, ZC3H13, ZFP36L2,* and *ZNF750*) has provided convincing evidence for their role in carcinogenesis as tumor suppressors. Interestingly, experimental evidence suggests that *TTK*, encoding dual specificity protein kinase TTK, is a proto-oncogene that may be converted to an OG by truncating mutations affecting the very C-terminal end of the protein, downstream of its kinase domain (for further details see Appendix 1). Our annotations suggest that *B3GALT1, ZNF276* are false positives whose apparent mutation pattern deviates significantly from those of PGs, but this deviation is not due to selection.

Based on functional annotation of the TSGs identified and validated in the present work (see Appendix 1), we have assigned them to various cellular processes of cancer hallmarks in which they are involved (*Table 1*).

Comparison of the list of 624 genes present in this dataset (CG_SSI$^{2SD}$ rNSM >0.125) with lists identified by others (*Supplementary file 1*) revealed that ~60–100 of our candidate TSG-like genes are also found in several gene lists identified by others through analyses of somatic mutations of tumor tissues. Many of the genes selected for annotation are present in at least one of the candidate gene lists identified by others; the genes of *MGA, RASA1, TGIF1, ZFP36L2,* and *ZNF750* are present in multiple cancer gene lists (*Supplementary file 1*). It is noteworthy, however, that *RNF128, SLC16A1, SPRED1, TNRC6B,* and *TTK* are novel in that they are found only among the candidate cancer genes identified by forward genetic screens in mice (*Abbott et al., 2015*) or among the genes whose expression changes in cancer (*Torrente et al., 2016*).

We have also analyzed the genes present in dataset CG_SO*$^{2SD}$_rNSM >3, that is, candidate cancer genes for which the observed rNSM values are more than threefold higher than expected taking into account mutational signature-derived substitution probabilities of tumors (*Supplementary file 2*). We have found that 164 (100%) of the 164 genes present in this dataset are also present in the dataset CG_SSI$^{2SD}$ rNSM >0.125. It is noteworthy that the majority of candidate TSGs selected for annotation (*B3GALT1, BMPR2, BRD7, ING1, MGA, PRRT2, RASA1, SLC16A1, SPRED1, TGIF1,*

**Table 1.** Assignment of novel positively or negatively selected cancer genes to key cellular processes of carcinogenesis.

| Hallmarks of cancer | Gene symbol |
| --- | --- |
| Defects of genome, epigenome, transcriptome, or proteome maintenance | *CDK8, FOXG1, IDH3B, MARCH7, MGA,* **NOVA1**, **PNCK**, *RNF128, TGIF1, TNRC6B,* **TWIST1**, *ZC3H13, ZFP36L1, ZFP36L2, ZNF750* |
| Sustained proliferation | *AURKA, BRD7, ING1,* **FOXG1**, **MAPK13**, **PNCK**, *PRRT2, RASA1, RIT1, SPRED1,* **TRIB2**, *TTK, YAP1, YES1, ZFP36L1, ZFP36L2, ZNF750* |
| Evasion of growth suppressors | |
| Reprogramming of metabolism | *BRD7,* **G6PD**, *SLC16A1,* **SLC16A3**, **SLC2A1**, **SLC2A8**, *YAP1, YES1* |
| Replicative immortality | **NOVA1** |
| Evasion of cell death | *BRD7, ING1,* **MAPK13**, **PNCK**, *PRRT2,* **TP73**, *TRIB2, TTK, YAP1, YES1, ZNF750* |
| Evasion of immune destruction | |
| Tumor promoting inflammation | *BMP2R,* **CCR2**, **CCR5**, **CX3CR1**, **MAPK13** |
| Inducing angiogenesis | **CCR2** |
| Activation of invasion and metastasis | **CCR2**, **CCR5**, **CX3CR1**, *RASA1,* **TBXA2R** |

For annotation of novel genes identified in the present study see Appendix 1. The names of negatively selected genes are marked by bold underline.

ZNF276, ZFP36L2, and ZNF750) are present among the genes shared by the two datasets that show the strongest signals of positive selection for nonsense substitutions.

## Novel cancer genes positively selected for missense and negatively selected for nonsense mutations

We have selected genes positively selected for missense and negatively selected for inactivating mutations from the list of candidate transcripts using the restriction that genes with rMSN <3.00 (440) were excluded, thereby removing the majority of TSGs and TEGs (*Supplementary file 1*). Out of the 440 genes that satisfy these criteria, we have subjected *AURKA, CDK8, IDH3B, MARCH7, RIT1, YAP1*, and *YES1* to further analysis.

Annotation of these genes has confirmed that they play important roles in carcinogenesis as OGs. Three of these genes encode kinases (Aurora kinase A, also known as breast tumor-amplified kinase; cyclin-dependent kinase 8; tyrosine-protein kinase Yes, also known as proto-oncogene c-Yes) but unlike many other oncogenic kinases, these OGs do not show significant clustering of missense mutations. In fact, only in the case of *IDH3B* and *RIT1* did we observe clustering of missense mutations, indicating that recurrent mutation is not an obligatory property of proto-oncogenes.

Based on functional annotation of the novel OGs identified and validated in the present work (see Appendix 1), we have assigned them to various cellular processes of cancer hallmarks in which they are involved (*Table 1*).

Comparison of this list of 440 genes (CG_SO$^{2SD}$ rMSN >3.00) with the lists of cancer genes identified by others (*Supplementary file 1*) revealed that ~60–100 of our candidate OG-like genes are present in cancer gene lists identified by others through analyses of somatic mutations of tumor tissues.

Out of the genes that we have selected for annotation only the *RIT1* gene has been identified by others as an OG, based on the analysis of somatic mutations (*Supplementary file 1*). *AURKA* and *IDH3B* are not present in any of the lists of cancer genes, whereas *CDK8, MARCH7, YAP1*, and *YES1* are found among the more than 9000 candidate cancer genes identified by forward genetic screens in mice (*Abbott et al., 2015*). Interestingly, *TTK*, identified as a gene positively selected for truncating mutations (see list CG_SSI$^{2SD}$ rNSM >0.125), but annotated as an OG, is also present in the list of genes positively selected for missense mutations (CG_SO$^{2SD}$ rMSN >3.00).

We have also analyzed the genes present in dataset CG_SO*$^{2SD}$–rMSN >1.50, that is, genes for which the observed rMSN values are more than 1.5-fold higher than expected taking into account mutational signature-derived substitution probabilities of tumors (*Supplementary file 2*). We have found that 119 (98.3%) of the 121 genes present in this dataset are also present in the dataset CG_SO$^{2SD}$ rMSN >3.00. It should be noted that the majority of candidate OGs selected for annotation (*AURKA, RIT1, YAP1*, and *YES1*) are found among the genes shared by the two datasets, showing strong signals of positive selection for missense substitutions.

## Negatively selected genes

We have selected putative TEGs from the list of candidate cancer genes using the restriction that we have excluded genes with rSMN <0.5 to eliminate OGs and TSGs (*Supplementary file 3*). Out of the 505 genes, we have subjected *CX3CR1, FOXG1, FOXP2, G6PD, MAPK13, MLLT3, NOVA1, PNCK, RUNX2, SLC16A3, SLC2A1, SLC2A8, TBP, TBXA2R, TP73*, and *TRIB2* to further analysis.

Our analyses have confirmed that in the majority of cases (*CX3CR1, FOXG1, G6PD, MAPK13, NOVA1, PNCK, SLC16A3, SLC2A1, SLC2A8, TBXA2R, TP73, TRIB2*) the high synonymous-to-nonsynonymous and nonsense mutation rates could be interpreted as evidence for purifying selection during tumor evolution. There were, however, several examples (e.g. *DSPP, FOXP2, MLLT3, RUNX2, TBP*) where high synonymous-to-nonsynonymous and nonsense mutation rates were found to reflect increased rates of synonymous substitution (due to the presence of mutation hotspots), rather than decreased rates of nonsynonymous and nonsense substitutions that could be due to purifying selection (for details see Appendix 1).

Annotations of the genes *CX3CR1, FOXG1, G6PD, MAPK13, NOVA1, PNCK, SLC16A3, SLC2A1, SLC2A8, TBXA2R, TP73*, and *TRIB2* have confirmed that all of them play important roles in carcinogenesis (see Appendix 1) permitting their assignment to various cellular processes of cancer hallmarks (*Table 1.*). As discussed below (and in Appendix 1), they fulfill pro-oncogenic functions by

promoting cell proliferation (*FOXG1*, *MAPK13*, *PNCK*, *TRIB2*), evasion of cell death (*MAPK13*, *PNCK*, *TP73*), replicative immortality (*NOVA1*), reprogramming of energy metabolism of cancer cells (*G6PD*, *SLC16A3*, *SLC2A1*, *SLC2A8*), inducing tumor promoting inflammation (*CX3CR1*, *MAPK13*) and invasion and metastasis (*CX3CR1*, *TBXA2R*). In view of the pro-oncogenic role of these proteins, it is noteworthy, that *G6PD*, *MAPK13*, *PNCK*, *SLC16A3*, and *SLC2A1* are among the candidate cancer genes identified by forward genetic screens in mice (*Abbott et al., 2015*).

Comparison of our list of 505 negatively selected genes (CG_SO$^{2SD}$_rSMN > 0.5) with those identified by others have revealed very little similarity (*Supplementary file 3*). Out of the 147 genes of *Weghorn and Sunyaev, 2017*, only one is present in the list of top-ranking negatively selected genes identified in the present study. Similarly, only four of the 25 genes of *Zapata et al., 2018* and only five of the 91 genes of *Pyatnitskiy et al., 2015* are found in our list of negatively selected genes (*Supplementary file 3*).

We observed a greater similarity when we compared our list of negatively selected genes with that of *Zhou et al., 2017*; 32 of the 112 genes identified by *Zhou et al., 2017* are also present among the 505 negatively selected genes identified in the present work (*Supplementary file 3*). It is noteworthy that top-ranking genes present in both lists include the *ACKR3*, *TBP*, and *MLLT3* genes. As discussed in Appendix 1, the apparent signals of negative selection (high synonymous-to-nonsynonymous rates) of genes like *DSPP*, *FOXP2*, *MLLT3*, *RUNX2*, and *TBP* may reflect the presence of mutation hotspots generating silent mutations and not purifying selection. *Zhou et al., 2017* have also noted that "some cancer genes also show negative selection in cancer genomes, such as the OG *MLLT3*" and that "interestingly, *MLLT3* has recurrent synonymous mutations at amino acid positions 166 to 168". Apparently, the authors did not realize that this observation of recurrent silent substitutions (in a poly-Ser region of the protein) questions the validity of the claim that the unusually low nonsynonymous to synonymous rate is due to negative selection (for more detail see Appendix 1).

In summary, the pro-oncogenic, negatively selected genes annotated and validated in the present work are missing from the earlier lists of negatively selected genes (*Zhou et al., 2017*; *Pyatnitskiy et al., 2015*; *Weghorn and Sunyaev, 2017*; *Zapata et al., 2018*). A possible explanation for the lack of similarity of top-ranking negatively selected genes identified in the present study with those identified by others is that we have limited our work to transcripts that have at least 100 somatic mutations. It is noteworthy that a large fraction of genes identified by others did not pass this requirement (see Materials and methods).

We have also analyzed the genes present in dataset CG_SO*$^{2SD}$ rSMN >1.50, that is, candidate cancer genes for which the observed rSMN values are more than 1.5-fold higher than expected taking into account mutational signature-derived substitution probabilities of tumors (*Supplementary file 4*). We have found that 200 (86.5%) of the 231 genes present in this dataset are also present in dataset CG_SO$^{2SD}$ rSMN >0.5. It should be noted that the majority of candidate TEGs selected for annotation (*CX3CR1*, *FOXG1*, *FOXP2*, *MAPK13*, *MLLT3*, *NOVA1*, *RUNX2*, *SLC16A3*, *SLC2A8*, *TBP*, *TBXA2R*, and *TRIB2*) are found among the 200 genes shared by the two datasets and that show the strongest signals of negative selection for missense and nonsense substitutions.

## Negative selection, cell essentiality, and tumor essentiality of genes

As we have emphasized in the Introduction, the conclusions drawn from earlier studies searching for signs of negative selection are highly controversial. A highly publicized study has propagated the conclusion that negative selection has no role in tumor evolution (*Martincorena et al., 2017*; *Bakhoum and Landau, 2017*; *Koch, 2017*; *Vitale and Galluzzi, 2018*). *Martincorena et al., 2017* have argued that the practical absence of purifying selection during tumor evolution is due to the buffering effect of diploidy and functional redundancy of most cellular pathways.

A recent study has examined the influence of functional redundancy on the essentiality of genes (*De Kegel and Ryan, 2019*). The authors have used CRISPR score profiles of 558 genetically heterogeneous tumor cell lines and converted continuous values of gene CRISPR scores to binary essential and nonessential calls. These analyses have shown that 1014 genes belong to a category of 'broadly essential genes', that is, these genes were found to be essential in at least 90% of the 558 cell lines. *De Kegel and Ryan, 2019* have shown that, compared to singleton genes, paralogs are less frequently essential and that this is more evident when considering genes with multiple paralogs or

with highly sequence-similar paralogs. In harmony with these conclusions, *López et al., 2020* have found that purifying selection of essential genes is significant in early phases of tumor evolution but in later phases whole-genome doubling allows the accumulation of deleterious alterations.

Since the group of negatively selected genes identified by *Weghorn and Sunyaev, 2017* were shown to be enriched in cell-essential genes (*Wang et al., 2015a*), the authors have proposed that the major cause of negative selection during tumor evolution is the maintenance of genes that are responsible for basal cellular functions. Nevertheless, *Weghorn and Sunyaev, 2017* have pointed out that negative selection is also expected to act on neoantigens, expanding the possible scope of purifying selection beyond cell essentiality.

Although analyses of negatively selected genes have led *Zapata et al., 2018* to conclude, "Processes that are most strongly conserved are those that play fundamental cellular roles such as protein synthesis, glucose metabolism, and molecular transport" they also emphasized the possible importance of less basic functions. Since the immune system is capable of discriminating cancer cells by recognizing mutated epitope sequences the authors have hypothesized that native epitope sequences would be protected from nonsynonymous mutations during tumor evolution. In harmony with this hypothesis, the authors have observed signals of selection in the immunopeptidome and proteins of the epitope presentation machinery, arguing for their importance in the evasion of immune surveillance by tumors.

Gene Ontology analysis of the negatively selected 'essential cancer proteins' identified by *Pyatnitskiy et al., 2015* have revealed enrichment of essential proteins related to membrane and cell periphery, leading the authors to speculate that this could be a sign of immune system-driven negative selection of cancer neo-antigens.

In summary, there is some disagreement about the significance of purifying selection in tumor evolution and whether tumor essential functions can be equated with basic cellular functions.

In order to assess the contribution of cell-essentiality to purifying selection during tumor evolution, we have plotted various measures of negative selection of human genes as a function of their cell-essentiality scores determined by *De Kegel and Ryan, 2019*. These analyses have shown that there is a very weak, positive correlation (Pearson's r = 0.05345, p<0.05) between rSMN (a measure of purifying selection) and the cell-essentiality scores of transcripts (*Figure 8*, *Supplementary file 5*). Since, by definition, there is a negative correlation between the essentiality of genes and their cell-essentiality scores (*De Kegel and Ryan, 2019*), our data indicate that cell essentiality does not contribute significantly to purifying selection during tumor evolution.

It is also noteworthy that the cell essentiality scores of negatively selected genes (CG_SO$^{2SD}$ rSMN >0.5) are not significantly different from those of PGs (*Figure 8*, *Supplementary file 5*). Comparison of CRISPR scores (−0.07665 ± 0.17269) of the cluster of negatively selected genes of CG_SO$^{2SD}$ rSMN >0.5 listed in *Supplementary file 3* with CRISPR scores (−0.09506 ± 0.24168) of the cluster of PGs (PG_SO$^{r3\_1SD}$) revealed that they are not significantly different (p>0.05). This indicates that basic cell-essentiality per se does not explain the purifying selection observed for this cluster of genes.

Comparison of the lists of negatively selected genes identified in the present work with the 1014 'broadly essential genes' defined by *De Kegel and Ryan, 2019* has revealed that there is practically no overlap between the two groups. Only six of the 1014 broadly essential genes are included in our list of negatively selected genes (*Supplementary file 3*). This observation also suggests that cell-essentiality defined by CRISPR scores determined experimentally on cell lines is not relevant for negative selection during tumor evolution in vivo.

Our analyses of cases of strong purifying selection suggest that it has more to do with a function *specifically* required by the tumor cell for its growth, survival, and metastasis than with general basic cellular functions (*Table 1*). It is noteworthy in this respect, that the genes showing the strongest signals of negative selection include several plasma membrane receptor proteins (e.g. *ACKR3, CCR2, CCR5, CX3CR1, TBXA2R*) that cancer cells utilize to promote migration, invasion, and metastasis (Appendix 1). Significantly, these proteins exert their biological functions (in cell migration, inflammation, angiogenesis etc.) primarily at the organism level, therefore their cell-essentiality scores may have little to do with their overall essentiality for tumor growth and metastasis. Inspection of the data of *De Kegel and Ryan, 2019* shows that *ACKR3, CX3CR1, TBXA2R* were not assigned to the essential category in any of the 558 tumor cell lines tested.

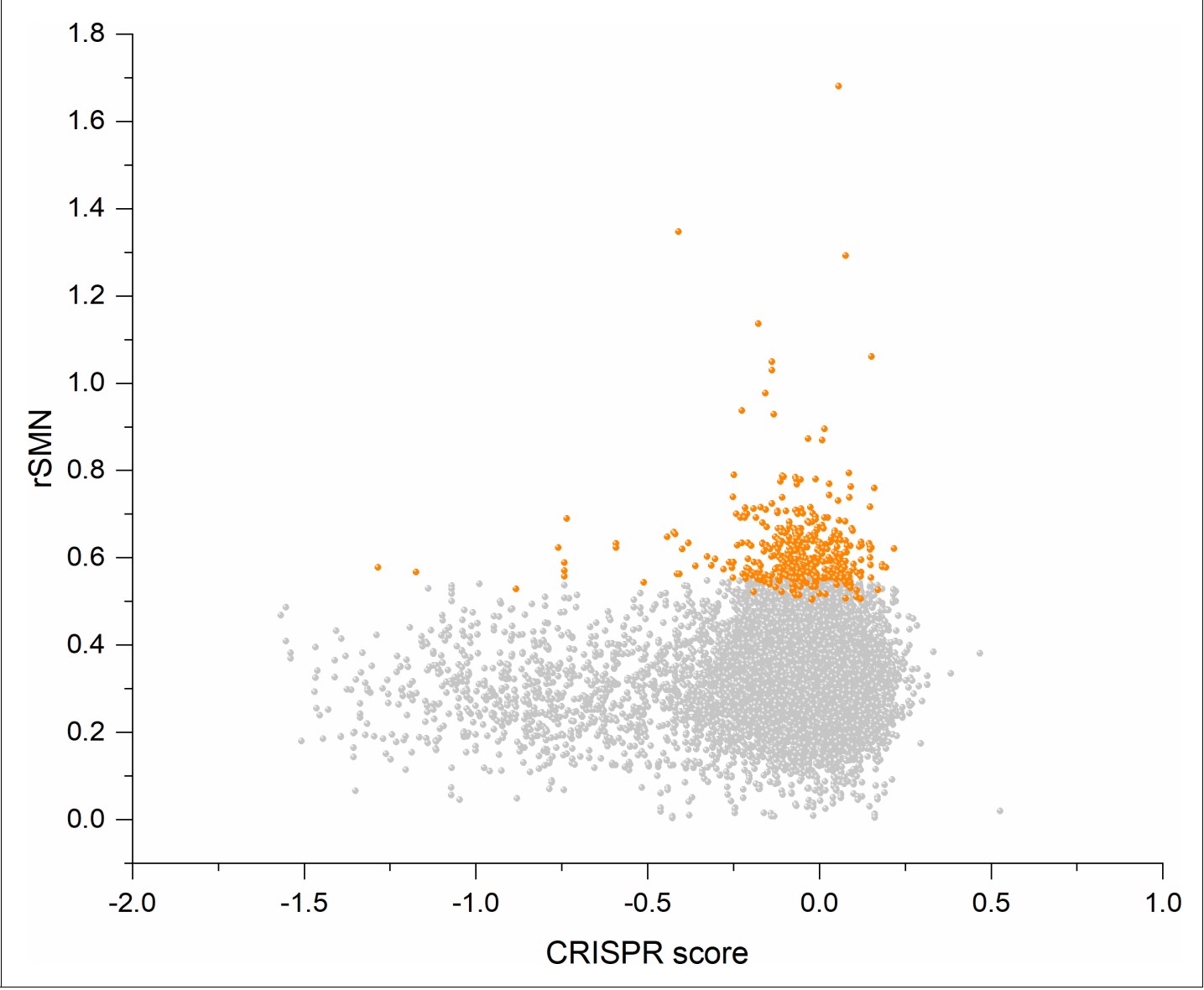

**Figure 8.** Cell-essentiality scores of human genes and negative selection during tumor evolution. The figure shows the results of the analysis of transcripts containing at least 100 subtle, confirmed somatic, non-polymorphic mutations from tumor tissues. The abscissa indicates the cell-essentiality score of the genes, the ordinate shows the rSMN parameters of the transcripts. Each ball represents a human transcript. Transcripts showing strongest signals of negative selection (CG_SO$^{2SD}$ rSMN >0.5) are represented by dark orange balls.

Negatively selected, TEGs identified in the present study do include proteins involved in cell-level processes: they promote cell proliferation (*FOXG1*, *MAPK13*, *PNCK*, and *TRIB2*), evasion of cell death (*MAPK13*, *PNCK*, and *TP73*), replicative immortality (e.g. *NOVA1*), or they are crucial for the reprogramming of energy metabolism in cancer cells (e.g. *GAPD*, *SLC16A3*, *SLC2A1*, and *SLC2A8*). Nevertheless, their negative selection is unlikely to be a mere reflection of their basic cellular functions. Rather, it reflects the exceptional role of the corresponding cancer hallmarks (evasion of cell death, replicative immortality, reprogramming of metabolism) in carcinogenesis (*Figure 1*). In harmony with this conclusion *NOVA1*, *SLC16A3*, *SLC2A8*, and *TP73* were assigned to the essential category by *De Kegel and Ryan, 2019* in less than 10% of the 558 tumor cell lines tested. *SLC2A1* (glucose transporter 1) is an exception in as much as it was found to be cell-essential in 41% of the cell lines. Significantly, several nutrient transporter genes (*SLC16A3*, *SLC2A1*, and *SLC2A8*) were found among the genes showing the strongest signs of purifying selection. It must be mentioned

here that *Zapata et al., 2018* have also noted that the glucose transporters *SLC2A1* and *SLC2A8* and the lactate transporter *SLC16A3* show signs of purifying selection, although they did not list these genes among the 25 genes with significant negative selection.

The most likely explanation for the tumor essentiality of transporter protein genes *SLC16A3, SLC2A1*, and *SLC2A8* is that tumor cells have an increased demand for nutrients and this demand is met by enhanced cellular entry of nutrients through upregulation of specific transporters (*Ganapathy et al., 2009*). The uncontrolled cell proliferation of tumor cells involves major adjustments of energy metabolism in order to support cell growth and division in the hypoxic microenvironments in which they reside. Otto Warburg was the first to observe an anomalous characteristic of cancer-cell energy metabolism: even in the presence of oxygen, cancer cells limit their energy metabolism largely to glycolysis, leading to a state that has been termed 'aerobic glycolysis' (*Warburg, 1956a*; *Warburg, 1956b*). Cancer cells are known to compensate for the lower efficiency of ATP production through glycolysis than oxidative phosphorylation by upregulating glucose transporters, such as facilitated glucose transporter member 1, GLUT1 (encoded by the *SLC2A1* gene), thus increasing glucose import into the cytoplasm (*Jones and Thompson, 2009*; *DeBerardinis et al., 2008*; *Hsu and Sabatini, 2008*).

The markedly increased uptake of glucose has been documented in many human tumor types, by visualizing glucose uptake through positron emission tomography. The reliance of tumor cells on glycolysis is also supported by the hypoxia response system: under hypoxic conditions, not only glucose transporters but also multiple enzymes of the glycolytic pathway are upregulated (*Jones and Thompson, 2009*; *DeBerardinis et al., 2008*; *Semenza, 2010a*; *Semenza, 2010b*; *Kroemer and Pouyssegur, 2008*).

In our view, the central role of GLUT1 in cancer metabolism is reflected by the fact that the *SLC2A1* gene encoding this glucose transporter is among the genes that show the strongest signals of purifying selection. The key importance of GLUT1 in cancer may be illustrated by the fact that high levels of GLUT1 expression correlates with a poor overall survival and is associated with increased malignant potential, invasiveness, and poor prognosis (*Wang et al., 2017a*; *Deng et al., 2018*; *de Castro et al., 2019*). The strict requirement for GLUT1 in the early stages of mammary tumorigenesis highlights the potential for glucose restriction as a breast cancer preventive strategy (*Wellberg et al., 2016*). The tumor essentiality of GLUT1 may also be illustrated by the fact that knockdown of GLUT1 inhibits cell glycolysis and proliferation and inhibits the growth of tumors (*Xiao et al., 2018*). In view of its essentiality for tumor growth, GLUT1 is a promising target for cancer therapy (*Shibuya et al., 2015*; *Noguchi et al., 2016*; *Chen et al., 2017d*).

Recent studies suggest that the *YAP1-TEAD1-GLUT1* axis plays a major role in reprogramming of cancer energy metabolism by modulating glycolysis (*Lin and Xu, 2017*). These authors have shown that *YAP1* and *TEAD1* are involved in transcriptional control of the glucose transporter *GLUT1*, whereas knockdown of *YAP1* inhibited glucose consumption, and lactate production of breast cancer cells, overexpression of GLUT1 restored glucose consumption and lactate production.

Besides GLUT1 another glucose transporter, GLUT8 (encoded by the *SLC2A8* gene) also shows strong signals of negative selection, arguing for its importance in tumor survival. In harmony with this interpretation, there is evidence that GLUT8 is overexpressed in and is required for proliferation and viability of tumors (*Goldman et al., 2006*; *McBrayer et al., 2012*).

Due to abnormal conversion of pyruvic acid to lactic acid even under normoxia, the altered metabolism of glucose consuming tumors must rapidly efflux lactic acid to the microenvironment to maintain a robust glycolytic flux and to prevent poisoning themselves (*Mathupala et al., 2007*). Survival and maintenance of the glycolytic phenotype of tumor cells is ensured by monocarboxylate transporter 4 (MCT4, encoded by the *SLC16A3* gene) that efficiently transports L-lactate out of the cell (*Ganapathy et al., 2009*). Significantly, MCT4, encoded by the *SLC16A3* gene also shows strong signals of negative selection, in harmony with its importance in tumor survival. As high metabolic and proliferative rates in cancer cells lead to production of large amounts of lactate, extruding transporters are essential for the survival of cancer cells as illustrated by the fact that knockdown of MCT4 increased tumor-free survival and decreased in vitro proliferation rate of tumor cells (*Andersen et al., 2018*). Using a functional screen *Baenke et al., 2015* have also demonstrated that monocarboxylate transporter four is an important regulator of breast cancer cell survival: MCT4 depletion reduced the ability of breast cancer cells to grow, suggesting that it might be a valuable therapeutic target. In harmony with the essentiality of MCT4 for tumor growth, several studies

indicate that expression of the hypoxia-inducible monocarboxylate transporter MCT4 is increased in tumors and its expression correlates with clinical outcome, thus it may serve as a valuable prognostic factor (*Witkiewicz et al., 2012*; *Doyen et al., 2014*; *Baek et al., 2014*). Consistent with the key importance of MCT4 for the survival of tumor cells, its selective inhibition to block lactic acid efflux appears to be a promising therapeutic strategy against highly glycolytic malignant tumors (*Choi et al., 2016*; *Todenhöfer et al., 2018*; *Choi et al., 2018*; *Zhao et al., 2019b*).

Interestingly, the thromboxane A2 receptor gene (*TBXA2R*) as well as several chemokine receptor protein genes (*CCR2, CCR5, CX3CR1*) were also found among the genes showing strong signs of purifying selection (see Appendix 1). (Note that *Pyatnitskiy et al., 2015* have also identified *CCR5* as a negatively selected gene). The most likely explanation for their essentiality for tumor growth is that tumor cells rely on these receptors in various steps of invasion and metastasis (see Appendix 1). It is noteworthy in this respect that another member of the family of chemokine receptors, the atypical chemokine receptor 3, *ACKR3* is also among the genes showing very high values of rSMN, suggesting negative selection of missense and nonsense mutations (*Supplementary file 3*). (Note that *Zhou et al., 2017* have also identified *ACKR3* as a negatively selected gene). Significantly, *ACKR3* is a well-known OG, present in Tier 1 of the Cancer Gene Census. Several studies support the key role of *ACKR3* in tumor invasion and metastasis (*Li et al., 2014*; *Stacer et al., 2016*; *Zhao et al., 2017*; *Puddinu et al., 2017*; *Melo et al., 2018*; *Qian et al., 2018*). Since knock-down or pharmacological inhibition of *ACKR3* has been shown to reduce tumor invasion and metastasis, ACKR3 is a promising therapeutic target for the control of tumor dissemination (for further details see Appendix 1).

## Negative selection of germline mutations in the human population versus negative selection of somatic mutations in cancer

The data discussed in the previous section indicate that the importance ('essentiality') of a given gene is a question of perspective. Cell-essential genes may be non-essential for tumor growth, whereas TEGs with tumor-specific functions do not necessarily have cell-essential functions. Similarly, we may assume that the importance of a gene might be quite different from the perspective of tumor cells and from the perspective of the entire organism. One could speculate that somatic mutations of genes with functions that have no relevance for tumor growth (PGs) experience neutral evolution during tumor growth, whereas germline mutations of the same genes may be subject to purifying selection at the level of organismal evolution, as is true for the majority of genes (*Gorlov et al., 2006*). One may also assume that genes with tumor essential, tumor-specific functions may be subject to purifying selection during both tumor evolution and organism evolution, but the strength of purifying selection of these genes is increased in tumors relative to those of genes that do not have tumor-specific functions.

To test these assumptions, we have determined the signals of selection of germline mutations (*Supplementary file 6*) and compared them with those determined for the same genes in the case of somatic mutations of cancer. Comparison of the patterns of germline and somatic mutations of human transcripts (*Supplementary file 7*) has revealed that the proportion of silent substitutions is significantly higher for germline mutations than for somatic mutations of tumors (fS$^g$: 0.33900 versus fS$^s$: 0.24604, p<0.05). Conversely, the proportions of nonsense and missense mutations are significantly lower for germline mutations than for somatic mutations of tumors (fN$^g$: 0.02329 versus fN$^s$: 0.04669, p<0.05; fM$^g$: 0.63771 versus fM$^s$: 0.70727, p<0.05). These observations are in harmony with the dominance of purifying selection in the human population (*Gorlov et al., 2006*).

As shown in *Figure 9*, the pattern of the distribution of transcripts in 3D scatter plots of fM, fN, and fS parameters for germline mutations are strikingly different from those observed in the case of fM, fN, and fS parameters of somatic mutations in cancer (compare *Figure 9A and B*). In addition to a general shift of germline mutations to lower fN and fM and higher fS values, in the case of germline mutations the fN, fM, and fS parameters of transcripts of TSGs, OGs, and TEGs do not separate from those of the central cluster of genes. Similarly, the distribution of transcripts in 3D scatter plots of rS**, rM**, and rN** parameters for germline mutations are different from those observed in the case of rS*, rM*, and rN* parameters of somatic mutations in cancer (compare *Figure 9C and D*): cancer genes do not separate from the central cluster of genes.

Comparison of the fS, rSM, and rSMN parameters of germline and somatic mutations of transcripts (*Figure 10*, *Supplementary file 7*) has shown that there is only weak correlation between the strength of purifying selection of genes during tumor evolution and organismal evolution. The

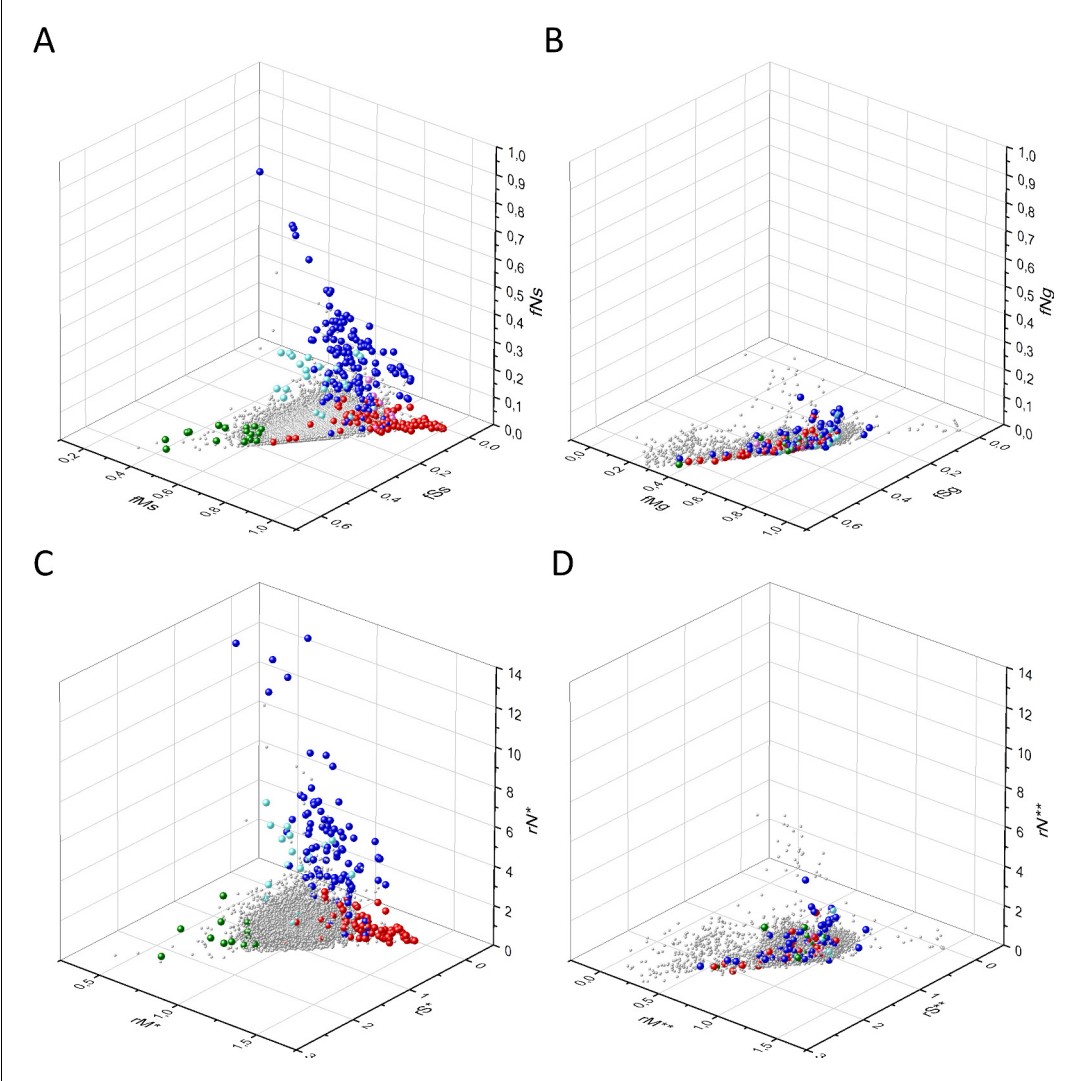

**Figure 9.** Comparison of the patterns of germline mutations of genes with those of somatic mutations observed during tumor evolution. Panel A: fS, fM, and fN scores of somatic mutations in cancer, Panel B: fS, fM, and fN scores of germline mutations. Panel C: rS*, rM*, and rN* scores of somatic mutations in cancer, Panel D: rS**, rM**, and rN** scores of germline mutations. Each ball represents a human transcript. The positions of transcripts of the genes identified by *Vogelstein et al., 2013* as oncogenes (OGs, large red balls) or tumor suppressor genes (TSGs, large blue balls) are highlighted. Novel proto-oncogenes, tumor suppressors and tumor essential genes identified in the present work are highlighted in magenta, cyan, and green, respectively.

Pearson's r values for the correlations of the fS, rSM, and rSMN parameters of germline and somatic mutations are 0.1127, 0.05757, and 0.02635, p<0.05, respectively.

These comparisons have also revealed that – relative to other genes – the candidate TEGs identified in the present study (CG_SO2SD_rSMN >0.5) display significantly stronger signals of purifying selection during tumor evolution than during organismal evolution (*Figure 10*, *Supplementary file 7*). The fS, rSM, and rSMN parameters of somatic mutations of candidate TEGs are significantly higher than those of other genes (fS[s]: 0.38322 versus 0.24045, p<0.05; rSM[s]: 0.66013 versus 0.34375, p<0.05; rSMN[s]: 0.62774 versus 0.32356, p<0.05). The fS, rSM, and rSMN parameters of the germline mutations of candidate TEGs, however, differ much less from the corresponding parameters of other genes (fS[g]: 0.36487 versus 0.33831, p<0.05; rSM[g]: 0.64054 versus 0.56394, p<0.05; rSMN[g]: 0.61264 versus 0.56178, p<0.05). These observations indicate that the negative selection of candidate TEGs during tumor evolution is not a simple reflection of their essentiality at the organism level; it is more likely that they serve tumor-specific functions.

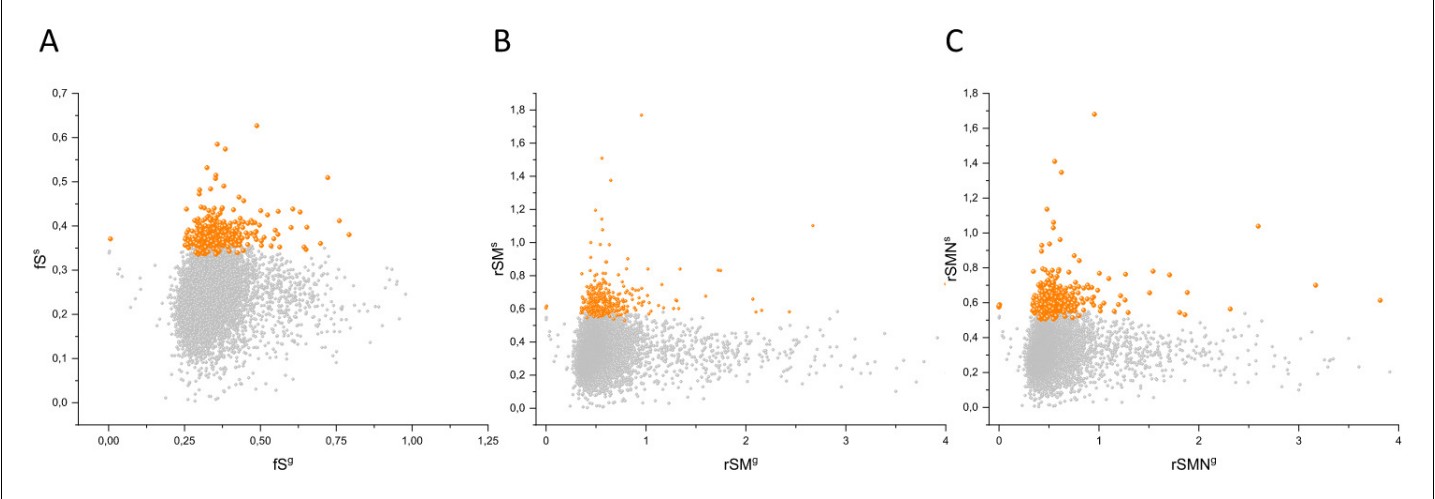

**Figure 10.** Comparison of fS, rSM, and rSMN scores of genes determined for somatic mutations in tumors with those determined for germline mutations. The abscissas indicate the $fS^g$ (panel A), $rSM^g$ (panel B), and $rSMN^g$ (panel C) scores of germline mutations of human genes and the ordinates shows the corresponding $fS^s$, $rSM^s$, and $rSMN^s$ scores of somatic mutations of tumors for the same genes. Each ball represents a human gene. Transcripts showing the strongest signals of negative selection during tumor evolution (CG_SO$^{2SD}$ rSMN >0.5) are represented by dark orange balls.

In order to assess the contribution of cell-essentiality to purifying selection during organismal evolution we have plotted $rSMN^g$, a measure of negative selection of germline mutations of human genes, as a function of their cell-essentiality scores determined by *De Kegel and Ryan, 2019*. These analyses have shown that there is a very weak negative correlation (Pearson's r = −0.03662, p<0.05) between the strength of purifying selection of transcripts ($rSMN^g$) and their cell-essentiality scores (*Figure 11*, *Supplementary file 7*). This observation also indicates that essentiality of cell-level functions measured by cell-essentiality scores contribute to, but do not explain the strength of purifying selection observed during organismal evolution.

## Discussion

One of the major goals of cancer research is to identify all 'cancer genes', that is genes that play a role in carcinogenesis. In the last two decades, several types of approaches have been developed to achieve this goal, but the implicit assumption of most of these studies was that a distinguishing feature of cancer genes is that they are positively selected for mutations that drive carcinogenesis. As a result of combined efforts, the PCAWG driver list identifies a total of 722 protein-coding genes as cancer driver genes and 22 non-coding driver mutations (*Rheinbay et al., 2020*; *Campbell et al., 2020*).

In a recent editorial, commenting on a suite of papers on the genetic causes of cancer, Nature has expressed the view that the core of the mission of cancer-genome sequencing projects—to provide a catalogue of driver mutations that could give rise to cancer—has been achieved (*Editorial, 2020*). It is noteworthy, however, that, although on average, cancer genomes were shown to contain four to five driver mutations, in around 5% of cases no drivers were identified in tumors (*Campbell et al., 2020*). As pointed out by the authors, this observation suggests that cancer driver discovery is not yet complete, possibly due to failure of the available bioinformatic algorithms. The authors have also suggested that tumors lacking driver mutations may be driven by mutations affecting cancer-associated genes that are not yet described for that tumor type, however, using driver discovery algorithms on tumors with no known drivers, no individual genes reached significance for point mutations (*Campbell et al., 2020*).

In our view, these observations actually suggest that a rather large fraction of cancer genes remains to be identified. Assuming that tumors, on average, must have driver mutations affecting at least four or five cancer genes and that known and unknown cancer genes play similar roles in carcinogenesis, the observation that a 0.05 fraction of tumors has no known drivers (i.e. they are driven

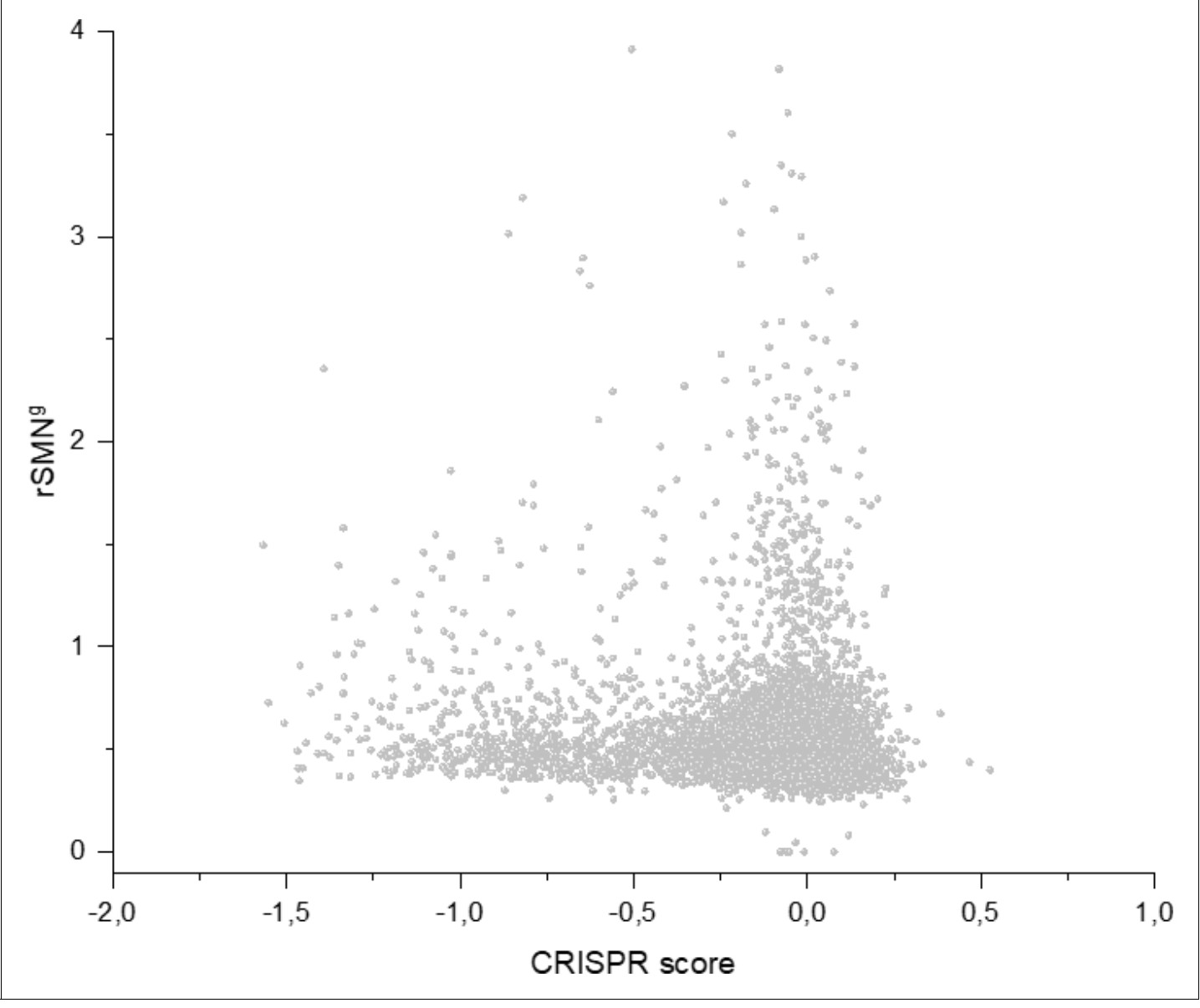

**Figure 11.** Cell-essentiality scores of human genes and negative selection on single-nucleotide polymorphisms (SNPs). The figure shows the results of the analysis of transcripts containing at least 100 polymorphic mutations. The abscissa indicates the cell-essentiality score of the genes, the ordinate shows the $rSMN^g$ parameters of the transcripts. Each ball represents a human transcript. Note that there is a weak negative correlation (Pearson's r = −0.03662, p<0.05) between the strength of purifying selection of transcripts ($rSMN^g$) and their cell-essentiality scores.

by four to five unknown cancer drivers) indicates that about half of the drivers is still unknown. If we assume that ~50% of cancer genes is still unknown 3–6% ($0.5^5$–$0.5^4$, i.e. 0.03125–0.0625 fraction) of tumors is expected to lack any of the known driver genes, and to be driven by four or five unknown driver mutations. Since the list of known drivers used in the study of the ICGC/TCGA Pan-Cancer Analysis of Whole Genomes Consortium (*Campbell et al., 2020*) comprises 722 driver genes, these observations suggest that hundreds of cancer driver genes remain to be identified.

In the present work, we have used analyses that combined multiple types of signals of selection, permitting improved detection of positive and negative selection. Our analyses have identified a large number of novel positively selected cancer gene candidates, many of which could be shown to play significant roles in carcinogenesis as tumor suppressors and OGs. Significantly, our analyses have identified a major group of human genes that show signs of negative selection during tumor

evolution, suggesting that the integrity of their function is essential for the growth and survival of tumor cells. Our analyses of representative members of negatively selected genes have confirmed that they play crucial pro-oncogenic roles in various cancer hallmarks (*Table 1*). It is important to emphasize that a survey of the group of OGs and pro-oncogenic TEGs reveals that they form a continuum in as much as there are numerous known OGs where negative selection also dominates (e.g. *ACKR3, BCL2*).

Although several groups have investigated the role of negative selection in tumor evolution earlier (*Zhou et al., 2017*; *Pyatnitskiy et al., 2015*; *Weghorn and Sunyaev, 2017*; *Martincorena et al., 2017*; *Zapata et al., 2018*; *López et al., 2020*; *Tilk et al., 2020*; *Van den Eynden and Larsson, 2017*), the study that received the greatest attention has reached the conclusion that negative selection has no role in tumor evolution (*Martincorena et al., 2017*; *Bakhoum and Landau, 2017*; *Koch, 2017*; *Vitale and Galluzzi, 2018*). The data presented here contradict the latter conclusion.

We believe that the approach reported here will promote the identification of numerous novel OGs, TSGs, and pro-oncogenic TEGs that may serve as therapeutic targets.

## Materials and methods

### Somatic mutation data

Cancer somatic mutation data were extracted from COSMIC v88, the Catalogue Of Somatic Mutations In Cancer (https://cancer.sanger.ac.uk/cosmic/download), which includes single nucleotide substitutions and small insertions/deletions affecting the coding sequence of human genes. The downloaded file (CosmicMutantExport.tsv, release v88) contained data for 29,415 transcripts (*Supplementary file 8*). For all subsequent analyses we have retained only transcripts containing mutations that were annotated under' Mutation description' as substitution or subtle insertion/deletion. This dataset contained data for 29,405 transcripts containing 6,449,721 mutations (substitution and short indels, SSI) and 29,399 transcripts containing 6,141,650 substitutions only (SO). *Supplementary file 9* contains the metadata for these SO and SSI datasets.

Since we were interested in the selection forces that operate during tumor evolution, only confirmed somatic mutations were included in our analyses. In COSMIC such mutations are annotated under' Mutation somatic status' as Confirmed Somatic, that is confirmed to be somatic in the experiment by sequencing both the tumor and a matched normal tissue from the same patient. *Supplementary file 10* indicates the contribution of major tumor types ('Tumor Primary site') to the somatic mutations of the dataset. As to' Sample Type, Tumor origin': we have excluded mutation data from cell-lines, organoid-cultures, xenografts since they do not properly represent human tumor evolution at the organism level. We have found that by excluding cell lines we have eliminated many artifacts of spurious recurrent mutations caused by repeated deposition of samples taken from the same cell-line at different time-points. To eliminate the influence of polymorphisms on the conclusions we retained only somatic mutations flagged 'n' for SNPs. (*Supplementary file 8*).

To increase the statistical power of our analyses, we have limited our work to transcripts that have at least 100 somatic mutations; *Supplementary file 5* contains the metadata for transcripts containing at least 100 confirmed somatic, non polymorphic mutations identified in tumor tissues. Hereafter, unless otherwise indicated, our analyses refer to datasets containing transcripts with at least 100 somatic mutations. This limitation eliminated ~38% of the transcripts that contain very few mutations but reduced the number of total mutations only by 9% (*Supplementary file 8*).

It should be noted that requiring a higher minimum number of somatic mutations increases the statistical power of the analyses but may disfavor the identification of negatively selected genes that tend to be undermutated. To assess the influence of the cut-off value of the minimum number of mutations on the robustness of the conclusions about negatively or positively selected genes, we have compared the results of analyses in which the minimum number of somatic mutation per gene was set as 0, 50, 100, or 500 (*Supplementary files 11–13*, *Figure 12*).

The choice of the minimum number of somatic mutations was found to have a strong influence on the pattern of observed fN, fS, and fM scores (*Supplementary files 11–13*, *Figure 12*). In the case of dataset N0 (no requirement for a minimum number of mutations), a large number of transcripts with less than 50 substitutions had scores of zero for one or two of the fN, fS, and fM parameters

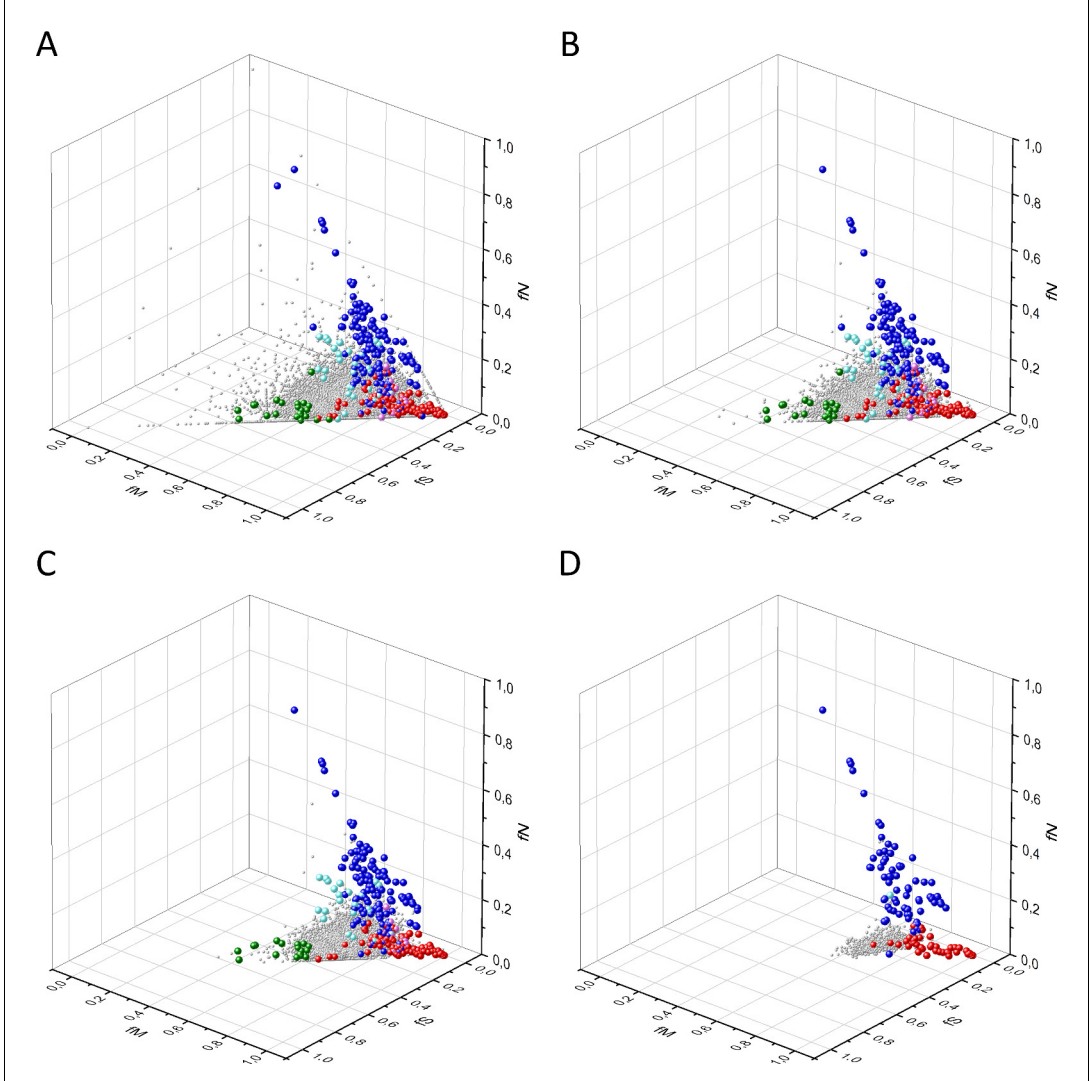

**Figure 12.** Analyses of fS, fM, and fN parameters of datasets N0, N50, N100, and N500 containing transcripts of human protein-coding genes with at least 0, 50, 100, or 500 somatic substitutions in tumors. The figure shows the results of the analysis of 29,333, 21,307, 13,803, and 997 transcripts present in datasets N0 (panel A), N50 (panel B), N100 (panel C), and N500 (panel D), respectively. Axes x, y, and z represent the fractions of somatic single nucleotide substitutions that are assigned to the synonymous (fS), nonsynonymous (fM), and nonsense (fN) categories. Each gray ball represents a human transcript. The positions of transcripts of the genes identified by *Vogelstein et al., 2013* as oncogenes (OGs, large red balls) or tumor suppressor genes (TSGs, large blue balls) are highlighted; novel proto-oncogenes, TSGs, and negatively selected tumor essential genes validated in the present work are represented by large magenta, cyan, and green balls, respectively. It is noteworthy that the requirement of at least 50 somatic mutations per transcript eliminates transcripts where the signal-to-noise ratio is too low to permit detection of signals of selection through the analysis of fS, fM, and fN parameters (compare panel A and panel B). It should also be noted that the requirement of at least 500 somatic mutations per transcript eliminates transcripts of negatively selected genes (compare panel C and panel D), consistent with the view that they tend to be undermutated.

due to the absence of somatic mutations in those categories (*Figure 12A*, *Supplementary file 11*). Increasing the minimum number of somatic mutations per transcript to 50, 100, and 500 resulted in loss of these transcripts and elimination of a diffusely scattered group of transcripts that do not cluster with PGs, known OGs, TSGs, and TEGs (compare *Figure 12A and B,C and D*, *Supplementary file 11*). These observations indicate that we cannot draw valid conclusions about the significance of selection from fN, fS, and fM scores in cases where the number of mutations of a given transcript is too low to permit meaningful analyses. Our analyses have also revealed that a large proportion (22%) of the transcripts unique to the N0$^{2SD}$ dataset (containing fewer than 50

substitutions), correspond to short transcript fragments encoding less than 100 amino acids (*Supplementary file 12*). This finding suggests that the requirement for a minimum number of somatic mutations would not only increase statistical power, but also increases the biological relevance of the conclusions with the elimination of fragments that do not properly represent the full-length coding sequences.

Our analyses, however, have shown that the requirement of more than 500 somatic mutations per transcript (dataset N500) is too stringent. Although the majority of the 86 transcripts of the N500$^{2SD}$ dataset correspond to known OGs (25) or TSGs (48), most OGs and TSGs are not represented in the N500$^{2SD}$ dataset (*Supplementary file 12*). Furthermore, none of the negatively selected TEGs validated in the present study is present among the 86 transcripts of the N500$^{2SD}$ dataset or among the 997 transcripts of the N500 dataset (*Figure 12D*, *Supplementary file 12*). This observation is consistent with the view that since negatively selected genes tend to have fewer mutations, they are less likely to pass the requirement for a high number of somatic mutations.

Our analyses suggest that the choice of 50 or 100 as the minimum number of somatic mutations per transcript represent acceptable trade-off between statistical power and loss of negatively or positively selected genes. As shown in *Supplementary file 12*, both the N50$^{2SD}$ dataset (1846 transcripts) and the N100$^{2SD}$ dataset (1060 transcripts) contained the majority of known OGs, known TSGs, and TEGs validated in the present work, but since the choice of 100 offers higher statistical power, we have used this dataset in our analyses.

Our choice of 100 as the minimum number of somatic mutation per transcript may also have some relevance for the lack of more extensive similarity of our list of negatively selected genes with those identified by others. As shown in *Supplementary file 13*, only 48%, 64%, 77%, and 89% of the negatively selected genes identified by *Weghorn and Sunyaev, 2017*, *Zapata et al., 2018*, *Zhou et al., 2017*, and *Pyatnitskiy et al., 2015*, respectively are present in dataset N100 containing 13,803 transcripts with at least 100 somatic mutations (*Supplementary file 11*). It thus appears that one of the reasons for the differences observed is that, with respect to minimum number of mutations, we have used rather stringent criteria to increase the robustness of our estimates. We wish to point out that, in order to obtain more reliable estimates of purifying selection, *Pyatnitskiy et al., 2015* have also excluded genes from their analysis that carried a low number of mutations but they excluded only those with less than 11 mutations.

The COSMIC database of somatic mutations used in the present study contains data obtained by three main types of sequencing: whole-genome sequencing (WGS), whole-exome sequencing (WES) and targeted sequencing. As shown in *Supplementary file 8*, targeted screens provided substitution mutation data for only 13,120 transcripts of human genes, whereas genome wide screens covered 29,407 human transcripts as opposed to 29,415 transcripts covered by targeted plus genome wide screens. The contribution of targeted screens to somatic point mutations is even more restricted: only 508,124 (8.3%) of the 6,141,650 somatic point mutations of the entire COSMIC database were identified by targeted sequencing (*Supplementary file 8*). To check the impact of targeted sequencing on the dataset, in some analyzes we have used somatic mutation data only from genome-wide screens, excluding those obtained by targeted sequencing. We have found that omission of the data from targeted screens had no significant effect on the conclusions drawn from our analyses. Several factors may explain this observation. First, targeted screens usually focus on known cancer genes and they usually just reinforce the 'known cancer gene' status of the targeted genes. Second, since only a small fraction of the somatic mutations originates from targeted screens their impact is limited even in the case of the targeted genes. Finally, inclusion or omission of data from targeted screens has no impact on the number and pattern of mutations of non-targeted genes identified in genome wide screens.

## Germline mutation data

Information on SNPs affecting the coding regions of human genes was downloaded from the dbSNP database (https://www.ncbi.nlm.nih.gov/snp/). For each SNP, we extracted nucleotide and amino acid variants from the original dbSNP file. In cases where two or three mutant variant was reported for a specific rsID, each variant was treated as an independent polymorphism. The retrieved SNPs were assigned to three functional categories: (i) Nonsense or Stop_gained mutations (N), which change an amino acid-encoding codon into a stop codon, (ii) Missense mutations (M), which change an amino acid into a mutant amino acid, and (iii) Synonymous or silent mutations (S), which do not

change the amino acid. We have focused only on SNPs of genes that were also found to contain at least 100 confirmed somatic, non polymorphic mutations in the COSMIC database (*Supplementary file 5*). *Supplementary file 6* shows the numbers and fractions of SNPs affecting the coding sequences of the various human genes, according to the functional categories of the point mutations.

## Substitution metrics

The 61 sense codons can undergo 549 single base substitutions and, depending on the wild type and mutant codon, each substitution can be assigned to the silent, missense or nonsense mutation category. Out of the 549 single-base substitutions, 392 result in missense mutation, 134 lead to silent mutation, and 23 generate nonsense mutation, thus – assuming equal codon frequency, equal probability of the different types of substitutions and neutrality – the expected fractions of nonsense, missense and silent substituions are fN = 0.04189, fM = 0.71403, and fS = 0.24408, respectively.

Codons, however, differ significantly in the probability that their mutation would lead to nonsense (N), missense (M), or silent (S) mutation (*Supplementary files 14*, *15*, *16*) and since the 61 sense codons (amino acids) do not occur with the same frequency in the coding region of human genes this may have a significant influence on the expected fN, fM, and fS values. We have calculated the probability that a substitution would lead to nonsense, missense or silent mutation taking into account the codon frequency of the proteome of *Homo sapiens* (https://www.kazusa.or.jp/codon/cgi-bin/showcodon.cgi?species=9606). This calculation yielded values of fN = 0.0419, fM = 0.7299, fS = 0.2282 for the proteome, slightly different from the values of fN = 0.0419, fM = 0.7140, fS = 0.2441, assuming equal frequency of codons.

The amino acid composition and codon usage of some individual proteins (especially short fragments) may deviate significantly from average, therefore we have calculated the expected proportion of silent, missense, and nonsense mutations for all transcripts, assuming equal probability of different substitutions classes (*Supplementary file 17*). For these calculations, we have downloaded the coding sequences of 53,190 transcripts of human protein coding genes (All_COSMIC_Genes. fasta.gz) from the COSMIC database (https://cancer.sanger.ac.uk/cosmic) and their codon usage and amino acid composition were determined using the SMS server (https://www.bioinformatics. org/sms2/codon_usage.html, *Stothard, 2000*).

Different classes of substitutions, however, do not occur with equal probability, moreover the various normal and tumor tissues show characteristic differences in the spectrum of substitutions classes (*Alexandrov et al., 2013*; *Alexandrov et al., 2020*). Substitutions are assigned to six classes (C>A, C>G, C>T, T>A, T>C, and T>G) referred to by the pyrimidine of the mutated Watson–Crick base pair. It is of crucial importance to take differences in the probability of the six mutation classes into account since—due to the unique structure of the genetic code—the six types of substitutions differ markedly in the probability that they would lead to nonsense (N), missense (M), or silent (S) mutation of the coding region of protein-coding genes. As shown in *Supplementary files 18–25*, there are significant differences in the impact of different substitution classes on the expected proportion of missense, silent, and nonsense mutations of codons (assuming equal codon frequency). For example, the dominance of C>G increases the proportion of missense substitutions, whereas higher rates of C>T and T>C substitutions increase the proportion of silent substitutions. Since mutation bias favoring C>T substitutions is expected to decrease the ratio of missense to silent mutations, decreased dN/dS values may not be taken as evidence for negative selection in the case of tumors, such as malignant melanoma, where the vast majority of all somatic mutations is C>T substitution (*Van den Eynden and Larsson, 2017*).

To take into account differences in mutation bias, we have calculated the contribution of the C>A, C>G, C>T, T>A, T>C, and T>G mutations to the pattern of single base substitutions in tumors. We have downloaded the files containing 'Mutational Signatures v3.1' and 'Attributions of the SBS Signatures to Mutations in Tumors' from the COSMIC website (https://cancer.sanger.ac.uk/cosmic/signatures/SBS/index.tt). The contributions of C>A, C>G, C>T, T>A, T>C, and T>G mutations to the pattern of Single Base Substitutions in the tumors listed in the PCAWG_sigProfiler_SBS_-signatures_in_samples file are summarized in *Supplementary file 26*. The C>T substitution accounts for the largest fraction of substitutions in most tumors (0.3726), followed by T>C (0.1842), C>A (0.1583), C>G (0.1162), T>G (0.0891), and T>A (0.0796). There are, however, differences in the relative contribution of the six mutation classes to different tumors. For example, the contribution of

C>A mutation is higher than average for colon cancer and lung cancer, the role of C>G mutation is above average for bladder cancer and some breast cancers. The contribution of C>T mutation is very high in the case of skin-melanoma, whereas the T>A mutation contributes significantly to some kidney cancers. The T>C mutation plays a significant role in biliary and liver cancer, whereas the T>G mutation is more significant in colon cancer and esophageal cancer than in other tumors (see *Supplementary file 26*).

In order to correct for the influence of mutation bias on fN, fM, and fS values of transcripts in tumor tissues, we have calculated the expected fN*, fM*, and fS* values for all human transcripts using the average values of the six substitution types observed across tumors (*Supplementary file 27*). It is noteworthy that the average values of expected fN*, fM*, and fS* (fN*=0.04483, fM*=0.69114, and fS*=0.26402) are similar to those (fN = 0.04189, fM = 0.71403, and fS = 0.24408) assuming equal codon frequency and equal probability of the different types of substitutions.

In the case of germline cells, we have also calculated the expected fN**, fM**, and fS** values for all human transcripts using the mutation probabilities characteristic of these cells (*Supplementary file 28*). It has been shown earlier that the human germline mutation spectrum can be recapitulated by a combination of the cancer signatures SBS1 and SBS5 (*Alexandrov et al., 2015*; *Rahbari et al., 2016*; *Heredia-Genestar et al., 2020*). In the present work, we have combined the effect of mutation signatures SBS1 and SBS5 on the germline mutation spectrum of proteins according to the formula (0.1 × SBS1 + 0.9 × SBS5) recommended by *Heredia-Genestar et al., 2020*. It is noteworthy that the average values of expected fN**, fM**, and fS** (fN**=0.03791, fM**=0.68653, and fS**=0.27556) are similar to those expected for tumor tissues (fN*=0.04483, fM*=0.69114, and fS*=0.26402).

## Per-gene detection of selection signals in tumor tissues

We have used two approaches to determine the observed fM, fS, and fN values of transcripts: one in which we have restricted our analyses to single nucleotide substitutions (hereafter referred to as SO for 'substitution only') and a version in which we have also taken into account subtle indels (hereafter referred to as SSI for 'substitutions and subtle indels').

In the first case, we have calculated for each transcript the fraction of somatic substitutions that could be assigned to the synonymous (fS), nonsynonymous (fM), and nonsense mutation (fN) category (*Supplementary file 5* and *9*). In the version that also included data for subtle indels, we have calculated the fraction of mutations corresponding to synonymous substitutions (indel_fS), but have merged nonsynonymous substitutions and short inframe indels in the category of mutations that lead to changes in the amino acid sequence (indel_fM). Nonsense substitutions and short frame-shift indels were included in the third category of mutations (indel_fN) as both types of mutation lead eventually to stop codons that truncate the protein (*Supplementary file 5* and *9*).

Analyses of datasets (*Supplementary file 5*) containing substitutions only have shown that in 3D scatter plots transcripts form a cluster (*Figure 2A*) characterized by values of 0.2436 ± 0.0619, 0.7090 ± 0.0556, and 0.0475 ± 0.0322 for fractions of silent, missense, and nonsense substitutions, respectively. The mean fS, fM, and fN values of the transcripts in this cluster are close to those expected if we assume that the structure of the genetic code has the most important role in determining the probability of somatic substitutions during tumor evolution of human genes (*Supplementary file 29*). Based on the structure of the genetic code, assuming equal usage of the codons and equal probability of different point mutations, in the absence of selection one would expect that a fraction of 0.24408 would be silent, 0.71403 of the single-base substitutions would be missense and 0.04189 would be nonsense mutations.

It is noteworthy, however, that the fS, fM, and fN values of the best known cancer genes (*Vogelstein et al., 2013*) deviate from those characteristic of the majority of human genes (*Figure 2B*). The genes in the central cluster, deviating from mean fM, fS, and fN values by ≤1 SD, are characterized by fraction values of 0.24548 ± 0.03079, 0.71084 ± 0.0274, and 0.04368 ± 0.01572 for synonymous, nonsynonymous and nonsense substitutions, respectively. Note that these values are very close to those expected from the structure of the genetic code in the absence of selection, assuming equal frequency of codons and equal probability of the different classes of mutations (*Supplementary file 29*). This central cluster of genes (*Supplementary file 5*) is hereafter referred to as PG_SO$^{f\_1SD}$ (for Passenger Gene_Substitution Only deviating from mean fM, fS, and fN values by ≤1 SD) because it is likely to be enriched in genes that play no major role in carcinogenesis.

In harmony with earlier observations, the values for OGs show a significant ($p<0.05$) shift of fM to higher values ($0.8563 \pm 0.08224$) relative to those of PGs ($0.71084 \pm 0.0274$), reflecting positive selection for missense mutations (*Supplementary file 29*). On the other hand, the fN values of TSGs are significantly ($p<0.05$) higher ($0.1964 \pm 0.11063$) than those of PGs ($0.04368 \pm 0.01572$), reflecting positive selection for truncating nonsense mutations (*Supplementary file 29*).

The genes (1060 transcripts) with values that deviate from mean values of fS, fM, and fN by more than 2SD, however, are likely to be subject to selection. In harmony with this expectation, this group contains transcripts of the majority of known driver genes (62 OG and 119 TSG driver gene transcripts). This gene set, defined by 2SD cut-off value, is hereafter referred to as CG_SO$^{f\_2SD}$ (for Cancer Gene_Substitution Only deviating from mean fM, fS, and fN values by more than 2SD) because it is likely to be enriched in cancer genes (*Supplementary file 29*). Out of the 1060 transcripts present in CG_SO$^{f\_2SD}$, 737 transcripts are derived from genes that are not included in the OG, TSG, and CGC cancer gene lists (*Supplementary files 5* and *29*). Since the majority of these 737 transcripts have parameters that cluster them with known OGs or TSGs, we assume that they qualify as candidate OGs or TSGs. However, a group of genes deviates from both the central PG cluster and the clusters of OGs and TSGs (*Figure 2C*). The high fS and low fM and fN values of the genes in this cluster suggest that they experience purifying selection during tumor evolution, raising the possibility that they may correspond to TEGs important for the growth and survival of tumors.

Known cancer genes (OGs and TSGs) also separate from the majority of human genes in 3D scatter plots of parameters rSM, rNM, rNS defined as the ratio of fS/fM, fN/fM, fN/fS, respectively (*Figure 3*). The central cluster of genes that deviate from mean rSM, rNM and rNS values by $\leq 1$ SD is hereafter referred to as PG_SO$^{r2\_1SD}$ (Passenger Gene_Substitution Only deviating from mean rSM, rNM, and rNS values by $\leq 1$ SD) since it is likely to be enriched in PGs. Conversely, the group of transcripts that deviate from mean rSM, rNM, and rNS values by more than 2SD is referred to as CG_SO$^{r2\_2SD}$ (Cancer Gene_Substitution Only deviating from mean rSM, rNM, rNS values by more than 2SD) because it is likely to be enriched in cancer genes (*Supplementary file 29*). The CG_SOr2_2SD gene set (780 transcripts) contains the majority of driver gene transcripts (40 transcripts of OGs, 103 transcripts of TSGs genes), 79 transcripts of CGC genes and 558 transcripts derived from 468 genes that are not found in the OG, TSG, and CGC cancer gene lists (*Supplementary file 29*).

In these scatter plots OGs separate from the central cluster in having significantly ($p<0.05$) lower rSM ($0.13971 \pm 0.10621$) and rNM ($0.03936 \pm 0.0313$) values than those of the central cluster of PGs (rSM: $0.34523 \pm 0.06137$; rNM: $0.0607 \pm 0.02595$, *Supplementary file 29*), reflecting positive selection for missense mutations and negative selection of nonsense mutations. Interestingly, in these plots some OGs (e.g. BCL2) have unusually high values of rSM and low values of rNM (e.g. *Figure 3A1, A2* and *Supplementary file 5*) suggesting that in the case of these OGs purifying selection may dominate over positive selection for amino acid changing mutations.

TSGs also separate from the central cluster: they have significantly ($p<0.05$) higher rNS ($3.92588 \pm 5.66261$) and rNM ($0.31524 \pm 0.31575$) values than those of PGs (rNS: $0.18403 \pm 0.09138$; rNM: $0.0607 \pm 0.02595$; *Figure 3A1, A2*. *Supplementary file 29*), reflecting the dominance of positive selection for inactivating mutations.

As mentioned above, the candidate cancer gene set defined by a cut-off value of 2SD also contains 558 transcripts derived from 468 genes that are not found in the OG, TSG, or CGC lists.

Since the majority of these 558 transcripts have parameters that cluster them with known OGs or TSGs, they can be regarded as candidate OGs or TSGs. There is, however, a group of genes that deviate from the clusters of PGs, OGs, and TSGs in that they have unusually high rSM values and low rNM and rNS values. Since these values may be indicative of purifying selection, we assumed that they might correspond to TEGs important for the growth and survival of tumors.

The separation of known cancer genes from the majority of human genes is even more obvious in 3D scatter plots of parameters rSMN, rMSN, and rNSM defined as the ratio of fS/(fM+fN), fM/(fS+fN), and fN/(fS+fM), respectively (*Figure 4 A1, A2*). In these plots, the gene transcripts are present in a three-pronged cluster, with OGs and TSG being present on separate spikes of this cluster (*Figure 4*).

We refer to the central cluster of genes, deviating from mean rSMN, rMSN, and rNSM values by $\leq 1$ SD as PG_SO$^{r3\_1SD}$ (Passenger Gene_Substitution Only deviating from mean rSMN, rMSN, and rNSM values by $\leq 1$ SD) as they are likely to be enriched in PGs. Similarly, we refer to the gene

set defined by 2SD cut-off value (*Supplementary files 5* and *29*) as CG_SO$^{r3\_2SD}$ (Cancer Gene_Substitution Only deviating from mean rSMN, rMSN, and rNSM values by more than 2SD) as it is likely to be enriched in candidate cancer genes. This gene set has 751 transcripts, containing the majority of transcripts of known driver genes (35 OGs, 103 TSGs), 80 transcripts of CGC genes and 533 transcripts (derived from 448 genes) not found in the OG, TSG, and CGC cancer gene lists (*Supplementary files 5* and *29*).

The mean parameters of TSGs differ significantly (p<0.05) from those of PGs in as much as rNSM values of TSGs are higher (0.27937 ± 0.2783) but rSMN (0.10865 ± 0.06128) values are lower than those of PGs (rNSM: 0.04812 ± 0.02561; rSMN: 0.3259 ± 0.09265, *Supplementary file 29*), reflecting the dominance of positive selection for inactivating nonsense mutations.

In the case of OGs the rMSN values are significantly (p<0.05) higher (15.35971 ± 30.07472) and the rSMN values are significantly lower (0.13363 ± 0.10266) than those of PGs (rMSN: 2.58911 ± 0.68355; rSMN: 0.3259±0.09265 *Supplementary file 29*), reflecting positive selection for missense mutations. The rNSM values of OGs (0.03394 ± 0.02621) are also significantly (p<0.05) lower than those of PGs (0.04812 ± 0.02561), reflecting purifying selection avoiding nonsense mutations. Interestingly, some OGs have unusually high scores of rSMN (*Figure 4 A1, A2*, *Supplementary file 5*) suggesting that in these cases (e.g. *BCL2*) purifying selection dominates over positive selection for amino acid changing mutations.

As mentioned above, the candidate cancer gene set defined by a cut-off value of 2SD contains 533 transcripts (derived from 448 genes) not found in the OG, TSG, or CGC lists. Since the majority of these genes have parameters that assign them to the clusters containing OGs or TSGs, they can be regarded as candidate OGs or TSGs. There is, however, a group of genes that deviates from the clusters of PGs, OGs, and TSGs (*Figure 4*). Their high rSMN and low rMSN and rNSM values suggest that they experience purifying selection during tumor evolution, raising the possibility that this group may be enriched in genes essential for the survival of tumors as pro-oncogenes or TEGs.

The three types of analyses described for Substitutions Only (illustrated in *Figures 2–4*) were also carried out for datasets in which both substitutions and subtle indels (Substitutions and Subtle Indels, SSI) were used (for details of these analyzes see Appendix 2).

Comparison of the data obtained by SO and SSI analyses (*Supplementary file 5*) revealed that inclusion of indels has only minor influence on the separation of the clusters of PGs and CGs. The lists of PGs identified with 1SD cut-off values for SO analyes (PG_SO$^{f\_1SD}$, PG_SO$^{r2\_1SD}$, PG_SO$^{r3\_1SD}$) and SSI analyses (PG_SSI$^{f\_1SD}$, PG_SSI$^{r2\_1SD}$, PG_SSI$^{r3\_1SD}$) show more than 90% identity in the case of the relevant SO/SSI pairs (*Supplementary file 30*). Similarly, the lists of CGs identified with 2SD cut-off values for SO analyses (CG_SO$^{f\_2SD}$, CG_SO$^{r2\_2SD}$, CG_SO$^{r3\_2SD}$) and SSI analyses (CG_SSI$^{f\_2SD}$, CG_SSI$^{r2\_2SD}$, CG_SSI$^{r3\_2SD}$) show 78%, 87%, and 92% identity, respectively, for the relevant SO/SSI pairs (*Supplementary file 30*).

The parameters of the 1158 transcripts present in at least one of the various CG_SO$^{2SD}$ lists and the 1333 transcripts present in at least one of the various CG_SSI$^{2SD}$ lists (*Supplementary file 31*) were used to assign them to three distinct clusters. (1) Cluster of genes positively selected for missense mutations and negatively selected for nonsense mutations; (2) Cluster of genes positively selected for nonsense mutations; (3) Clusters of negatively selected genes (see *Figure 2C*, *Figure 3 B1, B2* and *Figure 4 B1, B2*). To check the validity and predictive value of the assumption that the genes assigned to these clusters play significant roles in carcinogenesis, we have selected a number of genes for further analyses from the 1457 transcripts present in the combined list (CG_SO$^{2SD}$_SSI$^{2SD}$) of candidate cancer genes (*Supplementary file 31*). The results of these analyses are summarized in the Results section.

As outlined in the section on Substitution metrics, a limitation of the analyses discussed above is that they did not take into account the impact of differences in mutation probability on the fN, fM, and fS values of transcripts. In order to eliminate this source of error, we have calculated the expected fN*, fM*, and fS* values for all human transcripts using the probability of the six substitution types observed across tumors (*Supplementary file 27*). The various types of observed/expected ratios (rN*, rM*, rS*; rSM*, rNM*, rNS*; rSMN*, rMSN*, and rNSM*) of somatic mutations were calculated for all transcripts (*Supplementary file 32*) and the data were analyzed in 3D scatter plots as described above for the observed values.

As shown in *Figures 5–7*, the distribution of transcripts in these 3D scatter plots are similar to those observed in the corresponding *Figures 2–4*, in that known OGs, TSGs, and TEGs are separated from the central cluster of PGs as well as from each other (*Supplementary file 32*).

### Per-gene detection of selection signals in the database of human single-nucleotide polymorphisms

As a reference, we have carried out similar analyses of the fN, fM, and fS parameters of germline mutations, through the analysis of the human database of human single-nucleotide polymorphisms (SNPs; *Supplementary file 6*). *Supplementary file 33* contains the various types of observed/expected ratios (rN**, rM**, rS**; rSM**, rNM**, rNS**; rSMN**, rMSN**, and rNSM**) of germline mutations calculated for all transcripts. Data were analyzed in 3D scatter plots as described for somatic mutations. Details of these analyses are presented in the Results section.

### Cancer gene list

As the gold standard of 'known' cancer genes we have used the lists of OG and TSGs identified by *Vogelstein et al., 2013*. As another list of known cancer genes we have also used the genes of the Cancer Gene Census (*Sondka et al., 2018*).

### Statistical analyses

The statistical package of Origin 2018 was used for all data processing and statistical analysis. We report details of statistical tests in the Supplementary files of the respective sections. Statistical significance was set as a p value of $< 0.05$.

## Acknowledgements

LB, KK, MT, and LP are supported by the GINOP-2.3.2-15-2016-00001 grant of the Hungarian National Research, Development and Innovation Office (NKFIH), OC is supported by the NVKP_16-1-2016-0005 grant of the Hungarian National Research, Development and Innovation Office (NKFIH).

## Additional information

### Funding

| Funder | Grant reference number | Author |
| --- | --- | --- |
| Hungarian National Research, Development and Innovation Office | GINOP-2.3.2-15-2016-00001 | László Bányai<br>Maria Trexler<br>Krisztina Kerekes<br>László Patthy |
| Hungarian National Research, Development and Innovation Office | NVKP_16-1-2016-0005 | Orsolya Csuka |

The funders had no role in study design, data collection and interpretation, or the decision to submit the work for publication.

### Author contributions

László Bányai, Formal analysis, Validation, Investigation, Methodology, Writing - original draft, Writing - review and editing; Maria Trexler, Krisztina Kerekes, Formal analysis, Validation, Writing - original draft, Writing - review and editing; Orsolya Csuka, Funding acquisition, Validation, Writing - original draft, Writing - review and editing; László Patthy, Conceptualization, Supervision, Funding acquisition, Validation, Methodology, Writing - original draft, Project administration, Writing - review and editing

### Author ORCIDs

László Patthy (iD) https://orcid.org/0000-0003-1329-0484

**Decision letter and Author response**
Decision letter https://doi.org/10.7554/eLife.59629.sa1
Author response https://doi.org/10.7554/eLife.59629.sa2

## Additional files

### Supplementary files

• Supplementary file 1. Comparison of the lists of genes in datasets CG_SSI$^{2SD}$_rNSM > 0.125 and CG_SO$^{2SD}$_rMSN > 3.00 with the lists of cancer genes identified by others (VOG, *Vogelstein et al., 2013*; TAM, *Tamborero et al., 2013*; LAW, *Lawrence et al., 2014*; ABB, *Abbott et al., 2015*; TOR, *Torrente et al., 2016*; ZHO, *Zhou et al., 2017*; MAR, *Martincorena et al., 2017*; BAI, *Bailey et al., 2018*; SON, *Sondka et al., 2018*; ZHA, *Zhao et al., 2019a*). Transcripts of OGs (oncogenes) and TSGs (tumor suppressor genes) of the cancer gene list of *Vogelstein et al., 2013* are highlighted by brick red and blue backgrounds, respectively. Transcripts of CGC genes (SON, *Sondka et al., 2018*) that do not correspond to OGs or TSGs of the cancer gene list of *Vogelstein et al., 2013* are highlighted by yellow background. Novel positively or negatively selected cancer genes validated in the present work are highlighted in dark green background.

• Supplementary file 2. Comparison of the lists of genes in datasets CG_SSI$^{2SD}$_rNSM > 0.125 and CG_SO$^{2SD}$_rMSN > 3.00 with the lists of genes in datasets CG_SO*$^{2SD}$_rNSM > 3 and CG_SO*$^{2SD}$_rMSN > 1.50, respectively.

• Supplementary file 3. Comparison of the list of negatively selected genes, CG$^{2SD}$_rSMN > 0.5 with the lists of negatively selected genes (WEG, ZHOU, ZAPATA, PYATNITSKIY), defined by *Zhou et al., 2017*, *Weghorn and Sunyaev, 2017*, *Zapata et al., 2018*, *Pyatnitskiy et al., 2015*, respectively as well as the list of genes (De Kegel) identified by *De Kegel and Ryan, 2019* as broadly essential genes. Negatively selected genes discussed in detail in the present work are highlighted in dark green background.

• Supplementary file 4. Comparison of the list of genes in dataset CG$^{2SD}$_rSMN > 0.5 with the list of genes in dataset CG_SO*$^{2SD}$_rSMN > 1.50.

• Supplementary file 5. SO (Substitution Only) and SSI (Substitutions and Subtle Indel) analyses of somatic mutations of transcripts of human protein coding genes that have at least 100 confirmed somatic, non-polymorphic mutations identified in tumor tissues. The table also contains lists of passenger genes (PG_SO$^{f\_1SD}$, PG_SO$^{r2\_1SD}$, PG_SO$^{r3\_1SD}$, PG_SSI$^{f\_1SD}$, PG_SSI$^{r2\_1SD}$, PG_SSI$^{r3\_1SD}$) whose parameters deviate from the mean values by ≤1 SD as well as lists of candidate cancer genes (CG_SO$^{f\_1SD}$, CG_SO$^{r2\_1SD}$, CG_SO$^{r3\_1SD}$, CG_SSI$^{f\_1SD}$, CG_SSI$^{r2\_1SD}$, CG_SSI$^{r3\_1SD}$) whose parameters deviate from the mean values by >1 SD. Table also contains lists of candidate cancer genes (CG_SO$^{f\_2SD}$, CG_SO$^{r2\_2SD}$, CG_SO$^{r3\_2SD}$, CG_SSI$^{f\_2SD}$, CG_SSI$^{r2\_2SD}$, CG_SSI$^{r3\_2SD}$) whose parameters deviate from the mean values by >2 SD as well as lists of passenger genes (PG_SO$^{f\_2SD}$, PG_SO$^{r2\_2SD}$, PG_SO$^{r3\_2SD}$, PG_SSI$^{f\_2SD}$, PG_SSI$^{r2\_2SD}$, PG_SSI$^{r3\_2SD}$) whose parameters deviate from the mean values by <2 SD. Transcripts of OGs (oncogenes) and TSGs (tumor suppressor genes) of the cancer gene list of *Vogelstein et al., 2013* are highlighted by brick red and blue backgrounds, respectively. Transcripts of CGC (Cancer Gene Census) genes (*Sondka et al., 2018*) that do not correspond to OGs or TSGs of the cancer gene list of *Vogelstein et al., 2013* are highlighted by yellow background.

• Supplementary file 6. Numbers and fractions of missense, nonsense, and silent single-nucleotide polymorphisms (SNPs) affecting the coding sequences of the human genes. Transcripts of OGs (oncogenes) and TSGs (tumor suppressor genes) of the cancer gene list of *Vogelstein et al., 2013* are highlighted by brick red and blue backgrounds, respectively. Transcripts of CGC genes (SON, *Sondka et al., 2018*) that do not correspond to OGs or TSGs of the cancer gene list of *Vogelstein et al., 2013* are highlighted by yellow background. Novel positively or negatively selected cancer genes validated in the present work are highlighted in dark green background.

• Supplementary file 7. Comparison of fS, rSM, and rSMN scores of genes determined for somatic mutations in tumors with those determined for germline mutations.

• Supplementary file 8. Statistics of transcripts and subtle somatic mutations of human protein coding genes of the different datasets analyzed.

• Supplementary file 9. SO (Substitution Only) and SSI (Substitutions and Subtle Indel) analyses of somatic mutations of transcripts of human protein coding genes. Transcripts of OGs (oncogenes) and TSGs (tumor suppressor genes) of the cancer gene list of *Vogelstein et al., 2013* are highlighted by brick red and blue backgrounds, respectively. Transcripts of CGC (Cancer Gene Census) genes (*Sondka et al., 2018*) that do not correspond to OGs or TSGs of the cancer gene list of *Vogelstein et al., 2013* are highlighted by yellow background.

• Supplementary file 10. Contribution of major types of tumors ('Tumor Primary site') to subtle somatic substitutions of the human protein coding genes analyzed.

• Supplementary file 11. Analyses of fS, fM, and fN parameters of transcripts of human protein coding genes that have at least 0 (N0), 50 (N50), 100 (N100), or 500 (N500) somatic substitutions in tumors, respectively. Transcripts of OGs (oncogenes) and TSGs (tumor suppressor genes) of the cancer gene list of *Vogelstein et al., 2013* are highlighted by brick red and light blue backgrounds, respectively. Transcripts of CGC (Cancer Gene Census) genes (*Sondka et al., 2018*) that do not correspond to OGs or TSGs of the cancer gene list of *Vogelstein et al., 2013* are highlighted by yellow background. Novel proto-oncogenes, TSGs and negatively selected tumor essential genes validated in the present work are shown in brown, dark blue, and green colors, respectively. For 3D representations of the data, see *Figure 12*.

• Supplementary file 12. Analyses of fS, fM, and fN parameters of transcripts of human protein coding genes that have at least 0 (N0$^{2SD}$), 50 (N50$^{2SD}$), 100 (N100$^{2SD}$), or 500 (N500$^{2SD}$) somatic substitutions in tumors and deviate from average values of fS, fM, and fN by more than 2SD (Sheet 'CG_SOf_2SD'). Transcripts of OGs (oncogenes) and TSGs (tumor suppressor genes) of the cancer gene list of *Vogelstein et al., 2013* are highlighted by brick red and light blue backgrounds, respectively. Transcripts of CGC (Cancer Gene Census) genes (SON, *Sondka et al., 2018*) that do not correspond to OGs or TSGs of the cancer gene list of *Vogelstein et al., 2013* are highlighted by yellow background. Novel proto-oncogenes, TSGs and negatively selected tumor essential genes (TEGs) validated in the present work are shown in brown, dark blue, and green colors, respectively. Sheet 'statistics' contains a summary of the fS, fM, and fN parameters of datasets N0, N50, N100, N500, N0$^{2SD}$, N50$^{2SD}$, N100$^{2SD}$, N500$^{2SD}$ and indicates the number of known and novel OGs, TSGs and TEGs that are present in the different datasets.

• Supplementary file 13. Negatively selected genes in datasets N0, N50, N100, and N500. Sheet 'SO' lists the genes/transcripts in datasets N0, N50, N100, and N500 that contain transcripts of human protein coding genes with at least 0, 50, 100, or 500 somatic substitutions in tumors, respectively. The lists of negatively selected genes identified by others were taken from the publications of *Weghorn and Sunyaev, 2017*, *Zapata et al., 2018*, *Zhou et al., 2017* and *Pyatnitskiy et al., 2015*. Sheet 'statistics' indicates the number of negatively selected genes identified by others that are present in the N0, N50, N100, and N500 datasets. Note that only 48%, 64%, 77%, and 89% of the negatively selected genes identified by *Weghorn and Sunyaev, 2017*, *Zapata et al., 2018*, *Zhou et al., 2017* and *Pyatnitskiy et al., 2015*, respectively, are present in the dataset N100 that we have analyzed in the present work.

• Supplementary file 14. Expected fractions of nonsense, missense, and silent substitutions of various codons in the absence of selection assuming that there is no difference in the probability of the substitution classes C>A, C>G, C>T, T>A, T>C, and T>G.

• Supplementary file 15. Expected fractions of nonsense, missense, and silent substitutions of various codons in the absence of selection assuming that there is no difference in the probability of the substitution classes C>A, C>G, C>T, T>A, T>C, and T>G.

• Supplementary file 16. Expected fractions of nonsense, missense, and silent substitutions of various codons in the absence of selection assuming that there is no difference in the probability of the substitution classes C>A, C>G, C>T, T>A, T>C, and T>G.

• Supplementary file 17. Expected fraction of silent, missense, and nonsense mutations of coding sequences of human protein-coding genes, assuming equal probability of different substitutions classes.

- Supplementary file 18. Expected fractions of nonsense, missense, and silent substitutions of various codons in the absence of selection assuming that only C>A and G>T mutations occur.

- Supplementary file 19. Expected fractions of nonsense, missense, and silent substitutions of various codons in the absence of selection assuming that only C>G and G>C mutations occur.

- Supplementary file 20. Expected fractions of nonsense, missense, and silent substitutions of various codons in the absence of selection assuming that only C>T and G>A mutations occur.

- Supplementary file 21. Expected fractions of nonsense, missense, and silent substitutions of various codons in the absence of selection assuming that only T>A and A>T mutations occur.

- Supplementary file 22. Expected fractions of nonsense, missense, and silent substitutions of various codons in the absence of selection assuming that only T>C and A>G mutations occur.

- Supplementary file 23. Expected fractions of nonsense, missense, and silent substitutions of various codons in the absence of selection assuming that only T>G and A>C mutations occur.

- Supplementary file 24. Expected fractions of nonsense, missense and silent substitutions of various codons in the absence of selection assuming that only C>A or C>G or C>T or T>A or T>C or T>G mutations occur.

- Supplementary file 25. Expected fractions of nonsense, missense, and silent substitutions in the absence of selection assuming equal codon frequency and that only C>A or C>G or C>T or T>A or T>C or T>G mutations occur.

- Supplementary file 26. Contributions of C>A, C>G, C>T, T>A, T>C, and T>G mutations to the pattern of Single Base Substitutions in tumors.

- Supplementary file 27. Expected fractions of nonsense (fN*), missense (fM*), and silent (fS*) mutations of human protein-coding genes taking into account the probability of different substitutions classes in tumors.

- Supplementary file 28. Expected fractions of nonsense (fN**), missense (fM**), and silent (fS**) mutations of human protein-coding genes taking into account the probability of different substitutions classes in germline cells.

- Supplementary file 29. Statistics of the results of SO (Substitution Only) and SSI (Substitutions and Subtle Indel) analyses of the data presented in *Supplementary file 5*. The column marked 'Expected' indicates the parameters expected if we assume that the structure of the genetic code determines the probability of somatic substitutions.

- Supplementary file 30. Comparison of the results of SO (Substitution Only) and SSI (Substitutions and Subtle Indel) analyses.

- Supplementary file 31. Lists of genes (CG_SO^f_2SD, CG_SO^r2_2SD, CG_SO^r3_2SD, CG_SSI^f_2SD, CG_SSI^r2_2SD, CG_SSI^r3_2SD) whose parameters deviate from the mean values by >2 SD. Transcripts of OGs (oncogenes) and TSGs (tumor suppressor genes) of the cancer gene list of *Vogelstein et al., 2013* are highlighted by brick red and blue backgrounds, respectively. Transcripts of CGC (Cancer Gene Census) genes (*Sondka et al., 2018*) that do not correspond to OGs or TSGs of the cancer gene list of *Vogelstein et al., 2013* are highlighted by yellow background.

- Supplementary file 32. Observed/expected parameters (rN*, rM*, rS*; rSM*, rNM*, rNS*; rSMN*, rMSN*, and rNSM*) of somatic mutations affecting the coding sequences of the human genes in cancer. Transcripts of OGs (oncogenes) and TSGs (tumor suppressor genes) of the cancer gene list of *Vogelstein et al., 2013* are highlighted by brick red and blue backgrounds, respectively.

- Supplementary file 33. Observed/expected parameters (rN**, rM**, rS**; rSM**, rNM**, rNS**; rSMN**, rMSN**, and rNSM**) of single-nucleotide polymorphisms (SNPs) affecting the coding sequences of the human genes. Transcripts of OGs (oncogenes) and TSGs (tumor suppressor genes) of the cancer gene list of *Vogelstein et al., 2013* are highlighted by brick red and blue backgrounds, respectively.

- Transparent reporting form

Data availability

All data generated or analysed during this study are included in the manuscript and supporting files.

The following datasets were generated:

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

# Appendix 1

## Examples of genes with strong signatures of positive and/or negative selection

The assignments of the genes to key cellular processes of carcinogenesis are summarized in *Table 1* of the main text.

## Novel cancer genes positively selected for truncating mutations

### Beta-1,3-galactosyltransferase 1, encoded by the *B3GALT1* gene

B3GALT1 belongs to the glycosyltransferase 31 family. It transfers galactose from UDP-alpha-D-galactose to substrates with a terminal beta-N-acetylglucosamine residue. B3GALT1 is involved in the biosynthesis of the carbohydrate moieties of glycolipids and glycoproteins.

It has been suggested that loss of the activity of B3GALT1 may play an important role in aberrant protein glycosylation and tumor progression in colorectal cancers (*Venkitachalam et al., 2016*). Although such a role would be consistent with positive selection for inactivating mutations, analysis of the distribution of nonsense mutations along the protein sequence suggests that the high rNSM value is an artefact, rather than a signature of positive selection for iactivating mutations. The high rate of nonsense substitutions vs. sense substitutions is due to the fact that the majority of sequences contain nonsense substitution at a single site (p.R199*). Since there is no reason why selection would favor nonsense mutation at a single site it seems more likely that it reflects some sort of data deposition error. It is noteworthy in this respect that all the samples containing the p.R199* mutations originate from different regions of pancreatic tumor tissue samples from a single study (*Yachida et al., 2016*).

### Bone morphogenetic protein receptor type-2, encoded by the *BMPR2* gene

Bone morphogenetic protein receptor type-2, a member of the TGF beta family of growth factor receptors. Upon ligand binding, it forms a receptor complex consisting of two type II and two type I transmembrane serine/threonine kinases and activates SMAD transcriptional regulators.

There is convincing evidence in the literature that BMPR2 is a tumor suppressor. The *BMPR2* gene has been shown to contain several somatic frameshift mutations and to be inactivated in gastric and colorectal cancers with microsatellite instability (*Kodach et al., 2008*; *Park et al., 2010*). Loss of BMPR2 function has been found to result in increased tumorigenicity in human prostate cancer cells (*Kim et al., 2004*). More recent studies have shown that disruption of BMPR2 expression promotes mammary carcinoma metastases (*Owens et al., 2012*; *Pickup et al., 2015*). It was shown that loss of BMPR2 results in increased chemokine expression, which facilitates inflammation by a sustained increase in myeloid cells. The chemokines increased in *BMPR2* deleted cells correlated with poor outcome in human breast cancer patients, suggesting that BMPR2 has tumor suppressive functions in the stroma by regulating inflammation (*Pickup et al., 2015*).

### Bromodomain-containing protein 7, encoded by the *BRD7* gene

BRD7 is a crucial component of both functional p53 and BRCA1 pathways and recent studies have fully established BRD7 as a tumor suppressor. The expression of BRD7 was shown to be downregulated in various cancers, including breast cancer, nasopharyngeal carcinoma, gastric cancer, colorectal carcinoma, ovarian cancer, lung adenocarcinoma, NSCLC, hepatocellular carcinoma, and prostate cancer. Moreover, BRD7 inhibited cancer cell growth and metastasis and promoted apoptosis in vitro and in vivo (*Yu et al., 2016*; *Gao et al., 2016*; *Chen et al., 2016*; *Li et al., 2015*).

Recent studies suggest that BRD7 exerts it tumor suppressive role through multiple pathways, by suppressing cell proliferation, initiating cell apoptosis, and reducing aerobic glycolysis (*Niu et al., 2018*). These studies suggest that BRD7 inhibits the Warburg effect through inactivation of the HIF1α/LDHA axis.

## Inhibitor of growth protein 1 encoded by the *ING1* gene

*ING1* encodes a nuclear, cell cycle-regulated protein, overexpression of which efficiently blocks cell growth and is capable of inducing apoptosis in different experimental systems (*Toyama et al., 1999*). ING1 is known to cooperate with p53/TP53 in the negative regulatory pathway of cell growth by modulating p53-dependent transcriptional activation.

The tumor suppressor status of *ING1* has been fully established since several studies have described the loss of ING1 protein expression in human tumors and *ING1* knockout mice were reported to have spontaneously developed tumors, B cell lymphomas, and soft tissue sarcomas (*Guérillon et al., 2013*).

ING1 levels were found to be lower in breast tumors compared to adjacent normal breast tissue (*Thakur et al., 2014*). Decreasing levels of ING1 increased, and increasing levels decreased migration and invasion of cancer cells in vitro. ING1 overexpression also blocked cancer cell metastasis in vivo and eliminated tumor-induced mortality in mouse models.

ING1 can inhibit the growth of lung cancer cell lines through the induction of cell cycle arrest and apoptosis by forming a complex with p53 (*Luo et al., 2011*; *Bose et al., 2014*)

Genetic alterations that abrogate the normal function of *ING1* may contribute to esophageal squamous cell carcinogenesis (*Chen et al., 2001*). Mutations of the *ING1* tumor suppressor gene (TSG) detected in human melanoma abrogate nucleotide excision repair activity of the protein (*Campos et al., 2004*). Nonsense mutations cluster in the region of residues 339–378. These mutations eliminate the Zn finger domain and polybasic region, which are involved in interaction with histone H3 trimethylated at Lys4 (H3K4me3). It is noteworthy thar histone H3K4me3 binding is required for the DNA repair and apoptotic activities of the ING1 tumor suppressor (*Peña et al., 2008*).

## MAX gene-associated protein, encoded by the *MGA* gene

MGA functions as a dual-specificity transcription factor, regulating the expression of both MAX-network and T-box family target genes. Suppresses transcriptional activation by MYC and inhibits MYC-dependent cell transformation. Recurrent inactivation of *MGA*, a suppressor of MYC, has been shown to occur in lymphocytic leukemia, and in both NSCLC and small cell lung cancer, colorectal cancer (*De Paoli et al., 2013*; *Romero et al., 2014*; *Jo et al., 2016*).

## Proline-rich transmembrane protein 2, encoded by the *PRRT2* gene

PPRT2, as a component of the outer core of AMPAR complex, is involved in ion channel functions. *PRRT2* has been shown to be significantly downregulated in glioblastoma tissues compared with normal brain tissue (*Bi et al., 2017*; *Li et al., 2018b*). Overexpression of PRRT2 strongly impaired the cell viability and promoted cell apoptosis. These antitumor effects indicate that PRRT2 acts as a tumor suppressor in glioma. PRRT2 has been shown to have an inhibitory effect on proliferation, consistent with the low expression level of PRRT2 in cancer versus normal samples (*Alves et al., 2017*).

## Ras GTPase-activating protein 1, encoded by the *RASA1* gene

RASA1 is an inhibitory regulator of the Ras-cyclic AMP pathway. Consistent with the tumor suppressor role of RASA1, the circular RNA circ-ITCH was shown to suppress ovarian carcinoma progression through targeting miR-145/*RASA1* signaling, by increasing the level of RASA1 (*Hu et al., 2018a*).

There is evidence that *RASA1* is a potent TSG that is frequently downregulated or inactivated in several human cancer types. RASA1 expression is frequently reduced in breast cancer tissues, and the reduced RASA1 expression is associated with breast cancer progression and poor survival and disease-free survival of patients (*Liu et al., 2015*).

In hepatocellular carcinoma patients, low level of RASA1 expression correlated with a significantly poorer survival compared to those with high level of RASA1 expression, suggesting that RASA1 could serve as an independent prognostic marker for hepatocellular carcinoma patients (*Chen et al., 2017a*).

Analyses of melanoma whole-genome sequencing data have led to the identification of two novel, clustered somatic missense mutations (Y472H and L481F) in RASA1 (*Sung et al., 2016*). Unlike

wild-type RASA1, these mutants, do not suppresses soft agar colony formation and tumor growth of melanoma cell lines. In addition to mutations, loss of RASA1 expression was frequently observed in metastatic melanoma samples and a low level of RASA1 mRNA expression was associated with decreased overall survival in melanoma patients. Thus, these data support that RASA1 is inactivated by mutations or by suppressed expression in melanoma and that RASA1 plays a tumor suppressive role.

The tumor suppressor role of RASA1 is also supported by the fact that knockdown or miR targeting of *RASA1* significantly enhanced invasion and migration of multiple pancreatic cancer cells (*Sun et al., 2015*; *Kent et al., 2016*).

## E3 ubiquitin-protein ligase RNF128, encoded by the *RNF128* gene

E3 ubiquitin-protein ligase RNF128 catalyzes 'Lys-48'- and 'Lys-63'-linked polyubiquitin chains formation. Consistent with its suggested tumor suppressor role, downregulation of *RNF128* was found to predict poor prognosis in patients with urothelial carcinoma and urinary bladder. Downregulation of *RNF128* was correlated with cancer invasiveness and metastasis as well as reduced survival in patients (*Lee et al., 2016*). *RNF128* downregulation was also shown to correlate with the malignant phenotype of melanoma (*Wei et al., 2019*).

## Monocarboxylate transporter 1, MCT1 encoded by the *SLC16A1* gene

SLC16A1 is a multipass plasma membrane protein that functions as a proton-coupled monocarboxylate transporter. It catalyzes the rapid transport across the plasma membrane of many monocarboxylates such as lactate. Depending on the tissue and on cicumstances, mediates the import or export of lactic acid. Deficiency of this lactate transporter may result in an acidic intracellular environment created by muscle activity with consequent degeneration of muscle.

Although the high values of rNSM would suggest a tumor suppressor role for *SLC16A1*, several studies suggest that the protein may serve a pro-oncogenic role. For example, depletion of *SLC16A1* was found to decrease cellular proliferation and invasion in both neuroblastoma and malignant cutaneous melanoma cell lines, suggesting its role as an oncogene (OG; *Avitabile et al., 2020*). The pro-oncogenic role of MCT1 is also supported by the results of studies on esophageal squamous cell carcinoma (ESCC). Kaplan-Meier survival analysis of ESCC patients in a high-MCT1 group had a lower overall survival and lower progression-free survival, whereas downregulation of MCT1 suppressed proliferation and survival of ESCC cells in vitro (*Chen et al., 2019*). Disrupting MCT1 function leads to an accumulation of intracellular lactate that rapidly disables tumor cell growth (*Doherty et al., 2014*).

MCT1 expression is elevated in glycolytic breast tumors, and high MCT1 expression predicts poor prognosis in breast and lung cancer patients. Similarly, the observations that MCT1 inhibition impairs proliferation of glycolytic breast cancer cells co-expressing MCT1 and MCT4 and that MCT1 loss-of-function decreases breast cancer cell proliferation and blocks growth of mammary fat pad xenograft tumors suggest a pro-oncogenic or tumor essential role for MCT1 (*Hong et al., 2016*).

A recent study, however, has led to the conclusion that MCT1 and MCT4 have opposing roles in carcinogenesis (*Sukeda et al., 2019*). In a retrospective survey conducted on patients who underwent surgical resection for pancreatic ductal adenocarcinoma, the expression of MCT1, MCT4, and GLUT1 was assessed in tumor cells and cancer-associated fibroblasts (CAFs) and the impact of their expression on patient outcome was also analyzed. In tumor cells, MCT1 expression was associated with extended overall and progression-free survival and decreased nodal metastasis. Conversely, MCT4 expression in CAFs was associated with shortened survival. In other words, in tumor cells, MCT1 expression is associated with better prognosis and reduced nodal metastasis in pancreatic cancer, contrary to findings of previous studies.

It is noteworthy in this respcect that based on the pattern of mutations SLC16A1/MCT1 appers to be a tumor suppressor rather than a tumor essential gene (TEG) in as much as it has a high proportion of truncating mutations. It seems possible that glycolytic tumor cells that must get rid of lactate are selected for increased efflux and decreased influx of lactate and this might be achieved by increased expression of MCT4 and decreased activity of MCT1.

## Sprouty-related, EVH1 domain-containing protein 1, encoded by the *SPRED1* gene

The *SPRED1* gene, which encodes a negative regulator of mitogen-activated protein kinase (MAPK) signaling, has been shown to function as a TSG in several types of cancer (*Pasmant et al., 2015*; *Ablain et al., 2018*; *Sun et al., 2019*).

## Homeobox protein TGIF1, encoded by the *TGIF1* gene

TGIF binds to a retinoid X receptor (RXR)-responsive element from the cellular retinol-binding protein II promoter (CRBPII-RXRE). Inhibits the 9-cis-retinoic acid-dependent RXR alpha transcription activation of the retinoic-acid-responsive element. Active transcriptional corepressor of SMAD2.

There is evidence that TGIF1 may function as a tumor suppressor. In pancreatic ductal adenocarcinoma genetic inactivation of *TGIF1* in the context of oncogenic KRASG12D, culminated in the development of highly aggressive and metastatic pancreatic ductal adenocarcinoma (*Parajuli et al., 2019*; *Weng et al., 2019*). These authors have found that TGIF1 associates with TWIST1 and inhibits TWIST1 expression and activity, and this function is suppressed in the vast majority of human pancreatic ductal adenocarcinoma by KRASG12D /MAPK-mediated TGIF1 phosphorylation. Ablation of TWIST1 in KRASG12D;TGIF1KO mice blocked pancreatic ductal adenocarcinoma formation, providing evidence that TGIF1 restrains KRASG12D-driven pancreatic ductal adenocarcinoma through its ability to antagonize TWIST1.

The majority of available evidence, however, suggests that the protein plays a cancer promoting role. TGIF1 has been shown to promote the growth and migration of cancer cells in NSCLC (*Xiang et al., 2015*). The authors have shown that expression of TGIF1 is elevated in NSCLC tissues, that TGIF1 promoted the growth and migration of cancer cells and that knocking down the expression of *TGIF1* inhibited the growth and migration of NSCLC cells. These studies have also shown that TGIF1 exerted its oncogenic role through beta-catenin/TCF signaling.

Studies on triple negative breast cancer have revealed that high levels of TGIF expression correlate with poor prognosis since TGIF promotes Wnt-driven mammary tumorigenesis. As to the molecular mechanism of the oncogenic role of TGIF: it has been shown that TGIF interacts with and sequesters Axin1 and Axin2 into the nucleus, disassembles the β-catenin-destruction complex leading to the accumulation of β-catenin that activates expression of Wnt target genes (*Zhang et al., 2015*; *Razzaque and Atfi, 2016*).

In harmony with an oncogenic role of TGIF in breast cancer, silencing of *TGIF* was found to suppress the migration, invasion and metastasis of the human breast cancer cells in both in vitro and in vivo experiments (*Wang et al., 2018c*).

*TGIF1* has also been found to be significantly upregulated in some colorectal cancers and to promote adenoma growth in the context of mutant Apc (*Shah et al., 2019*). Overexpression of TGIF1 markedly promoted the proliferation of colorectal cancer cells through the activation of Wnt/β-catenin signaling (*Wang et al., 2017c*).

In summary, the majority of data suggest that *TGIF1* may act as an OG, despite the fact that the high proportion of truncating indel mutations would indicate a tumor suppressor function. Since the transcription regulator TGIF1 may play both pro-oncogenic and tumor suppressor functions (in different cellular processes), our observation that during tumor evolution selection for truncating mutations appears to dominate for TGIF1 suggests that the selection pressure to eliminate the tumor suppressor activity may override the pressure to preserve it oncogenic activities.

## Trinucleotide repeat-containing gene 6B protein, encoded by the *TNRC6B* gene

*TNRC6B* is a key miRNA-processing gene that plays a role in RNA-mediated gene silencing by both micro-RNAs (miRNAs) and short interfering RNAs (siRNAs). TNRC6B is required for miRNA-dependent translational repression and siRNA-dependent endonucleolytic cleavage of complementary mRNAs by argonaute family proteins.

Genomic analysis of liver cancer have identified *TNRC6B* as a significantly mutated gene, suggesting that it may be an important driver gene (*Li et al., 2018a*). Consistent with its putative tumor

suppressor role, DNA methylation of *TNRC6B* has been suggested to play a role in early carcinogenesis (*Joyce et al., 2018*).

## Dual specificity protein kinase TTK, encoded by the *TTK* gene

TTK, capable of phosphorylating serine, threonine, and tyrosine residues of proteins, plays a role in cell proliferation. Although, intuitively the high rate of truncating mutations would suggest a tumor suppressor role for TTK, all the available evidence indicates that it acts as an OG.

It has been shown that dual specificity kinase TTK is strongly overexpressed in human pancreatic ductal adenocarcinoma, suggesting a cancer promoting role. In harmony with such a role, following *TTK* knockdown cell proliferation was significantly attenuated, whereas apoptosis and necrosis rates were significantly increased. Apoptosis was associated with increased formation of micronuclei, suggesting that loss of TTK results in chromosomal instability and mitotic catastrophe (*Kaistha et al., 2014*).

Levels of TTK protein were also found to be significantly elevated in neoplastic tissues of liver cancer patients, when compared with adjacent hepatic tissues. In an experimental animal model, it was shown that in vitro knockdown of *TTK* effectively blocks intrahepatic growth of human hepatic carcinoma cell xenografts, suggesting that targeted TTK inhibition might have clinical utility in the therapy of liver cancer (*Miao et al., 2016*).

In a recent study, dual specificity protein kinase TTK has been identified as the most upregulated and differentially expressed kinase in glioma stem-like cells that are responsible for tumorigenesis and subsequent tumor recurrence in glioblastoma. TTK expression was highly enriched in glioblastoma and was inversely correlated with a poor prognosis (*Wang et al., 2018a*).

The deubiquitinase, USP9X has been implicated in multiple cancers and its oncogenic effects were shown to be exerted at least in part through dual specificity protein kinase TTK (*Chen et al., 2018d*). USP9X was found to stabilize TTK by efficient deubiquitination of the kinase; levels of USP9X and TTK were significantly elevated and positively correlated in tumor tissues, suggesting that the USP9X-TTK axis plays a critical role in carcinogenesis. In harmony with the synergism of these OGs, knockdown of *USP9X* or *TTK* inhibited cell proliferation, migration, and tumorigenesis.

The explanation for the apparent contradiction of the oncogenic role of TTK and the abundance of truncating mutations in the protein probably lies in the fact that – unlike in the case of typical TSGs – mutations are not randomly distributed along the protein sequence. The truncating mutations are practically restricted to the very C-terminal end of the protein (EKKRGKK, residues 851–857), downstream of the catalytic domain and missense mutations also cluster in this C-terminal end. It seems likely that this region is involved in some negative control of the activity of TTK and missense and truncating mutations liberate TTK from this negative control. It is unclear at present whether the mutations affecting this C-terminal motif activate the TTK proto-oncogene by interfering with its ubiquitination or by affecting its subcellular localization.

## Zinc finger CCCH domain-containing protein 13, encoded by the *ZC3H13* gene

ZC3H13 is associated with a complex that mediates N6-methyladenosine (m6A) methylation of RNAs, a modification that plays a role in the efficiency of mRNA splicing and RNA processing. It acts as a key regulator of m6A methylation by promoting m6A methylation of mRNAs at the 3′-UTR. ZC3H13 has been shown to serve as a tumor suppressor in colorectal cancer (*Zhu et al., 2019*).

## mRNA decay activator protein ZFP36L2, encoded by the *ZFP36L2* gene

*ZFP36L2* has been selected as a gene characterized by very high values of indel_rNSM, suggesting positive selection for truncating mutations. Although the closely related *ZFP36L1* gene is not present in the lists defined by the CG_SO and CG_SSI lists defined by the 2SD cut-off values, it is also characterized by very high values of rNSM (*Supplementary file 5*).

ZFP36L1 and ZFP36L2 zinc-finger RNA-binding proteins destabilize several cytoplasmic AU-rich element (ARE)-containing mRNA transcripts by promoting their poly(A) tail removal or deadenylation, and hence provide a mechanism for attenuating protein synthesis. The proteins are necessary

for thymocyte development and prevention of T-cell acute lymphoblastic leukemia transformation by promoting ARE-mediated mRNA decay of the mRNA of oncogenic factors.

Deletion of the genes *ZFP36L1* and *ZFP36L2* leads to perturbed thymic development and T lymphoblastic leukemia (*Hodson et al., 2010*).

ZFP36L1 and ZFP36L2 play a negative role in cell proliferation. Forced expression of ZFP36L1 or ZFP36L2 inhibited cell proliferation in colorectal cancer cell lines, whereas knockdown of these genes increased cell proliferation (*Suk et al., 2018*). ZFP36L2 has been validated as an important tumor-suppressor specific to oesophageal squamous cell carcinomas (*Lin et al., 2018*).

## Zinc finger protein 276, encoded by the *ZNF276* gene

Zinc finger protein is involved in transcriptional regulation.

It has been suggested that ZNF276 may be a tumor suppressor in breast cancer progression in colorectal cancers (*Wong et al., 2003*).

Although such a role would be consistent with positive selection for inactivating mutations, analysis of the distribution of nonsense mutations along the protein sequence suggests that the high rNSM value is an artefact, rather than a signature of positive selection for inactivating mutations. The high rate of nonsense substitutions vs. sense substitutions is due to the fact that the majority of sequences contain nonsense substitution at a single site (p.Q217*). Since there is no reason why selection would favor nonsense mutation at a single site, it seems more likely that it reflects some sort of data deposition error. It is noteworthy in this respect that all the samples containing the p.Q217* mutations originate from different regions of pancreatic tumor tissue samples from a single study (*Yachida et al., 2016*).

## Zinc finger protein 750, encoded by the *ZNF750* Gene

Zinc finger protein 750 is a transcription factor required for terminal epidermal differentiation, it acts downstream of p63/TP63. Its mutations have been shown to abolish the ability to induce epidermal terminal differentiation. In harmony with its mutation pattern, numerous studies suggest a tumor suppressor role for ZNF750.

Analysis of cancer genes across 21 tumor types identified *ZNF750* as a gene harboring many early frameshift and nonsense mutations in head and neck cancer and as the only known gene residing in a small current focal deletion in head and neck and lung squamous cancers (*Lawrence et al., 2014*). *ZNF750* has also been identified as a tumor suppressor in oral and esophageal squamous cell carcinoma (*Yang et al., 2017*; *Nambara et al., 2017*; *Hazawa et al., 2017*; *Otsuka et al., 2018*). Studies on the clonal evolution in esophageal squamous cell carcinoma revealed that the majority of driver mutations in this cancer occurred in the tumor-suppressor genes, including *TP53*, *KMT2D*, and *ZNF750* (*Hao et al., 2016*).

## Novel cancer genes positively selected for missense mutations

### Aurora kinase A encoded by the *AURKA* gene

AURKA, also known as a Breast tumor-amplified kinase, is a mitotic serine/threonine kinase that contributes to the regulation of cell cycle progression. It associates with the centrosome and the spindle microtubules during mitosis and plays a critical role in various mitotic events.

In harmony with the notion that AURKA's mutation pattern reflects a pro-oncogenic role for the protein, elevated expression of AURKA has been shown to induce oncogenic phenotypes (*Takahashi et al., 2015*; *Treekitkarnmongkol et al., 2016*).

Similarly, the observation that downregulation, inhibition or depletion of AURKA reduced viability and invasiveness of cancer cells (*Sillars-Hardebol et al., 2012*; *Li et al., 2018a*; *van Gijn et al., 2019*) also argues for an oncogenic role of the protein.

Significantly, specific knockdown of *AURKA* in cultured pancreatic cancer cells strongly suppressed in vitro cell growth and in vivo tumorigenicity (*Hata et al., 2005*). Recently, a novel AURKA mutation (V352I) was identified from clinical specimens and it was shown that AURKA (V352I)-induced carcinogenesis was earlier and much more severe than wild-type AURKA, implying that the V352I mutation may accelerate cancer progression (*Su et al., 2019*).

Although many *AURKA* mutations were identified in cancer patients, it is noteworthy that there is no evidence for the clustering or 'recurrence' of mutations. The most likely explanation for the lack of clustering of mutations is that since AURKA interacts with numerous proteins (e.g. PIFO, GADD45A, AUNIP, NIN, FRY, SIRT2, MYCN, HNRNPU, AAAS, KLHL18, CUL3, FOXP1) there may be multiple sites where missense mutations affecting these interactions may result in disregulation of the activity of **AURKA**.

In summary, although all the available experimental information argues for an oncogenic role of *AURKA*, there was no evidence for the clustering of its missense mutations. In our view, this observation illustrates that recurrence of missense mutations is not a *sine qua non* criterion of OGs.

Recent studies have also revealed that AURKA and TWIST1 are linked in a feedback loop controlling tumorigenesis and metastasis. AURKA phosphorylates TWIST1, inhibits its ubiquitylation, increases its transcriptional activity and favors its homodimerization. TWIST1 prevents AURKA degradation, thereby triggering a feedback loop. Ablation of either AURKA or TWIST1 completely inhibits epithelial-to-mesenchymal transition, suggesting that inhibition of AURKA and TWIST1 are synergistic in inhibiting tumorigenesis and metastasis (*Wang et al., 2017b*).

Although the *TWIST1* gene is not present in the datasets (*Supplementary files 5* and *31*) that contain the metadata for transcripts containing at least 100 confirmed somatic, non-polymorphic mutations identified in tumor tisses, inspection of the primary dataset (*Supplementary file 9*) indicates that it is characterized by very high value of rSMN (*Supplementary file 5*), indicating strong signature of purifying selection (see section on Negatively selected genes) consistent with the view that – in synergism with AURKA – it plays an important role in promoting tumorigenesis.

## Cyclin-dependent kinase 8, encoded by the *CDK8* gene

The *CDK8* gene is a coactivator involved in regulated gene transcription of nearly all RNA polymerase II-dependent genes.

*CDK8* is a colorectal cancer OG that regulates beta-catenin activity. Suppression of CDK8 expression inhibits proliferation in colon cancer cells characterized by high levels of CDK8 and beta-catenin hyperactivity (*Firestein et al., 2008*). CDK8 has been shown to promote SMAD1-driven epithelial-to-mesenchymal transition through YAP1 recruitment (*Serrao et al., 2018*). There is a large body of evidence that CDK8 is a key oncogenic driver in many cancers (*Philip et al., 2018*). *CDK8* was found to be amplified or overexpressed in many colon cancers and CDK8 expression correlated with shorter patient survival (*Liang et al., 2018*).

## Isocitrate dehydrogenase [NAD] subunit beta, mitochondrial, encoded by the *IDH3B* gene

IDH3B plays an essential role in the activity of isocitrate dehydrogenase. The heterodimer composed of the alpha (IDH3A) and beta (IDH3B) subunits and the heterodimer composed of the alpha (IDH3A) and gamma (IDH3G) subunits, have significant activity but the full activity of the heterotetramer (containing two subunits of IDH3A, one of IDH3B and one of IDH3G) requires the assembly of both heterodimers.

Our Pubmed search failed to identify publications with major relevance for the role of *IDH3B* in carcinogenesis. It is noteworthy, however, that the *IDH3B* gene contains recurrent somatic missense mutations at residue R131 that is equevalent with R132 and R140 of the paralogous enzymes, IDH1 and IDH2, respectively, that are affected by recurrent oncogenic missense mutations. These mutations of IDH1 and IDH2 result in loss of normal enzymatic function and the abnormal production of 2-hydroxyglutarate. 2-Hydroxyglutarate has been found to inhibit enzymatic function of many alpha-ketoglutarate-dependent enzymes, including histone and DNA demethylases, causing widespread epigenetic changes in the genome thereby promoting tumorigenesis. It seems likely that the R131 mutations of IDH3B may conntribute to carcinogenesis by a similar mechanism.

## E3 ubiquitin-protein ligase MARCH7, encoded by the *MARCH7* gene

March7 is an E3 ubiquitin-protein ligase, an enzyme that accepts ubiquitin from an E2 ubiquitin-conjugating enzyme and then directly transfer the ubiquitin to targeted substrates.

Several studies support an oncogenic role for the ubiquitin E3 ligase MARCH7. Studies on ovarian tissues have revealed that expression of MARCH7 was higher in ovarian cancer tissues than normal ovarian tissues. Silencing *MARCH7* decreased, whereas ectopic expression of MARCH7 increased cell proliferation, migration and invasion, suggesting that MARCH7 is oncogenic and a potential target for ovarian cancer therapy (*Hu et al., 2015*). The expression of MARCH7 was significantly higher in cervical cancer tissues than normal cervical tissues, suggesting that this OG may also serves as a potential target for cervical cancer therapy (*Hu et al., 2018b*).

The expression level of MARCH7 in endometrial cancer tissues was also found to be significantly higher than that in normal endometrium tissues, suggesting that it may be an oncogenic factor in endometrial cancer (*Liu et al., 2019*). The oncogenic role of MARCH7 is supported by the fact its knockdown inhibited the invasion and metastasis of endometrial cancer cells in vitro and in vivo, whereas the opposite effect was observed after overexpressing MARCH7.

## GTP-binding protein RIT1, encoded by the *RIT1* gene

The high value of rMSN reflects primarily the recurrence of substitutions (Met90Ile, Met90Val) of Met90 of RIT1 protein.

RIT1 plays a crucial role in the activation of MAPK signaling cascades that mediate a wide variety of cellular functions, including cell proliferation, survival, and differentiation.

Since the Met90Ile substitution has been shown to result in an increased MAPK-ERK signaling (*Aoki et al., 2013*; *Koenighofer et al., 2016*), it is plausible to assume that the high rate of missense mutations reflects positive selection of oncogenic driver mutations.

In harmony with this conclusion, studies on endometrial cancer have revealed that RIT1 mRNA and protein were significantly overexpressed in endometrial cancer cell lines and in endometrial cancer tissues compared to non-cancerous endometrial tissue samples (*Xu et al., 2015*). Elevated expression of RIT1 was significantly correlated with pathological type, clinical stage. Kaplan-Meier survival analysis indicated that RIT1 expression was associated with poor overall survival of endometrial cancer patients, suggesting that elevated expression of RIT1 may contribute to the progression of endometrial cancer.

In a study of lung adenocarcinoma cases, several somatic mutations (including Met90Ile) were identified in the *RIT1* gene that were found to cluster in a hotspot near the switch II domain of the GTPase protein (*Berger et al., 2014*). Ectopic expression of these mutated *RIT1* genes was found to induce cellular transformation in vitro and in vivo, confirming that these substitutions are driver mutations and that *RIT1* is an OG in lung adenocarcinoma.

## Yes-associated protein 1, encoded by the *YAP1* gene

Yes-associated protein one is known to be the critical downstream regulatory target in the Hippo signaling pathway that plays a pivotal role in tumor suppression by restricting proliferation and promoting apoptosis. This pathway is composed of a kinase cascade that eventually inactivates YAP1 since phosphorylation of YAP1 by the tumor suppressors LATS1/2 inhibits its translocation into the nucleus.

Several lines of evidence indicate that YAP1 is an OG. *YAP1* was found to act as oncogenic target of 11q22 amplification in multiple cancer subtypes, whereas *YAP1* silencing significantly decreases cell proliferation (*Lorenzetto et al., 2014*; *Hamanaka et al., 2019*). *YAP1* was shown to promote growth of prostate cancer, whereas knock down of its expression or inhibition of YAP1 function significantly suppressed tumor recurrence (*Jiang et al., 2017*). The key role of YAP1 in carcinogenesis is also supported by the fact that the tumor suppressor LATS2 inhibits the malignant behaviors of glioma cells by inactivating of YAP1 (*Shi et al., 2019*).

Although several *YAP1* mutations were identified in cancer patients, there is no evidence for the clustering or 'recurrence' of mutations. Similarly to the case of AURKA (see above), the most plausible explanation for the lack of clustering of mutations of this OG is that since YAP1 interacts with several proteins (e.g. YES kinase, LATS1, LATS2, TP73, RUNX1, WBP1, WBP2, TEAD1, TEAD2, TEAD3, TEAD4, HCK, MAPK8, MAPK9, CK1, ABL1) mutations at several different sites may affect these interactions and may result in disregulation of the activity of YAP1. In our view, the cases of

AURKA, YAP1 and YES1 illustrate that recurrence of missense mutations is not a *sine qua non* criterion of OGs.

## Tyrosine-protein kinase Yes, encoded by the *YES1* gene

Tyrosine-protein kinase Yes (also known as proto-oncogene c-Yes) is a multidomain non-receptor protein tyrosine kinase containing an SH3 domain, an SH2 domain and a protein kinase domain. YES1 is involved in the regulation of cell growth and survival, apoptosis, cell-cell adhesion, cytoskeleton remodeling, and differentiation. It plays a role in cell cycle progression by phosphorylating the cyclin-dependent kinase 4/CDK4 thus regulating the G1 phase. YES1 has been shown to phosphorylate YAP1, leading to the localization of a YAP1-TBX-β-catenin complex to the promoters of antiapoptotic genes, thereby promoting carcinogenes (*Rosenbluh et al., 2012*). A small-molecule inhibitor of YES1 impeded the proliferation of β-catenin-dependent cancers in both cell lines and animal models.

Several lines of evidence have established an oncogenic role for *YES1*.

It has been demonstrated recently that YES1 is essential for lung cancer growth and progression in NSCLC, suggesting that it is a promising therapeutic target in lung cancer. YES1 overexpression induced metastatic spread in preclinical in vivo models, whereas *YES1* genetic depletion by CRISPR/Cas9 technology significantly reduced tumor growth and metastasis (*Garmendia et al., 2019*).

In harmony with an oncogenic role of *YES1*, several microRNAs have been shown to inhibit the proliferation of tumor cells by targeting *YES1* (*Tan et al., 2015*; *Shen et al., 2019*; *Zhao et al., 2020*).

The oncogenic role of YES1 in cancer is also supported by the observation that it is amplified in several types of cancer, suggesting that it could be an attractive target for a cancer drug (*Fan et al., 2018*; *Hamanaka et al., 2019*). *Hamanaka et al., 2019* have generated a YES1 kinase inhibitor, and have shown that YES1 kinase inhibition by this drug led to antitumor activity against *YES1*-amplified cancers in vitro and in vivo. The authors have also shown that Yes-associated protein 1 (YAP1) played a role downstream of YES1 and contributed to the growth of *YES1*-amplified cancers. indicating that the regulation of YAP1 by YES1 plays an important role in *YES1*-amplified cancers. These findings identify YES1 as a targetable OG of significant potential for clinical utility (*Rai, 2019*).

Although *YES1* contains an increased proportion of nonsynonymous mutations there is no evidence for the clustering or 'recurrence' of mutations. Similarly to the cases of *AURKA* and *YAP1* (see above), the most plausible explanation for the lack of clustering of mutations of this OG is that since YES1 is a multidomain protein that interacts with several proteins, mutations at several different sites may affect these interactions and may result in disregulation of the activity of YAP1. In our view, the cases of AURKA, YAP1, and YES1 illustrate that recurrence of missense mutations is not a *sine qua non* criterion of OGs.

## Negatively selected TEGs

### Atypical chemokine receptor 3, encoded by the *ACKR3* (*CXCR7*) gene

ACKR3 is a member of the group of chemokine receptors that acts as a receptor for chemokines CXCL11 and CXCL12/SDF1. It is activated by CXCL11 in malignant hemapoietic cells, leading to phosphorylation of ERK1/2 (MAPK3/MAPK1) and enhanced cell adhesion and migration.

*ACKR3* is a known cancer gene, from Tier 1 of the Cancer Gene Census; it has a cancer hallmark annotation. Its importance in carcinogenesis is underlined by the fact that high expression of ACKR3 is associated with poor survival in several types of cancer.

As to the role of ACKR3 in hallmarks of cancer: it has been suggested that ACKR3 promotes proliferative signaling, angiogenesis, evasion of programmed cell death and invasion and metastasis.

Several studies support the key role of ACKR3 in tumor invasion and metastasis (*Li et al., 2014*; *Stacer et al., 2016*; *Zhao et al., 2017*; *Puddinu et al., 2017*; *Melo et al., 2018*; *Qian et al., 2018*). Since knock-down or pharmacological inhibition of *ACKR3* has been shown to reduce tumor invasion and metastasis, ACKR3 is a promising therapeutic target for the control of tumor dissemination.

## CX3C chemokine receptor 1, encoded by the *CX3CR1* gene

CX3CR1 is a member of the group of chemokine receptors that play a major role in tumor metastasis. The interactions of chemokines, also known as chemotactic cytokines, with their receptors regulate immune and inflammatory responses. However, recent studies have demonstrated that cancer cells subvert the normal chemokine role, transforming them into fundamental constituents of the tumor microenvironment with tumor-promoting effects. CX3CR1 is the receptor for the CX3C chemokine fractalkine (CX3CL1) that mediates both its adhesive and migratory functions.

CX3CR1 expression has been shown to be associated with the process of cellular migration in vitro and tumor metastasis of clear cell renal cell carcinoma in vivo (*Yao et al., 2014*).

Recent studies indicate that tumor-associated macrophages MΦ can influence cancer progression and metastasis and that CCR2 and CX3CR1 play important roles in metastasis. *Schmall et al., 2015* have shown that coculturing of tumor-associated macrophages with mouse Lewis lung carcinoma caused up-regulation of CCR2/CCL2 and CX3CR1/CX3CL1 in both the cancer cells and the macrophages. In vivo, MΦ depletion and genetic ablation of *CCR2* and *CX3CR1* all inhibited LLC1 tumor growth and metastasis, and enhanced survival. Furthermore, mice treated with CCR2 antagonist mimicked genetic ablation of CCR2, showing reduced tumor growth and metastasis. These findings indicate that tumor-associated MΦ play a central role in lung cancer growth and metastasis, with bidirectional cross-talk between MΦ and cancer cells via CCR2 and CX3CR1 signaling. These studies suggest that the therapeutic strategy of blocking CCR2 and CX3CR1 may prove beneficial for halting metastasis.

CX3CR1 is highly expressed in gastric cancer tissues and is related to lymph node metastasis and larger tumor size. CX3CR1 overexpression promoted gastric cancer cell migration, invasion, proliferation, and survival (*Wei et al., 2015*).

CX3CR1 is overexpressed in human breast tumors and cancer cells utilize the chemokine receptor CX3CR1 to exit the blood circulation and metastize to the skeleton. To assess the clinical potential of targeting CX3CR1 in breast cancer *Shen et al., 2016* have used neutralizing antibody for this receptor, transcriptional suppression by CRISPR interference as well as a potent and selective small-molecule antagonist of CX3CR1 in preclinical animal models of metastasis. The authors have found that inactivation of CX3CR1 impairs the lodging of circulating tumor cells to the skeleton and impairs further growth of established metastases. These data suggest that CX3CR1 has a important role in promoting metastasis activity and that CX3CR1 antagonists may valuable as drugs of tumor therapy.

## C-C chemokine receptor type 2, encoded by the *CCR2* gene

## C-C chemokine receptor type 5, encoded by the *CCR5* gene

Although the *CCR2* gene of C-C chemokine receptor type two and the *CCR5* gene of C-C chemokine receptor type five are not present in the CG_SO and CG_SSI lists defined by the 2SD cut-off values they are also characterized by very high values of rSNM (*Supplementary file 5*), suggesting that they may also play importantant roles in tumor metastasis.

CCR2 is the key functional receptor for the chemokine ligand CCL2. Its binding with CCL2 on monocytes and macrophages mediates chemotaxis and migration induction. Recent studies indicate that CCR2 and CX3CR1 play important roles in metastasis (*Schmall et al., 2015*). The CCL2-CCR2 signaling axis has generated increasing interest in recent years due to its association with the progression of cancer. The CCL2-CCR2, signaling pair has been shown to have multiple pro-tumorigenic roles, mediating tumor growth and angiogenesis (*Lim et al., 2016*).

CCR5 serves as a receptor for a number of inflammatory CC-chemokines including CCL3/MIP-1-alpha, CCL4/MIP-1-beta. Recent studies have revealed that C-C chemokine receptor type 5 plays a key role in progression of tumorigenesis, Expression of CCR5 augments regulatory T cell differentiation and migration to sites of inflammation. The misexpression of CCR5 in epithelial cells, induced upon oncogenic transformation, hijacks this migratory phenotype (*Aldinucci and Casagrande, 2018*; *Jiao et al., 2019*).

## Dentin sialophosphoprotein, encoded by the *DSPP* gene

The *DSPP* gene has been selected as a gene showing very high values of rSMN, suggesting negative selection of missense and nonsense mutations (*Supplementary file 5*). It must be pointed out that

based on the high silent/missense ratio *DSPP* has also been identified by others as a gene showing signs of strong negative selection (*Zhou et al., 2017*).

Dentin sialophosphoprotein is a secreted protein that has been shown to play an important role in dentinogenesis. It binds high amount of calcium and facilitates initial mineralization of dentin matrix collagen as well as regulate the size and shape of the crystals, therefore it seemed surprising that its gene would qualify as a negatively selected TEG.

There is evidence in the scientific literature that the protein may have a tumorigenic role in oral cancer (*Chaplet et al., 2006*; *Joshi et al., 2010*; *Saxena et al., 2015*; *Gkouveris et al., 2018*; *Nikitakis et al., 2018*). Nevertheless, the high silent to missense rate is not a reflection of the importance of *DSPP* for carcinogenesis. The *DSPP* gene contains a 2 kb repeat domain containing over 200 tandem copies of a nominal 9-basepair (AGC AGC GAC) repeat encoding a series of tandem Ser Ser Asp repeats and the unusully high rate of silent mutations is restricted to this region of the gene.

A study of 188 normal human chromosomes revealed that the repeat domain of *DSPP* is hyper-variable with extraordinary rates of change including slip-replication indel events and predominantly C-to-T transition SNPs (*McKnight et al., 2008*). In harmony with the increased rate and predominance of C-to-T transition in the AGC AGC GAC (Ser-Ser-Asp) repeats, the vast majority of substitutions in this repeat region of the *DSSP* gene are silent. The unusually high silent to missense mutation ratio of the *DSPP* gene is thus not due to purifying selection of a TEG.

## Forkhead box protein G1, encoded by the *FOXG1*

FOXG1 is a member of the FOX (Forkhead box) protein family of transcription factors that play important roles in regulating the expression of genes involved in cell growth, proliferation, differentiation, and longevity. FOXG1 localizes to mitochondria and coordinates cell differentiation and bioenergetics (*Pancrazi et al., 2015*).

The tumor promoting role of FOXG1 is supported by the observation that childhood medulloblastomas are characterized by 2–7-fold copy gain for *FOXG1*. *FOXG1* copy gain (>2 to 21 folds) was seen in 93% of a validating set of tumors and showed a positive correlation with protein expression (*Adesina et al., 2007*).

The oncogenic role of FOXG1 is also supported by the observation that a decrease of FOXG1 in medulloblastoma cells offers a survival advantage in mice (*Adesina et al., 2015*), whereas high expression of FOXG1 was associated with poor survival of glioblastoma patients (*Robertson et al., 2015*).

The carcinogenesis promoting activity of FOXG1 is supported by the observation that endogenous FOXG1 expression levels were positively correlated to the glioblastoma multiforme disease progression (*Wang et al., 2018b*). Overexpression of FOXG1 protein resulted in increased cell viability, and it was suggested that FOXG1 functions as an onco-factor by promoting proliferation and inhibiting differentiation.

Recent studies on glioblastoma have shown that transcription factors FOXG1 and TLE1 promote glioblastoma propagation by supporting maintenance of brain tumor-initiating cells (*Dali et al., 2018*). Since the expressions of caspase family members were significantly altered in response to change of FOXG1 expression, it has been suggested that FOXG1 also contributes to carcinogenesis as a negative regulator of glioma cell apoptosis (*Chen et al., 2018a*).

## Forkhead box protein P2, encoded by *FOXP2* gene
### Forkhead box protein P2 (FOXP2) is a transcriptional repressor

The role of FOXP2 in cancer is somewhat controversial, it appears to have oncogenic or tumor suppressor roles, depending on the cellular and histological features of tumors. While FOXP2 has been found to be down-regulated in breast cancer, hepatocellular carcinoma and gastric cancer biopsies, overexpressed FOXP2 has been reported in multiple myelomas, several subtypes of lymphomas, as well as in neuroblastomas and some prostate cancers (*Herrero and Gitton, 2018*).

Numerous recent studies indicate a tumor suppressor like role for FOXP2 (*Campbell et al., 2010*; *Cuiffo et al., 2014*; *Yan et al., 2015*; *Diao et al., 2018*; *Song et al., 2017*; *Chen et al., 2018b*;

*Li et al., 2019*), others present evidence for an OG-like role of the protein (*Campbell et al., 2010*; *Zhong et al., 2017*; *Wu et al., 2018*; *Wang et al., 2019*).

The high silent to missense ratio of substitution mutations observed in the case of the *FOXP2* gene does not seem to be a reflection of purifying selection, that might be in harmony of an OG-like role, but definitely not with a tumor suppressor role.

The translated region of the *FOXP2* gene contains a long stretch of CAG repeats (residiues 177–216), corresponding to the polyQ segment of the protein. Silent mutations are clustered in the polyQ tract of the protein encoded by the imperfect polymorphic region, suggesting that the increased silent to missense rate of substitutions in this gene has much less to do with purifying selection than with microsatellite instability.

## Glucose-6-phosphate 1-dehydrogenase, encoded by the *G6PD* gene

Glucose-6-phosphate 1-dehydrogenase catalyzes the rate-limiting step of the oxidative pentose-phosphate pathway, its main function is to provide reducing power (NADPH) and pentose phosphates for fatty acid and nucleic acid synthesis. There is strong support for the importance of G6PD for tumor growth. Progression of tumor cells to more aggressive phenotypes requires not only the upregulation of glygolysis but also the pentose phosphate pathway as a provider of reducing power and ribose phosphate to the cell for maintenance of redox balance and biosynthesis of nucleotides and lipids, making G6PD a promising target in cancer therapy (*Zhang et al., 2014a*).

The key importance of G6PD for tumor growth is supported by the fact that elevated G6PD levels promote cancer progression in numerous tumor types, that high G6PD expression is a poor prognostic factor and that knockdown of G6PD suppresses cell viability and growth (*Wang et al., 2012*; *Pu et al., 2015*; *Wang et al., 2015b*; *Poulain et al., 2017*; *Chen et al., 2018c*; *Yang et al., 2018*; *Barajas et al., 2018*; *Yang et al., 2019*).

## Mitogen-activated protein kinase 13, encoded by the *MAPK13* gene

MAPK13 (p38δ mitogen-activated protein kinase) is a serine/threonine kinase which acts as an essential component of the MAP kinase signal transduction pathway. MAPK13 plays an important role in the cascades of cellular responses evoked by extracellular stimuli such as proinflammatory cytokines. The protein is involved in the regulation of epidermal keratinocyte differentiation, apoptosis, and skin tumor development.

Although MAPK13 shows signatures of negative selection that would suggest a pro-oncogenioc role for the protein, expermintal data are controversial as to its role in carcinogenesis: there is evidence for both a pro-oncogenic and tumor suppressor roles of MAPK13.

The observation that p38delta promotes cell proliferation and tumor development in epidermis suggests that it has a pro-oncogenic role (*Schindler et al., 2009*). Analyzes of the gene expression profiles have shown that MAPK13 is expressed in uterine, ovary, stomach, colon, liver, and kidney cancer tissues at higher levels compared with adjacent normal tissues. *MAPK13* gene knockdown has been shown to abrogate the tumor-initiating ability of cancer stem-like cells, indicating that the gene has a cancer-promoting role (*Yasuda et al., 2016*). The protein p38δ is highly expressed in all types of human breast cancers, whereas lack of p38δ resulted in reduced primary tumor size and blocked the metastatic potential to the lungs (*Wada et al., 2017*). The fact that mice with germline deletion of the p38δ gene are significantly protected from chemical skin carcinogenesis also suggests a cancer promoting role for the protein (*Kiss et al., 2016*). Interestingly, cell-selective targeted ablation of p38δ in keratinocytes and in immune (myeloid) cells on skin tumor development had different effects. Conditional keratinocyte-specific p38δ ablation reduced malignant progression in males and females relative to their wild-type counterparts. In contrast, conditional myeloid cell-specific p38δ deletion inhibited skin tumorigenesis in male but not female mice. These results reveal that cell-specific p38δ targeting modifies susceptibility to skin carcinogenesis in a context-, stage-, and sex-specific manner (*Kiss et al., 2019*).

The closely related MAPK14, MAPK12, and MAPK13 proteins are known to modulate the immune response, and since chronic inflammation is a known risk factor for tumorigenesis it seems possible that the role of MAPK13 in carcinogenesis may be associated with inflammation. *Del Reino et al., 2014* have analyzed the role of MAPK12 and MAPK13 in colon cancer associated to colitis and have

shown that the deficiency of MAPK12 and MAPK13 significantly decreased tumor formation, in parallel with a decrease in proinflammatory cytokine and chemokine production.

In contrast with the observations arguing for a pro-oncogenic role of the protein, loss of p38δ mitogen-activated protein kinase expression has been shown to promote oesophageal squamous cell carcinoma proliferation, migration, and anchorage-independent growth, suggesting that it has a tumor suppressor role (*O'Callaghan et al., 2013*). Similarly, inactivation of the gene in lung cancer cells has been shown to lead to upregulation of the stemness proteins, thus promoting the cancer stem cell properties of these cells (*Fang et al., 2017*). Promoter methylation of *MAPK13* was found to be present in the majority of primary and metastatic melanomas. Restoration of MAPK13 expression in melanoma cells exhibiting epigenetic silencing of this gene reduced proliferation, indicative of tumor suppressive functions for the protein (*Gao et al., 2013*).

In summary, although MAPK13 plays both pro-oncogenic and tumor suppressor functions in different cellular processes our observation that during tumor evolution negative selection dominates for MAPK13 suggests that the selection pressure to preserve the tumor promoting activities of MAPK13 activity overrides the pressure to eliminate its tumor suppressor activities.

## Protein AF-9, encoded by the *MLLT3* gene

The *MLLT3* gene (present in CGC list of cancer genes) has been selected as a gene showing very high values of rSMN, suggesting negative selection of missense and nonsense mutations (*Supplementary file 3*).

It must be pointed out that based on the high silent/missense ratio *MLLT3* (as well as *TBP* and *DSPP*) has also been identified by others as a gene subject to negative selection (*Zhou et al., 2017*).

Protein AF-9 is a component of a complex required to increase the catalytic rate of RNA polymerase II transcription by suppressing transient pausing by the polymerase at multiple sites along the DNA.

Several studies indicate that *MLLT3* is a proto-oncogene, its inactivation or downregulation suppresses lymphoma cell proliferation, invasion and inhibits metastasis and proliferation of prostate cancer (*Zhang et al., 2012*; *Meng et al., 2017*).

Despite the tumor promoting role of *MLLT3*, the high silent to missense ratio of substitution mutations does not seem to be a reflection of strong negative selection. The translated region of the *MLLT3* gene contains a long stretch of AGC repeats (encoding the polyS segment of the protein, residues 149–194). The 'excess' of silent mutations are clustered in the polyS tract of the protein encoded by the imperfect polymorphic AGC microsatellite region of the *MLLT3* gene, that is known to be highly unstable (*Walker et al., 1994*).

## Neuro-oncological ventral antigen 1 (Nova-1), encoded by the *NOVA1* gene

Nova-1 is an RNA-binding protein involved in the regulation of RNA splicing.

The importance of Nova1 for tumor growth is supported by the observation that overexpressed intratumoral NOVA1 was associated with poor survival rate and increased recurrence rate of hepatocellular carcinoma (HCC) and was an independent prognostic factor for overall survival rate and tumor recurrence. HCC cell lines over-expressing NOVA1 exhibited greater potentials in cell proliferation, invasion, and migration, while knockdown of NOVA1 had the opposite effects. All these findings indicate that NOVA1 may act as a prognostic marker for poor outcome and high recurrence in HCC (*Zhang et al., 2014b*).

Similarly, NOVA1 expression was found to be upregulated in melanoma samples and cell lines. and knockdown of NOVA1 suppressed melanoma cell proliferation, migration and invasion in both A375 and A875 cell lines These results suggested that NOVA1 acted as an OG in the development of melanoma (*Yu et al., 2018*).

Recent studies have shown that the tumor suppressor microRNA-592 suppresses the malignant phenotypes of thyroid cancer by downregulating NOVA1. Whereas overexpression of miR–592 resulted in decreased cell proliferation, migration, and invasion in thyroid cancer, ectopic NOVA1 expression effectively abolished the tumor-suppressing effects of miR–592 overexpression in thyroid cancer cells vitro and in vivo (*Luo et al., 2019*).

Recent studies have provided an explanation for the role of NOVA1 in carcinogenesis. *Sayed et al., 2019* have shown that NOVA1 as well as the polypyrimidine-tract binding protein PTBP1 acts as enhancers of full-length TERT splicing, increasing telomerase activity, promoting telomere maintenance in cancer cells, thereby favoring theit replicative immortality.

## Calcium/calmodulin-dependent protein kinase type 1B, encoded by the *PNCK* gene

Pregnancy upregulated non-ubiquitous calmodulin kinase PNCK is a calcium/calmodulin-dependent protein kinase belonging to a calcium-triggered signaling cascade. It phosphorylates and activates CAMK1 that, upon calcium influx, regulates transcription activators activity, cell cycle, hormone production, and cell differentiation.

Several lines of evidence suggest that PNCK promotes carcinogenesis.

PNCK has been found to be highly overexpressed in human primary human breast cancers compared with benign mammary tissue (*Gardner et al., 2000*). Increased expression of PNCK is associated with poor prognosis in clear cell renal cell carcinoma. The mRNA level of PNCK was significantly higher in tumorous tissues than in the adjacent non-tumorous tissues. Multivariate analysis indicated that PNCK expression was an independent predictor for poor survival of clear cell renal cell carcinoma patients (*Wu et al., 2013*). Overexpression of PNCK in breast cancer cells was shown to result in increased proliferation, clonal growth, and cell-cycle progression (*Deb et al., 2015*).

Recent studies have shown that *PNCK* depletion inhibits proliferation and induces apoptosis of human nasopharyngeal carcinoma cells in vitro and in vivo, suggesting that it might be a novel therapeutic target for treatment of nasopharyngeal carcinoma (*Xu et al., 2019*).

## Runt-related transcription factor 2, encoded by the *RUNX2* gene

The protein is a member of the RUNX family of transcription factors and has a Runt DNA-binding domain. RUNX2 is a transcription factor involved in osteoblastic differentiation and skeletal morphogenesis. RUNX2 plays a cell proliferation regulatory role in cell cycle entry and exit in osteoblasts. These functions are especially important when discussing bone cancer, particularly osteosarcoma development that can be attributed to aberrant cell proliferation control.

Several studies indicate that RUNX2 plays a key role in carcinogenseis. RUNX2 overexpression was found to promote aggressiveness and metastatic spreading, whereas *RUNX2* knockdown inhibits tumor growth and metastasis suggesting an oncogenic role for the protein (*Tandon et al., 2014*; *Tandon et al., 2016*; *Shin et al., 2016*; *Li et al., 2016*; *Wang et al., 2016*; *Sancisi et al., 2017*; *Lu et al., 2018*; *Ji et al., 2019*; *Herreño et al., 2019*).

Although strong purifying selection would not contradict the tumor promoting role of RUNX2, the high silent to missense ratio of substitution mutations is not a reflection of the strength of negative selection of missense and nonsense substitutions.

A noteworthy feature of the *RUNX2* gene is that its translated region contains a long stretch of CAG repeats (encoding the polyQ segment of the protein, residues 49–71). Interestingly, substitutions are not randomly distributed along the sequence of *RUNX2*: they are clustered in the polyQ tract of the protein encoded by the imperfect polymorphic CAG microsatellite region of the *RUNX2* gene. Since in cancer cells defective in mismatch-repair, microsatellites are known to become unstable due to increased frequency of replication error (*Benachenhou et al., 1998*), it seems likely that this increases and distorts mutation pattern in the polyQ region of *RUNX2*, and this mutation hotspot may give the false impression of strong purifying selection.

## Monocarboxylate transporter 4 (MCT 4), encoded by the *SLC16A3* gene

Monocarboxylate transporter 4 (MCT4) or Solute carrier family 16 member 3 (SLC16A3) is a member of the proton-linked monocarboxylate transporter. Protein family. It catalyzes the rapid transport across the plasma membrane of many monocarboxylates such as lactate.

Since due to abnormal conversion of pyruvic acid to lactic acid by tumor cells even under normoxia, the altered metabolism of glucose consuming tumors must rapidly efflux lactic acid to the microenvironment to maintain a robust glycolytic flux and to prevent poisoning themselves

(*Mathupala et al., 2007*). Survival and maintenance of the glycolytic phenotype of tumor cells is ensured by monocarboxylate transporter 4 (MCT4, encoded by the *SLC16A3* gene) that efficiently transports L-lactate out of the cell (*Ganapathy et al., 2009*).

As high metabolic and proliferative rates in cancer cells lead to production of large amounts of lactate, extruding transporters are essential for the survival of cancer cells. This point may be illustrated by the fact that knockdown of MCT4 increased tumor-free survival and decreased in vitro proliferation rate of tumor cells (*Andersen et al., 2018*).

Using a functional screen *Baenke et al., 2015* have also demonstrated that monocarboxylate transporter four is an important regulator of breast cancer cell survival: MCT4 depletion reduced the ability of breast cancer cells to grow, suggesting that it might be a valuable therapeutic target.

In harmony with the essentiality of MCT4 for tumor growth, several studies indicate that expression of the hypoxia-inducible monocarboxylate transporter MCT4 is increased in tumors and its expression correlates with clinical outcome, thus it may serve as a valuable prognostic factor (*Witkiewicz et al., 2012*; *Doyen et al., 2014*; *Baek et al., 2014*)

Consistent with the key importance of MCT4 for the survival of tumor cells, its selective inhibition to block lactic acid efflux appears to be a promising therapeutic strategy against highly glycolytic malignant tumors (*Todenhöfer et al., 2018*; *Choi et al., 2016*; *Choi et al., 2018*; *Zhao et al., 2019b*)

## Solute carrier family 2, facilitated glucose transporter member 1, encoded by the *SLC2A1* gene

SLC2A1 functions as a facilitative glucose transporter, which is responsible for glucose uptake.

Significantly, several nutrient transporter protein genes were found among the genes showing the strongest signs of purifying selection. The most likely explanation for the selective pressure to preserve their integrity is that tumor cells have an increased demand for nutrients and this demand is met by enhanced cellular entry of nutrients through upregulation of specific transporters (*Ganapathy et al., 2009*).

The uncontrolled cell proliferation of tumor cells involves not only deregulated control of cell proliferation but also major adjustments of energy metabolism in order to fuel cell growth and division in the hypoxic microenvironments in which they reside. Otto Warburg was the first to observe an anomalous characteristic of cancer cell energy metabolism: even in the presence of oxygen, cancer cells limit their energy metabolism largely to glycolysis, leading to a state that has been termed 'aerobic glycolysis' (*Warburg, 1956b*). Cancer cells are known to compensate for the lower efficiency of ATP production through glycolysis than oxidative phosphorylation by upregulating glucose transporters, such as GLUT1, thus increasing glucose import into the cytoplasm (*Jones and Thompson, 2009*; *DeBerardinis et al., 2008*; *Hsu and Sabatini, 2008*).

The markedly increased uptake of glucose has been documented in many human tumor types, by noninvasively visualizing glucose uptake through positron emission tomography using a radiolabeled analog of glucose as a reporter. This reliance of tumor cells on glycolysis is also supported by the hypoxia response system: under hypoxic conditions not only glucose transporters but also multiple enzymes of the glycolytic pathway are upregulated (*Jones and Thompson, 2009*; *DeBerardinis et al., 2008*; *Semenza, 2010a*; *Semenza, 2010b*; *Kroemer and Pouyssegur, 2008*)

In our view, the central role of GLUT1 in cancer metabolism is reflected by the fact that the gene (*SLC2A1* gene of solute carrier family member two protein) encoding this glucose transporter is among the genes that show the strongest signatures of purifying selection (see *Supplementary file 31*).

The key importance of GLUT1 in cancer may be illustrated by the fact that high levels of GLUT1 expression correlates with a poor overall survival and is associated with increased malignant potential, invasiveness, and poor prognosis (*Wang et al., 2017a*; *Deng et al., 2018*; *de Castro et al., 2019*).

The strict requirement for GLUT1 in the early stages of mammary tumorigenesis highlights the potential for glucose restriction as a breast cancer preventive strategy (*Wellberg et al., 2016*). The tumor essentiality of GLUT1 may also be illustrated by the fact that knockdown of GLUT1 inhibits cell glycolysis and proliferation and inhibits the growth of tumors (*Xiao et al., 2018*). In view of its

essentiality for tumor growth, GLUT1 is a promising target for cancer therapy (*Shibuya et al., 2015*; *Noguchi et al., 2016*; *Chen et al., 2017d*)

Recent studies suggest that the YAP1-TEAD1-GLUT1 axis plays a major role in reprogramming of cancer energy metabolism by modulating glycolysis (*Lin and Xu, 2017*). These authors have shown that YAP1 and TEAD1 are involved in transcriptional control of the the glucose transporter GLUT1, whereas knockdown of YAP1 inhibited glucose consumption, and lactate production of breast cancer cells. Overexpression of GLUT1 restored glucose consumption and lactate production.

## Solute carrier family 2, facilitated glucose transporter member 8, encoded by the *SLC2A8* gene

The SLC2A8/GLUT8 is a member of the glucose transporter superfamily that mediates the transport of glucose and fructose.

In harmony with the strong signatures of negative selection, there is evidence that GLUT8 plays an important role in carcinogenesis: it is overexpressed in and is required for proliferation and viability of tumors (*Goldman et al., 2006*; *McBrayer et al., 2012*).

## TATA-box-binding protein, encoded by the *TBP* gene

The *TBP* gene has been selected as a gene showing very high values of rSMN, suggesting negative selection of missense and nonsense mutations (*Supplementary file 3*). It must be pointed out that based on the high silent/missense ratio *TBP* (as well as *DSPP* and *MLLT3*) has also been identified by others as a gene subject to negative selection (*Zhou et al., 2017*).

The protein is a general transcription factor that functions at the core of the DNA-binding multi-protein factor TFIID. Binding of TFIID to the TATA box is the initial transcriptional step of the pre-initiation complex, playing a role in the activation of eukaryotic genes transcribed by RNA polymerase II. In view of such a basic cell essential function, it seemed justified to assume that it is the indispensability of the gene for the survival of tumor cells (just like any other cell) that subjects it to strong purifying selection and the high silent/missense ratio is a reflection of this negative selection. TBP has been thought to be an invariant housekeeping protein, however, several studies have shown that TBP expression is significantly increased in both colon adenocarcinomas as well as adenomas relative to normal tissue, supporting the idea that increases in TBP expression actually drive tumorigenesis (*Johnson et al., 2003a*; *Johnson et al., 2003b*; *Johnson et al., 2017*).

Inspection of the spectrum of somatic mutations of the *TBP* gene suggests that the high silent/missense ratio is unlikely to be simply due to negative selection that may hold for both OGs and TEGs. A noteworthy feature of the *TBP* gene is that its translated region contains a long stretch of CAG repeats (encoding the polyQ segment of the protein, residues 57–95). The distribution of silent mutations is markedly non-random: they are clustered in the polyQ tract of the protein encoded by the imperfect polymorphic CAG microsatellite region of the *TBP* gene. Since in cancer cells defective in mismatch-repair, microsatellites are known to become unstable due to increased frequency of replication error (*Benachenhou et al., 1998*), it seems likely that this is why the rate of mutation in the polyQ region of TBP is much higher than in other regions of the gene. The high silent to missense rate is thus not due to negative selection acting on missense and nonsense substitutions. Rather, it may reflect the fact that the imperfect polymorphic CAG microsatellite region of the *TBP* gene serves as a mutation hotspot, with a biased substitution pattern.

## Thromboxane A2 receptor, encoded by the *TBXA2R* gene

TBXA2R is a plasma membrane protein that serves as a receptor for thromboxane A2, a potent stimulator of platelet aggregation. The activity of this receptor is mediated by a G-protein that activates a phosphatidylinositol-calcium second messenger system.

Studies on the expression of thromboxane A2 receptor, TBXA2R in a cohort of human breast cancer patients revealed that breast tumor tissues expressed higher levels of TBXA2R compared with normal mammary tissues and that TBXA2R expression was most significantly increased in grade three tumors. Kaplan-Meier survival analysis has also shown that patients with high levels of TBXA2R had significantly shorter disease-free survival. The observation that TBXA2R is highly expressed in

aggressive tumors and linked with poor prognosis indicates that TBXA2R has a significant prognostic value in clinical breast cancer (*Watkins et al., 2005*).

The role of TBXA2R in carcinogenesis is also supported by the observation that Thromboxane A2 was shown to enhance tumor metastasis and that the tumor promoting activity required intact TBXA2 receptor (*Matsui et al., 2012*). These studies revealed that TBXA2-TBXA2R signaling plays a critical role in tumor colonization through P-selectin-mediated interactions between platelets-tumor cells and tumor cells-endothelial cells, suggesting that blockade of this signaling might be useful in the treatment of tumor metastasis.

Although the involvement of TBXA2-TBXA2R signaling in cancer invasion and metastasis apperars to be clearly established, there may be other mechanisms by which TBXA2 promotes these processes. *Li and Tai, 2013* have shown that a TBXA2 mimetic induced the expression of the monocyte chemoattractant chemokine ligand protein CCL2, suggesting that TBXA2 may also stimulate invasion of cancer cells through CCL2-CCR2 mediated macrophage recruitment.

Recent studies on Triple Negative Breast Cancer (TNBC) cell lines revealed that TBXA2R expression was higher in these cell lines and that *TBXA2R* knockdowns consistently showed dramatic cell killing in TNBC cells (*Orr et al., 2016*). It has also been shown that TBXA2R enhanced TNBC cell migration, invasion, indicating that the gene is required for the survival and migratory behavior of a subset of TNBCs.

A phenome-wide association study has shown that a single-nucleotide polymorphism in the gene *TBXA2R* is associated with increased metastasis in multiple primary cancers, suggesting the requirements for thromboxane A2 (TXA2) and TBXA2R in the basic mechanism of metastasis, and the clinical applicability of TBXA2R antagonists as adjuvant therapy in multiple cancers (*Pulley et al., 2018*).

## Tumor protein p73, encoded by the *TP73* gene

The protein is known to participate in the apoptotic response to DNA damage: isoforms containing the N-terminal transactivation domain are pro-apoptotic, isoforms lacking the transactivation domain are anti-apoptotic.

Although p73 shows substantial homology with p53, despite the established role of p53 as a tumor suppressor, p73 does not have a similar tumor suppressor role in malignancy: unlike p53-/- mice, p73 knockout mice do not develop tumors. In fact, N-terminally truncated p73 isoforms, lacking the transactivation domain were shown to possess oncogenic potential (*Stiewe and Pützer, 2002*; *Stiewe et al., 2002*).

Numerous studies have shown that ΔNp73, the oncogenic isoform of p73 lacking the transactivation domain, is frequently upregulated in many carcinomas and is indicative of poor prognosis (*Zaika et al., 2002*; *Petrenko et al., 2003*; *Domínguez et al., 2006*; *Hassan et al., 2014b*; *Hassan et al., 2014a*; *Lucena-Araujo et al., 2015*).

Our observation that p73, an oncogenic protein, shows only strong signatures of purifying selection provides one of the clearest examples illustrating the point that in the case of OGs purifying selection is not necessarily associated with positive selection for driver mutations. It must be pointed out here that it has been noted earlier by others that, despite its clear role in carcinogenesis, the *TP73* gene is almost never mutated (*Bisso et al., 2011*; *Maas et al., 2013*). One may argue that in this case the molecular change that drives carcinogenesis is the change of splicing that favors the formation of the oncogenic isoform of p73.

## Tribbles homolog 2, encoded by the *TRIB2* gene

TRIB2 is a pseudokinase member of the pseudoenzyme class of signaling/scaffold proteins. It interacts with MAPK kinases and regulates activation of MAP kinases.

TRIB2 has been shown to be important in the maintenance of the oncogenic properties of melanoma cells, as its silencing reduces cell proliferation, colony formation. Tumor growth was also substantially reduced upon RNAi-mediated TRIB2 knockdown in an in vivo melanoma xenograft model, suggesting that TRIB2 provides the melanoma cells with growth and survival advantages (*Zanella et al., 2010*).

TRIB2 expression is elevated in primary human lung tumors and in NSCLC cells, resulting from gene amplification. *TRIB2* knockdown was found to inhibit cell proliferation and in vivo tumor growth, indicating that TRIB2 is a potential driver of lung tumorigenesis (*Grandinetti et al., 2011*).

High TRIB2 expression is observed in T cell acute lymphoblastic leukemias (*Hannon et al., 2012*). TRIB2 has been shown to be critical for both solid and non-solid malignancies and is functionally important for liver cancer cell survival and transformation. TRIB2 was found to be upregulated in liver cancer cells compared with other cells (*Wang et al., 2013a*; *Wang et al., 2013b*).

TRIB2 is emerging as a pivotal target of transcription factors in acute leukemias as evidenced by the fact *TRIB2* knockdown resulted in a block in acute myeloid leukemia cell proliferation (*Rishi et al., 2014*)

In the case of lung adenocarcinoma, patients with higher TRIB2 levels had poorer survival (*Zhang et al., 2016*). The tumor-promoting role of this protein is supported by the observation that TRIB2 expression is significantly increased in tumor tissues from patients with extremely poor clinical outcome (*Hill et al., 2017*; *Wang et al., 2020*).

TRIB2 has been shown to be important for the survival of leukemia cells during MLL-TET1-related leukemogenesis and for maintaining differentiation blockade of leukemic cells: *TRIB2* knockdown relieved the inhibition of myeloid cell differentiation induced by the MLL-TET1 fusion protein (*Kim et al., 2018*).

TRIB2 expression has been shown to be elevated in colorectal cancer tissues compared to normal adjacent tissues and high TRIB2 expression indicated poor prognosis of colorectal cancer patients (*Hou et al., 2018*). Depletion of TRIB2 inhibited cancer cell proliferation, induced cell cycle arrest and promoted cellular senescence, whereas overexpression of TRIB2 accelerated cell growth, cell cycle progression and blocked cellular senescence.

## Twist-related protein 1, encoded by the *TWIST1* gene

The *TWIST1* gene is characterized by very high value of rSMN (*Supplementary file 5*), indicating strong signature of purifying selection, suggesting that it plays an important role in promoting tumorigenesis.

Twist-related protein 1, TWIST1 is a transcription factor and master regulator of the epithelial-to-mesenchymal transition that significantly contributes to tumor growth and metastasis. TWIST1 is overexpressed in a variety of tumors and numerous studies have shown that targeting TWIST1 significantly inhibits tumor growth (*Wushou et al., 2014*; *Zhu et al., 2016*; *Xu et al., 2017a*; *Xu et al., 2017b*; *Mikheev et al., 2018*).

Recent studies have revealed that AURKA and TWIST1 are linked in as much as ablation of either AURKA or TWIST1 completely inhibits epithelial-to-mesenchymal transition (*Wang et al., 2017b*).

# Appendix 2

## Analyses of somatic substitutions and subtle indel mutations of human protein-coding genes of tumor tissues

We have used two major types of analyses of silent, amino acid changing and truncating somatic mutations of human protein-coding genes of tumor tissues: one in which we have restricted our analyses to single -nucleotide substitutions (SO or' substitution only' analyes, for details, see main text).

Here, we describe the analyes that also take into account subtle indels (SSI or' substitutions and subtle indels' analyses). In these analyses, subtle mutations affecting the coding sequences of protein coding genes were assigned to three categories: SIL, silent synonymous substitutions, MIS, merging nonsynonymous substitutions and short inframe indels that alter but do not disrupt coding sequence, and NON, merging nonsense substitutions and short frame-shift indels as both types of mutations lead eventually to stop codons that truncate the protein. Unless otherwise idicated, we have used datasets containing transcripts with at least 100 confirmed somatic, non-polymorphic mutations identified in tumor tisses.

We have used several approaches to analyze the contribution of silent, amino acid changing and truncating mutations to somatic mutations of human protein-coding genes during tumor evolution.

In the simplest case, we have calculated for each transcript the fraction of somatic mutations that could be assigned to the synonymous (indel_fS), nonsynonymous (indel_fM), and nonsense mutation (indel_fN) category.

Our analyses have shown that in the 3D representation of SSI mutations (see *Appendix 2—figure 1*, Panel A) genes are present in a cluster characterized by fraction values of $0.24082 \pm 0.06203$, $0.70086 \pm 0.05701$ and $0.05832 \pm 0.04151$ for indel_fS, indel_fM and indel_fN category, respectively. The mean values for indel_fS, indel_fM and indel_fN in this cluster are very similar to those observed for fS, fM, and fN in SO analyses (*Supplementary file 29*), consistent with the observation that in the dataset containing transcripts with at least 100 confirmed somatic, non-polymorphic mutations identified in tumor tissues subtle indels are much rarer than single-nucleotide substitutions (*Supplementary file 8*).

A                                    B

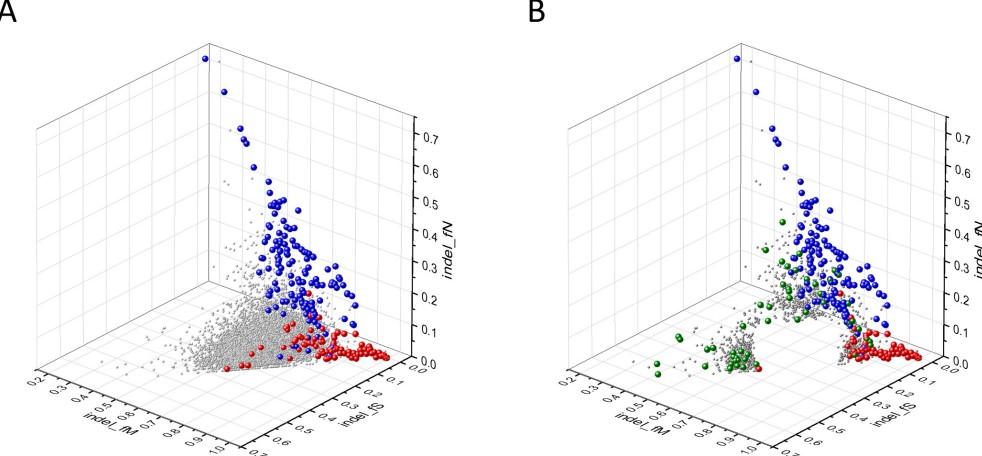

**Appendix 2—figure 1.** Analyses of indel_fS, indel_fM and indel_fN parameters of human protein-coding genes of tumor tissues. The figure shows the results of the analysis of 13,930 transcripts containing at least 100 subtle, confirmed somatic non-polymorphic mutations from tumor tissues. Axes *x, y* and *z* represent the fractions of somatic mutations that are assigned to the indel_fS, indel_fM and indel_fN categories. In Panel A, each ball represents a human transcript; note that the majority of human genes are present in a dense cluster. The positions of transcripts of the genes defined by *Vogelstein et al., 2013* as oncogenes (OGs, large red balls) or tumor suppressor genes (TSGs, large blue balls) are highlighted. It is noteworthy that these driver genes separate significantly from the central cluster and from each other: OGs have an increased fraction of indel_fM, whereas TSGs have markedly increased fraction of indel_fN. Panel B shows data only for

*Appendix 2—figure 1 continued on next page*

*Appendix 2—figure 1 continued*

candidate cancer genes present in the CG_SO$^{2SD}$_SSI$^{2SD}$ list (see *Supplementary file 31*). The positions of transcripts of the genes identified by *Vogelstein et al., 2013* as OGs (large red balls) or TSGs (large blue balls) are highlighted. The positions of novel cancer gene transcripts validated in the present work are highlighted as large green balls.

It is noteworthy, however, that the pattern of indel_fS, indel_fM and indel_fN of the best known cancer genes (*Vogelstein et al., 2013*) deviates significantly from that characteristic of the majority of human genes (see *Appendix 2—figure 1*, Panel A). The values for OGs show a marked incease in indel_fM, reflecting positive selection for missense mutations, whereas the values for TSGs show significant increase in indel_fN, reflecting primarily positive selection for truncating nonsense mutations (*Supplementary file 29*).

The set of genes (6139 transcripts) with values that deviate from mean values of indel_fS, indel_fM and indel_fN by more than 1SD have also included the majority of OGs and TSGs (only 5 OG and 1 TSG transcripts remained in the central cluster). It is noteworthy that the 6139 transcripts also contained the vast majority (443 out of 748) of the transcripts of CGC genes, suggesting that the mutation pattern of most CGC genes also deviates significantly from that of passenger genes (PGs; *Supplementary file 29*). The genes in the central cluster (*Supplementary file 29*) is hereafter referred to as PG_SSI$^{f-1SD}$ (for Passenger Gene_Substitution and Subtle Indels deviating from mean indel_fS, indel_fM and indel_fN values by ≤1 SD).

The set of genes (1211 transcripts) with values that deviate from mean values of indel_fS, indel_fM and indel_fN by more than 2SD included 62 OG and 123 TSG driver gene transcripts (see *Appendix 2—figure 1*, Panel B). Using this more stringent cut-off value, the number of CGC genes identified in the 1211 transcripts was reduced to 153 out of 748 (*Supplementary file 29*). The non-PG set defined by 2SD cut-off value is hereafter referred to as CG_SSI$^{f-2SD}$ for Cancer Gene_ Substitution and Subtle Indels deviating from mean indel_fS, indel_fM and indel_fN values by more than 2SD (*Supplementary file 29*).

The 1211 transcripts in the gene set of CG_SSI$^{f-2SD}$ has 873 transcripts not found in the OG, TSG, and CGC cancer gene lists (*Supplementary file 5*). Since the majority of these 873 transcripts (derived from 743 genes) have parameters that assign them to the OG or TSG clusters, we assume that they also qualify as candidate OGs or TSGs. There is, however, a third group of genes that deviate from both the central PG cluster and the clusters of OGs and TSGs: their high indel_fS and low indel_fM and indel_fN values suggest that they experience purifying selection during tumor evolution, suggesting that they may correspond to TEGs important for the growth and survival of tumors. The 743 putative cancer genes listed in CG_SSI$^{f-2SD}$ of *Supplementary file 5*, were subjected to further analyses to decide whether they qualify as candidate OGs, TSGs, TEGs or the deviation of their mutation pattern from those of PGs is not the result of natural selection. For some typical examples of these analyses see Appendix 1.

Known cancer genes (OGs and TSGs) also separate from the majority of human genes in 3D representations of parameters indel_rSM, indel_rNM, indel_rNS defined as the ratio of indel_fS/indel_fM, indel_fN/indel_fM, indel_fN/indel_fS, respectively (see *Appendix 2—figure 2*). In these representations (see *Appendix 2—figure 2*, Panels A1, A2), OGs separate from the central cluster in having significantly lower indel_rSM and indel_rNM values, whereas TSGs had significantly higher indel_rNS and indel_rNM values than those of the central cluster.

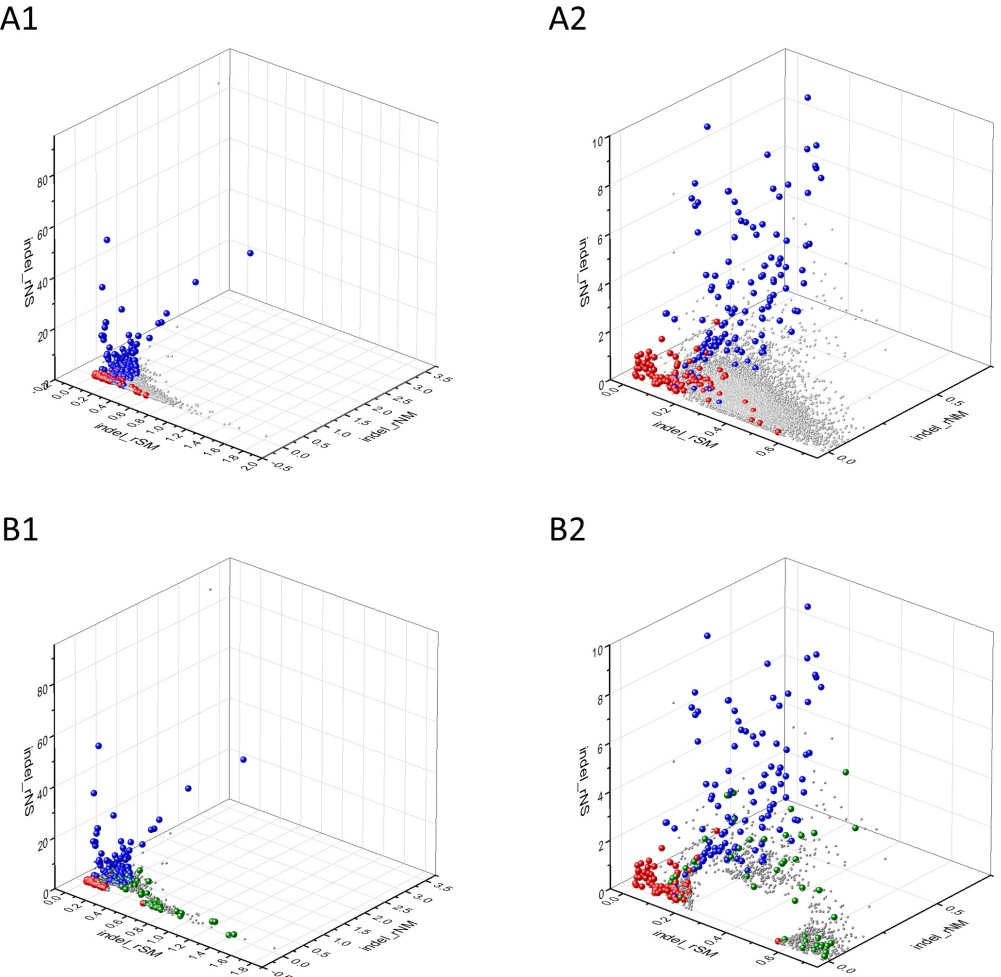

**Appendix 2—figure 2.** Analyses of indel_rSM, indel_rNM, indel_rNS parameters of human protein-coding genes of tumor tissues. The figure shows the results of the analysis of 13930 transcripts containing at least 100 subtle, confirmed somatic mutations from tumor tissues, including only mutations identified as not SNPs. Axes *x*, *y*, and *z* represent the indel_rSM, indel_rNM, indel_rNS values defined as the ratio of indel_fS/ indel_fM, indel_fN/ indel_fM, indel_fN/ indel_fS, respectively. Each ball represents a human transcript; the positions of transcripts of the genes identified by *Vogelstein et al., 2013* as oncogenes (OGs, large red balls) or tumor suppressor genes (TSGs, large blue balls) are highlighted. Panels A1, A2 show the distribution of the 13,930 transcripts at different magnification. Note that the majority of human genes are present in a dense cluster but known OGs and TSGs separate significantly from the central cluster and from each other. The rNS and rNM values of TSGs are higher, whereas the rSM and rNM values of OGs are lower than those of passenger genes. Panels B1,B2 show data only for candidate cancer genes present in the CG_SO$^{2SD}$_SSI$^{2SD}$ list (see *Supplementary file 31*). The positions of transcripts of the genes identified by *Vogelstein et al., 2013* as OGs (large red balls) or TSGs (large blue balls) are highlighted. The positions of novel cancer gene transcripts validated in the present work are highlighted as large green balls.

The set of genes (4518 transcripts) with values that deviate from the mean by more than 1SD contained 78 OG transcripts, 132 TSG transcripts, and 368 CGC gene transcripts (*Supplementary file 29*). The central cluster of genes (that deviate from mean rSM, rNM and rNS values by ≤1 SD) is hereafter referred to as PG_SSI$^{r2–1SD}$ (for Passenger Gene_Substitution and Subtle Indels deviating from mean indel_rSM, indel_rNM, indel_rNS values by ≤1 SD).

The non-PG set defined by 2SD cut-off value (see *Appendix 2—figure 2*, B1, B2, *Supplementary file 5*) is hereafter referred to as CG_SSI$^{r2–2SD}$ for Cancer Gene_ Substitution and Subtle Indels deviating from mean indel_rSM, indel_rNM, indel_rNS values by more than 2SD

(*Supplementary file 29*). This gene set has a total of 861 transcripts, containing 40 transcripts of OGs, 98 transcripts of TSGs genes, 86 transcripts of CGC genes and 637 transcripts (derived from 546 genes) not found in the OG, TSG, and CGC cancer gene lists (*Supplementary file 5*).

The mean parameters of TSGs differ markedly from those of PGs in that rNS and rNM values are higher, reflecting the dominance of positive selection for inactivating mutations. The parameters for OGs on the other hand, differ from those of PGs in that indel_rSM values of OGs are significantly lower, reflecting positive selection for missense mutations (see *Appendix 2—figure 2*, Panels A1, A2). Interestingly, in this representation some OGs (e.g. *BCL2*) have unusually high scores of indel_rSM suggesting that in the case of these OGs purifying selection may override positive selection for amino acid changing mutations.

As mentioned above, the non-PG set defined by a cut-off values of 2SD contains 637 transcripts (derived from 546 genes) not found in the OG, TSG, or CGC lists. Since the majority of these genes have paramerters that assign them to the OG or TSG clusters, they can be regarded as candidate OGs or TSGs. There is a group of genes that deviate from the clusters of PGs, OGs, and TSGs (see *Appendix 2—figure 2*, Panels B1, B2) in that they have unusually high indel_rSM values. Since high indel_rSM values may be indicative of purifying selection we assume that they may correspond to TEGs important for the growth and survival of tumors. The 546 putative cancer genes listed in CG_SO$^{indel\_r2\_2SD}$ of *Supplementary file 5*, were subjected to further analyses to decide whether they qualify as candidate OGs, TSGs, TEGs or the deviation of their mutation pattern from those of PGs is not the result of natural selection. For examples of these analyses see Appendix 1.

The separation of known cancer genes from the majority of human genes is also observed in 3D representations of parameters indel_rSMN, indel_rMSN and indel_rNSM defined as the ratio of indel_fS/(indel_fM+indel_fN), indel_fM/(indel_fS+indel_fN) and indel_fN/(indel_fS+indel_fM), respectively (see *Appendix 2—figure 3*, Panels A1, A2). In this representation the genes are present in a three pronged cluster.

A1 A2

B1 B2

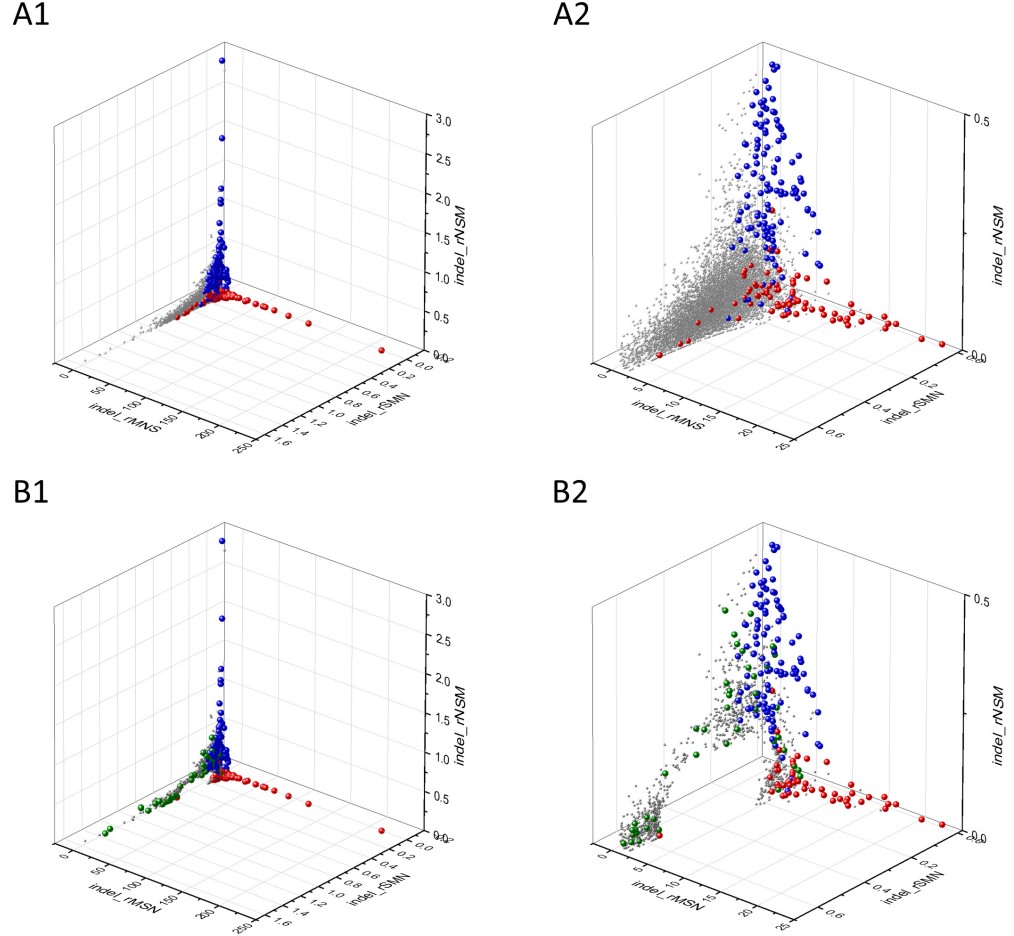

**Appendix 2—figure 3.** Analyses of indel_rSMN, indel_rMSN and indel_rNSM parameters of human protein-coding genes of tumor tissues. The figure shows the results of the analysis of 13930 transcripts containing at least 100 subtle, confirmed somatic mutations from tumor tissues. Axes *x*, *y*, and *z* represent paramerters indel_rSMN, indel_rMSN and indel_rNSM defined as the ratio of indel_fS/(indel_fM+indel_fN), indel_fM/(indel_fS+indel_fN) and indel_fN/(indel_fS+indel_fM), respectively. Each ball represents a human transcript; the positions of transcripts of the genes defined by *Vogelstein et al., 2013* as oncogenes (OGs, red balls) or tumor suppressor genes (TSGs, blue balls) are highlighted. Panels A1 and A2 show the distribution of the 13,930 transcripts at different magnification. Note that the majority of human genes are present in a dense cluster but known OGs and TSGs separate significantly from the central cluster and from each other. The indel_rNSM values of TSGs are higher, their indel_rMSN and indel_rSMN are lower than those of passenger genes. OGs also separate from passenger genes in that their indel_rMSN values are higher and their indel_rSMN values are lower than those of passenger genes. Panels B1,B2 show data at different magnification only for candidate cancer genes present in the CG_SO$^{2SD}$_SSI$^{2SD}$ list (see *Supplementary file 31*). The positions of transcripts of the genes identified by *Vogelstein et al., 2013* as OGs (large red balls) or TSGs (large blue balls) are highlighted. The positions of novel cancer gene transcripts validated in the present work are highlighted as large green balls.

The set of genes (4369 transcripts) with values that deviate from the mean by more than 1SD contained 78 OG transcripts, 132 TSG transcripts and 354 CGC gene transcripts (*Supplementary file 29*). The central cluster of genes, deviating from mean rSMN, rMSN and rNSM values by ≤1 SD is hereafter referred to as PG_SO$^{indel\_r3\_1SD}$ (for Passenger Gene_ Substitution and Subtle Indels deviating from mean indel_rSMN, indel_rMSN and indel_rNSM values by ≤1 SD).

The non-PG set defined by 2SD cut-off value (see *Appendix 2—figure 3*, Panels B1, B2, *Supplementary file 5*) is hereafter referred to as CG_SSI$^{r3\_2SD}$ for Cancer Gene_ Substitution and

Subtle Indels deviating from mean indel_rSMN, indel_rMSN, and indel_rNSM values by more than 2SD (*Supplementary file 29*). This gene set has a total of 823 transcripts, containing transcripts of 37 OGs, 100 TSGs, 86 CGC genes and 600 transcripts (derived from 510 genes) not found in the OG, TSG, and CGC cancer gene lists (*Supplementary file 5*).

The mean parameters of TSGs differ markedly from those of PGs in as much as indel_rNSM values of TSGs are higher and indel_rSMN values are lower, reflecting the dominance of positive selection for inactivating mutations. In the case of OGs on the other hand, indel_rMSN values are higher and indel_rNSM values are lower than those of PGs, reflecting positive selection for missense mutations and purifying selection avoiding nonsense mutations. Interestingly, some OGs have unusually high scores of indel_rSMN suggesting that in these cases (e.g. *BCL2*) purifying selection may override positive selection for amino acid changing mutations.

As mentioned above, the non-PG set defined by a cut-off value of 2SD contains 600 transcripts (derived from 510 genes) not found in the OG, TSG or CGC lists. Since the majority of these genes have paramerters that assign them to the OG or TSG clusters, they can be regarded as candidate OGs or TSGs.

In this representation, we also note the existence of a group of genes that deviates from the clusters of PGs, OGS, and TSGs (see *Appendix 2—figure 3*): their high indel_rSMN and low indel_rMSN and indel_rNSM values suggest that they experience purifying selection during tumor evolution, suggesting that they may be essential for the survival of tumors as OGs or TEGs. The 510 putative cancer genes listed in CG_SSI$^{r3\_2SD}$ of *Supplementary file 5*, were subjected to further analyses to decide whether they qualify as candidate OGs, TSGs and TEGs or the deviation of their mutation pattern from those of PGs is not the result of natural selection. For some typical examples of these analyses see Appendix 1.

