## [Decision Letter]

**Acceptance summary:**

This article describes a new method to re-examine the role of negative selection in cancer. The authors find new evidence of negative selection against inactivating somatic mutations, thus highlighting new mechanisms of tumor survival.

**Decision letter after peer review:**

Thank you for submitting your article "Use of signals of positive and negative selection to distinguish cancer genes and passenger genes" for consideration by *eLife*. Your article has been reviewed by two peer reviewers, and the evaluation has been overseen by a Reviewing Editor and Patricia Wittkopp as the Senior Editor. The reviewers have opted to remain anonymous.

The reviewers have discussed the reviews with one another and the Reviewing Editor has drafted this decision to help you prepare a revised submission.

Summary

This work describes a novel approach to address the important and still open question of the extent of negative selection in cancer and the potential implications. The authors use data from the catalogue of somatic mutations (COSMIC) and a straightforward approach comparing synonymous, nonsynonymous and nonsense mutation counts to separate genes into Oncogenes, Tumor suppressors and Essential genes. The authors conclude that negative selection plays an important role during tumor evolution.

Essential revisions

Reviewers agreed that this work is timely and relevant, but also agreed that there are several important aspects that need revision/improvement before it can be accepted for publication in *eLife*.

Structure of the paper

1) The reviewers agreed that there are various aspects of the structure of the paper that require especial attention. The Introduction is a bit lengthy and very focused. It introduces different questions, e.g. hallmarks, prediction of oncogenes and tumor suppressors, prediction of selection, etc and it reads like multiple introductions to different articles. Many parts (e.g. the discussion of cancer hallmarks) could be shortened substantially, which would make it easier to read the paper. One suggestion is to mainly introduce the models of cancer evolution with respect to the SNVs and indels, and the different models and limitations in the estimation of negative selection in cancer and why it is difficult to detect, see e.g. (Zapata et al., 2018, Lopez et al., 2020, Tilk et al., 2019).

2) Additionally, it will be important to include citations to previous work on the detection of negative selection in cancer that has been omitted. For example, in paragraph five of “Carcinogenesis as an evolutionary process” they should add the work from (Zapata et al., 2018, Van den Eynden et al., 2017, Martincorena et al., 2017, Pyatnitskiy et al., 2015).

3) Both reviewers agreed that the Results section is repetitive and unbalanced with respect to the Materials and methods section. The work would benefit from streamlining the Results part and moving details to the Materials and methods section.

4) Regarding the Discussion, it is also very lengthy and lack focus. The authors should make clearer the main results and take-home messages from their work. At the moment, this is not very clear.

5) For simplicity and to improve readability of the manuscript, it was suggested that the authors focus on 2 standard deviation through the manuscript, instead of describing repetitively the results with 1SD and 2SD.

6) Regarding the presentation of the results, the reviewers suggested to redesign the figures in such a way that they describe the methodological approach, present the major results of their analysis, and show a comparison of these results with previous methods, and lastly (currently as a table) show the association between the identified genes and the hallmarks of cancer.

Comparisons with previous studies

7) One of the problems with the present work raised by the reviewers is that the authors did not performed sufficient comparisons of their results with previous studies. The authors used a seemingly simple approach to measure selection, dividing fractions of frequencies of different mutation classes by each other, with relatively arbitrary cutoffs, e.g. 1 or 2 standard deviations from the mean, to define gene sets. The manuscript does not show the advantages of this method over previous approaches. The authors should clearly show that there is an advantage of their approach by comparing with previous approaches.

8) The authors should also compare their results with previous publications. One of them, which is cited in the manuscript, is Weghorn and Sunyaev. In fact, this work seems to be misquoted. The authors claim that Weghorn and Sunyaev "identified 147 genes with strong negative selection", but that study in fact found very few genes under significant negative selection (<10 applying a q-value cutoff of 0.1) and Weghorn and Sunyaev concluded that "the signal of negative selection is very subtle". Zapata et al., 2018 identified stronger signals of negative selection. The identified genes and functions were partly the same as in the here presented work (eg GLUT1). The authors should compare their results to these and other previous results.

9) Furthermore, there is recent evidence that correcting for mutational signatures and nucleotide-context composition has a large impact when quantifying selection (see e.g. Zapata et al., 2018, van den Eynden et al., 2017, Martincorena et al., 2017), and this is a relevant aspect in the current lines of discussion in the context of negative selection in tumor evolution (see for example Van den Eynden et al., Nature Genetics. 2019). The authors should show that their main observations hold when the mutational signatures and/or trinucleotide context is taken into account.

10) Related to this, the authors described a clustering-based method to detect genes that deviate from an average proportion of mutations (nonsynonymous, nonsense and synonymous) to infer selection. However, by only using the observed mutations (nonsyn, syn, nonsense), the underlying base-pair composition is ignored. Genes that have a high likelihood of acquiring nonsense mutations will show a deviation from the rest of the genes due to their composition and not due to selection. The authors should recalculate their metrics by performing this correction before reaching the conclusion on the number and identity of the genes.

Use of controls

11) The reviewers also indicated the lack of sufficient controls. To improve the robustness of their method, it was suggested to assess the results after varying several of the conditions. For instance, to circumvent the limitation of the lack of mutations to detect negative selection, the authors study only transcripts with more than 100 mutations. The authors should compare their results using different cut-offs for the minimum number of mutations (50,100,500), and check the performance of their method and whether their results are robust.

12) Other variations that the authors should consider is to stratify data based on tumor type and mutation burden, since mixing samples with different evolutionary histories might confound the signal of negative selection. As an additional control, a reviewer suggested to perform the same analyses using the germline mutations to separate the genes into cancer specific or cell essential.

13) An additional control to be performed by the authors was related to the origin of the mutations. The file CosmicMutantExport.tsv contains both mutation data from targeted and genome- / exome-wide screens. Targeted data should be excluded (if the authors didn't do so already). Otherwise their analysis will be highly biased towards well characterized cancer genes.

Statistical tests

14) The reviewers also agreed that there is a general lack of statistical tests in the results. For instance, "the mean parameters of TSGs differ markedly from those of passenger genes in that rNS and rNM values are higher", but these comparisons should be done with appropriate statistical tests to assess the significance. Similar tests should be performed throughout the manuscript.

15) A very interesting idea in the paper highlighted by the reviewers is that by combining their proposed metrics they can differentiate between oncogenes and tumor suppressors. It would be convenient to have a visual interpretation on how different genes can be only oncogenic, only tumor suppressors, or both, depending on which sites are hit. It is important to note though that similar classifiers have been developed (Schroeder et al., 2014), so it would strengthen the claims of the study to provide a comparison with those methods.

---

## [Author Response]

Essential revisionsReviewers agreed that this work is timely and relevant, but also agreed that there are several important aspects that need revision/improvement before it can be accepted for publication in eLife.Structure of the paper1) The reviewers agreed that there are various aspects of the structure of the paper that require especial attention. The Introduction is a bit lengthy and very focused. It introduces different questions, e.g. hallmarks, prediction of oncogenes and tumor suppressors, prediction of selection, etc and it reads like multiple introductions to different articles. Many parts (e.g. the discussion of cancer hallmarks) could be shortened substantially, which would make it easier to read the paper. One suggestion is to mainly introduce the models of cancer evolution with respect to the SNVs and indels, and the different models and limitations in the estimation of negative selection in cancer and why it is difficult to detect, see e.g. (Zapata et al., 2018, Lopez et al., 2020, Tilk et al., 2019).

We agree that the Introduction is a bit longer than usual. Our excuse is that the Introduction intends to provide answers to questions that are likely to be raised by readers, some of which have been raised by the reviewers. (See for example, Comment 15, Response 15 as to why we do not propose that our approach should be used as a tool to classify oncogenes and tumor suppressors.)

We think that the major part of the Introduction is essential for understanding the rationale and novelty of our approach (prediction of selection), the difficulties of detection of negative selection (detection of purifying selection) and the rationale for the biological interpretation of our results (cancer hallmarks).

Re: prediction of selection

Although this part of the Introduction might appear superfluous, we think that it is important for several reasons.

First, it allows us to present clear definitions for the terms used in the paper (cancer gene, driver gene, tumor essential gene, cell-essential gene, passenger gene, driver mutation, passenger mutation, deleterious mutation etc.). Second, this part of the Introduction explains the relationship of these terms to negative selection, positive selection or neutral evolution in the case of germline cells or tumor cells. We believe that these parts are important since in the scientific literature authors use some of these terms in a different sense and this may lead to significant confusion. Finally, the discussion of “prediction of selection” is important as it permits us to highlight differences between our approach and earlier approaches used for detection of signals of selection during tumor evolution.

Re: detection of negative selection

In accordance with the suggestion of the reviewers, we now discuss the various models of cancer evolution (with respect to point mutations) in detail with major emphasis on the problems associated with the estimation of negative selection in cancer. As recommended by the reviewers, in this section we also discuss the papers of Zapata et al., 2018, Lopez et al., 2020 and Tilk et al., 2019.

Re: cancer hallmarks

We think that a short introduction to cancer hallmarks (subsection “Hallmarks of cancer and the function of genes involved in carcinogenesis” and Figure 1 summarizing these hallmarks) is necessary as we have assigned (see Appendix 1) the various novel cancer genes identified in the present study (oncogenes, tumor suppressor genes, tumor essential genes) to cancer hallmarks defined in this section.

2) Additionally, it will be important to include citations to previous work on the detection of negative selection in cancer that has been omitted. For example, in paragraph five of “Carcinogenesis as an evolutionary process” they should add the work from (Zapata et al., 2018, Van den Eynden et al., 2017, Martincorena et al., 2017, Pyatnitskiy et al., 2015).

We have added discussion of these earlier studies to the Introduction (and to other sections of the revised manuscript as appropriate).

3) Both reviewers agreed that the Results section is repetitive and unbalanced with respect to the Materials and methods section. The work would benefit from streamlining the Results part and moving details to the Materials and methods section.

We have streamlined and restructured the Results and Materials and methods sections by moving all methodological details of our analyses from the Results section to the Materials and methods section. The revised version of Materials and methods also incorporates various analyses requested by the reviewers (see below).

4) Regarding the Discussion, it is also very lengthy and lack focus. The authors should make clearer the main results and take-home messages from their work. At the moment, this is not very clear.

We have restructured the Results, Discussion and Conclusion sections of the earlier version in accordance with the suggestion of the reviewers. The main results of our analyses of candidate cancer gene sets (previously constituting the major part of Discussion) are now included in the Results section, whereas the take-home message (previously found in Conclusion) is now included in the Discussion. Note that the Results section of the revised manuscript also incorporates the results of the analyses requested by the reviewers (see below).

5) For simplicity and to improve readability of the manuscript, it was suggested that the authors focus on 2 standard deviation through the manuscript, instead of describing repetitively the results with 1SD and 2SD.

In the revised manuscript, where we discuss candidate cancer genes, we just focus on 2 standard deviations throughout the manuscript.

6) Regarding the presentation of the results, the reviewers suggested to redesign the figures in such a way that they describe the methodological approach, present the major results of their analysis, and show a comparison of these results with previous methods, and lastly (currently as a table) show the association between the identified genes and the hallmarks of cancer.

In our presentation of the results (Results section) we have followed the advice of the reviewers (approach > results >comparison >hallmarks).

Comparisons with previous studies7) One of the problems with the present work raised by the reviewers is that the authors did not performed sufficient comparisons of their results with previous studies. The authors used a seemingly simple approach to measure selection, dividing fractions of frequencies of different mutation classes by each other, with relatively arbitrary cutoffs, e.g. 1 or 2 standard deviations from the mean, to define gene sets. The manuscript does not show the advantages of this method over previous approaches. The authors should clearly show that there is an advantage of their approach by comparing with previous approaches.

Re: comparison of results with those of previous studies; advantage compared with previous approaches

As mentioned above (Comment 2/Response 2) in the revised version we have included additional analyses comparing our results with those of previous studies. These comparisons have shown that our analyses led to the identification and validation of several novel candidate cancer genes (oncogenes, tumor suppressor genes, tumor essential genes that show significant signs of positive and/or negative selection during tumor evolution) most of which were missed by earlier studies. Furthermore, detailed annotation (Appendix 1) of candidate cancer genes identified in the present study has provided support for their role in carcinogenesis, verifying their cancer gene status and validating the approach we have used.

Re: use of arbitrary cutoffs to define gene sets

Just like all previous studies that measured positive or negative selection of genes during tumor evolution (by exploiting some signals of selection), we have also ranked the genes according to some measures of selection. All such studies generate gene lists in which genes are ranked according to the parameters defined in the actual study. Rather than discussing thousands of genes of such gene lists, authors use cut offs to focus on genes that show the strongest signs of selection. As will be mentioned below (e.g. Comment 8, Response 8) in many cases a single study uses different cut-offs, permitting the authors to focus on just the genes showing the strongest signs of selection, but also allows them to discuss genes that are included only in a larger gene set defined by less stringent cut-offs.

In the previous version we have also referred to different primary cut-offs (1SD, 2SD to define candidate cancer gene sets). In the revised manuscript, we focus on 2 standard deviation, as recommended by the reviewers.

8) The authors should also compare their results with previous publications. One of them, which is cited in the manuscript, is Weghorn and Sunyaev. In fact, this work seems to be misquoted. The authors claim that Weghorn and Sunyaev "identified 147 genes with strong negative selection", but that study in fact found very few genes under significant negative selection (<10 applying a q-value cutoff of 0.1) and Weghorn and Sunyaev concluded that "the signal of negative selection is very subtle". Zapata et al., 2018 identified stronger signals of negative selection. The identified genes and functions were partly the same as in the here presented work (eg GLUT1). The authors should compare their results to these and other previous results.

Re: possible misquotation of Weghorn and Sunyaev

In the earlier version we wrote “Based on pergene estimates of negative selection inferred from the pan-cancer analysis the authors have identified 147 genes with strong negative selection. The authors have noted that among the 13 genes showing the strongest signs of negative selection there are several genes (ATAT1, BCL2, CLIP1, GALNT6, CKAP5 and REV1) that are known to promote carcinogenesis.” We wrote: “Although Weghorn and Sunyaev [65] have acknowledged that in their analyses the signals of purifying selection were exceedingly weak, they have identified a group of negatively selected genes that was enriched in cell-essential genes [66].”

In the main text of the paper of Weghorn and Sunyaev we find: “Supplementary file 7 shows the pergene estimates of negative selection inferred from the pan-cancer analysis. Here, among 13 genes with qneg <0.4, we found 6 (46%) cancer-promoting genes (ATAT1, BCL2, CLIP1, GALNT6, CKAP5, and REV1).” In their Supplementary file 7, Weghorn and Sunyaev do list 147 “Negatively selected genes (q_neg<0.6), sorted by q_neg from pan-cancer analysis” so the essence of this critical remark of the reviewers is that it is not justified to call q_neg<0.6 as strong sign and qneg <0.4 as strongest sign of negative selection. In response to this criticism, in the revised version we have corrected the text by replacing the word “strong” by “significant”.

Re: comparison of our results with the results of others

In the Results section of the revised version, we have also compared our results with those of Pyatnitskiy et al., 2015 and Zapata et al., 2018.

Note that we do not fully agree with the reviewers’ comment on GLUT1. Zapata et al., have identified 39 significantly selected genes, 14 was found to be under positive selection and 25 to be under negative selection, but GLUT1 is not listed among the significantly selected genes (see their Table 1). Nevertheless, the authors discuss several negatively selected genes (including GLUT1) that are not included in their list of significantly selected genes. To acknowledge this, in the revised version we wrote: “(It must be mentioned here that Zapata et al., [x] have also noted that the glucose transporters SLC2A1 and SLC2A8 and the lactate transporter SLC16A3 show signs of purifying selection, although these genes are not listed among the 25 genes with significant negative selection.)”

9) Furthermore, there is recent evidence that correcting for mutational signatures and nucleotide-context composition has a large impact when quantifying selection (see e.g. Zapata et al., 2018, van den Eynden et al., 2017, Martincorena et al., 2017), and this is a relevant aspect in the current lines of discussion in the context of negative selection in tumor evolution (see for example Van den Eynden et al., Nature Genetics. 2019). The authors should show that their main observations hold when the mutational signatures and/or trinucleotide context is taken into account.10) Related to this, the authors described a clustering-based method to detect genes that deviate from an average proportion of mutations (nonsynonymous, nonsense and synonymous) to infer selection. However, by only using the observed mutations (nonsyn, syn, nonsense), the underlying base-pair composition is ignored. Genes that have a high likelihood of acquiring nonsense mutations will show a deviation from the rest of the genes due to their composition and not due to selection. The authors should recalculate their metrics by performing this correction before reaching the conclusion on the number and identity of the genes.

In the Materials and methods and Results sections of the revised version, we have included detailed analyses that show that the main observations hold when mutational signatures and compositional biases are taken into account.

Use of controls11) The reviewers also indicated the lack of sufficient controls. To improve the robustness of their method, it was suggested to assess the results after varying several of the conditions. For instance, to circumvent the limitation of the lack of mutations to detect negative selection, the authors study only transcripts with more than 100 mutations. The authors should compare their results using different cut-offs for the minimum number of mutations (50,100,500), and check the performance of their method and whether their results are robust.

We have included the requested analyses in the Materials and methods section. We have used different cut-offs for the minimum number of mutations (0, 50,100,500), as suggested by the reviewers. These studies show that increasing the minimum number of mutations increases the reliability of the detection of signals of selection, but diminishes the number of novel cancer genes detected, especially those subject to negative selection. Our analyses confirm that the choice of 100 as the minimum number of somatic mutations per transcript represents an acceptable trade-off between statistical power and loss of negatively or positively selected genes.

12) Other variations that the authors should consider is to stratify data based on tumor type and mutation burden, since mixing samples with different evolutionary histories might confound the signal of negative selection. As an additional control, a reviewer suggested to perform the same analyses using the germline mutations to separate the genes into cancer specific or cell essential.

Re: stratify data based on tumor type

We agree with the reviewers that mixing tumor samples with different evolutionary histories affects the detection of signals of selection as they may differ significantly in mutation signatures (see also Response 9).

In fact, there is evidence that even in the case of a given tumor-type different samples may have evolved by different evolutionary trajectories and this is reflected in differences in patterns of mutational signatures (see PCAWG_sigProfiler_SBS_signatures_in_samples). In the revised manuscript, we have calculated the probabilities of different substitution classes of different tumor types and used metrics that have taken into account these mutation probabilities.

Re: analyses of germline mutations

To decide how the “essentiality” of genes may differ from the perspective of the tumor cell and the entire organism, in the revised manuscript we have compared the per-gene selection signals manifested in the patterns of somatic mutations of tumor cells with those manifested in the patterns of germline mutations. These comparisons have revealed that – relative to other genes – tumor essential genes identified in the present study display significantly stronger signals of purifying selection during tumor evolution than during organismal evolution, indicating that they serve cancer-specific functions in tumors.

13) An additional control to be performed by the authors was related to the origin of the mutations. The file CosmicMutantExport.tsv contains both mutation data from targeted and genome- / exome-wide screens. Targeted data should be excluded (if the authors didn't do so already). Otherwise their analysis will be highly biased towards well characterized cancer genes.

We have carried out the analyses recommended by the reviewers. These analyses have led to the following conclusions (for details see Materials and methods):

The COSMIC database of somatic mutations used in the present study contains data obtained by three main types of sequencing: whole-genome sequencing (WGS), whole-exome sequencing (WES) and targeted sequencing. The contribution of targeted screens to somatic point mutations is very restricted: only 508124 (8.3%) of the 6141650 somatic point mutations of the entire COSMIC database were identified by targeted sequencing.

To check the impact of targeted sequencing on the dataset, in some analyzes we have used somatic mutation data only from genome-wide screens, excluding those obtained by targeted sequencing. We have found that omission of the data from targeted screens had no effect on the conclusions drawn from our analyses. Several factors explain this observation. First, targeted screens usually focus on known cancer genes and they usually just reinforce the “known cancer gene” status of the targeted genes. Second, since only a small fraction of the somatic mutations originates from targeted screens their impact is limited even in the case of the targeted genes. Finally, inclusion or omission of data from targeted screens has no impact on the number and pattern of mutations of non-targeted genes identified in genome wide screens.

Statistical tests14) The reviewers also agreed that there is a general lack of statistical tests in the results. For instance, "the mean parameters of TSGs differ markedly from those of passenger genes in that rNS and rNM values are higher", but these comparisons should be done with appropriate statistical tests to assess the significance. Similar tests should be performed throughout the manuscript.

The results of statistical tests are indicated throughout the manuscript. Details of the statistical tests are reported in the Supplementary files of the respective sections.

15) A very interesting idea in the paper highlighted by the reviewers is that by combining their proposed metrics they can differentiate between oncogenes and tumor suppressors. It would be convenient to have a visual interpretation on how different genes can be only oncogenic, only tumor suppressors, or both, depending on which sites are hit. It is important to note though that similar classifiers have been developed (Schroeder et al., 2014), so it would strengthen the claims of the study to provide a comparison with those methods.

We are glad that the reviewers find it interesting that our proposed metrics can differentiate oncogenes and tumor suppressors. Despite characteristic differences in selection signals of oncogenes/tumor suppressors, we did not want to claim that these features should be used directly as classifiers of oncogenes and tumor suppressors. As we emphasized in the Background section of the original version of our manuscript, cancer driver genes may be activated/inactivated by several alternative mechanisms other than subtle mutations in their coding region. All these mechanisms have to be taken into account to decide whether a given driver gene qualifies as an oncogene or as a tumor suppressor gene. Such a complex analysis is beyond the scope of the present work.